

# Identifying required model structures to predict global fire activity from satellite and climate data

Matthias Forkel[1], Wouter Dorigo[1], Gitta Lasslop[2], Irene Teubner[1], Emilio Chuvieco[3] and Kirsten Thonicke[4]

[1] Remote Sensing Research Group, Department of Geodesy and Geoinformation, Technische Universität Wien, Gusshausstraße 27-29, 1040 Vienna, Austria
[2] Max Planck Institute for Meteorology, Bundesstr. 53, 20146 Hamburg, Germany
[3] Department of Geology, Geography and the Environment, University of Alcalá, Colegios 2, 28801 Alcalá de Henares, Spain
[4] Potsdam Institute for Climate Impact Research, Telegraphenberg A62, 14412 Potsdam, Germany

*Correspondence to*: Matthias Forkel (matthias.forkel@geo.tuwien.ac.at)

## Abstract

Vegetation fires affect human infrastructures, ecosystems, global vegetation distribution, and atmospheric composition. In particular, extreme fire conditions can cause devastating impacts on ecosystems and human society and dominate the year-to-year variability in global fire emissions. However, the climatic, environmental and socioeconomic factors that control fire activity in vegetation are only poorly understood and consequently it is unclear which components, structures, and complexities are required in global vegetation/fire models to accurately predict fire activity at a global scale. Here we introduce the SOFIA (Satellite Observations for FIre Activity) modelling approach, which integrates several satellite and climate datasets and different empirical model structures to systematically identify required structural components in global vegetation/fire models to predict burned area. Models result in the highest performance in predicting the spatial patterns and temporal variability of burned area if they account for a direct suppression of fire activity at wet conditions and if they include a land cover-dependent suppression or allowance of fire activity by vegetation density and biomass. The use of new vegetation optical depth data from microwave satellite observations, a proxy for vegetation biomass and water content, reaches higher model performance than commonly used vegetation variables from optical sensors. The SOFIA approach implements and confirms conceptual models where fire activity follows a biomass gradient and is modulated by moisture conditions. The use of datasets on population density or socioeconomic development do not improve model performances, which indicates that the complex interactions of human fire usage and management cannot be realistically represented by such datasets. However, the best SOFIA models outperform a highly flexible machine learning approach and the state-of-the art global process-oriented vegetation/fire model JSBACH-SPITFIRE. Our results suggest using multiple observational datasets on climate, hydrological, vegetation, and socioeconomic variables together with model-data integration approaches to guide the future development of global process-oriented vegetation/fire models and to better understand the interactions between fire and hydrological, ecological, and atmospheric Earth system components.

## 1 Introduction

Wildland fires are important disturbances in the Earth system which affect ecosystems, global vegetation distribution, infrastructures and human assets, and contribute to atmospheric composition through the release of aerosols, reactive trace gases, and greenhouse gases (Bowman et al., 2009). The ignition and spread of fires in ecosystems depend on the availability and properties of fuel (i.e. biomass loads, composition, and moisture content), weather conditions, and human activities (Krawchuk and Moritz, 2011; Moritz et al., 2012). Human activities have a predominant role in fire ignition, and affect fire behaviour either directly through fire suppression or indirectly through land management and landscape structure (Bowman et al., 2011). Burned area is a key variable to describe fire impacts on ecosystems and vegetation distribution (Bond, 2005), and



to estimate fire emissions (Seiler and Crutzen, 1980). Recent estimates of average yearly global burned area range from 3.3 to 3.8 million km$^2$ (Chuvieco et al., 2016; Giglio et al., 2013) which is around 4 % of the global vegetated area (Randerson et al., 2012). On a global scale, burned area shows only a small inter-annual variability which is stabilized by the annual recurrent patterns of very large burned areas in African savannahs (Giglio et al., 2013). However, in boreal, temperate and tropical regions, burned area has a very high inter-annual variability which is strongly linked to the variability in atmospheric

circulation patterns such as to El Niño events (Andela and van der Werf, 2014; Balzter et al., 2005; Giglio et al., 2013; Hess et al., 2001). Such years with extreme fire activity in forests can cause large emissions of greenhouse gases (Kasischke and Bruhwiler, 2002; Vinogradova et al., 2015), dominate together with peatland fires the inter-annual variability of global fire emissions (Page et al., 2002; van der Werf et al., 2006, 2010), and thus strongly affect atmospheric composition (Langenfelds et al., 2002; Simpson et al., 2006). Consequently, to realistically assess current and future fire impacts on the Earth system it

is necessary to realistically simulate the spatial and temporal variability of burned area in Earth system models (ESMs) or dynamic global vegetation models (DGVMs).

Satellite observations of burned area or of active fires can be used to develop, evaluate, or improve global vegetation/fire models in order to better simulate fire impacts on terrestrial vegetation distribution and global carbon cycling (Poulter et al., 2015b). The first fire modules within DGVMs like GlobFIRM (global fire model, Thonicke et al. (2001)) were developed in

the late 1990s and early 2000s in absence of global burned area datasets as reference. Later, regional satellite-derived burned area datasets were used to evaluate new developed global fire models such as SPITFIRE (spread and intensity of fire, Thonicke et al. (2010)). The first global burned area datasets were derived in the mid-2000s from several optical satellite sensors such ATSR (Simon et al., 2004), MODIS (Roy et al., 2005), and SPOT (Grégoire et al., 2003; Tansey et al., 2008). The increasing temporal coverage of satellite observations enables to derive multi-year harmonized burned area datasets, e.g. the products

from the Global Fire Emissions Database (GFED) (Giglio et al., 2010, 2013) or from the European Space Agency (ESA) Climate Change Initiative (CCI) on fire (Fire CCI) (Chuvieco et al., 2016). Consequently, global burned area datasets are nowadays commonly used within model benchmarking systems (Kelley et al., 2013) or to evaluate further developments in process-oriented fire models (Kloster et al., 2010; Lasslop et al., 2014; Yue et al., 2014). Additionally, satellite and other observation-based datasets on climate, vegetation, and socioeconomic factors are increasingly used to identify controls on fire

activity and to develop and parameterize empirical fire models (Aldersley et al., 2011; Archibald et al., 2009; Bistinas et al., 2014; Chuvieco et al., 2008; Lasslop et al., 2015; Lehsten et al., 2010; Moritz et al., 2012). For example, SIMFIRE (simple fire model) uses empirical relations to predict annual fire frequency from vegetation conditions, fire weather conditions, and population density (Knorr et al., 2014). The development of SIMFIRE followed for the first time in global fire modelling a model-data integration (or model-data fusion) approach where model parameters are estimated based on observations within

a formal framework (Keenan et al., 2011; Williams et al., 2009). Moreover, model-data integration encompasses a continuous cycle from the definition of model structures (i.e. predictor variables and functional relations), estimation of model parameters, generalization or upscaling of the model, evaluation of model results, to model application and potentially back to a reformulation of the model structure (Williams et al., 2009). Global process-oriented vegetation/fire models use currently various model structures and associated complexities to represent fuel moisture, fuel loads, ignitions, fire suppression, and fire

spread and it is not yet clear what structures are needed to optimally simulate fire activity at a global scale. (Hantson et al., 2016). Consequently, model-data integration approaches might provide insight in the required model structures to predict fire activity at regional to global scales.

Satellite observations provide several datasets on vegetation and moisture conditions that can be used as predictor variables in global empirical fire models. Although time-variant biomass datasets would be the first choice to represent fuel loads in

empirical fire models because the availability of fuel is a prerequisite for fire activity (Krawchuk and Moritz, 2011), current global biomass maps are static (Avitabile et al., 2016; Saatchi et al., 2011; Thurner et al., 2014) and thus provide only limited information for fire modelling. Consequently, other proxies of vegetation biomass such as model-based net primary production



(NPP) (Bistinas et al., 2014; Moritz et al., 2012), satellite-derived vegetation cover (Bistinas et al., 2014; Lehsten et al., 2010), or the fraction of absorbed photosynthetic active radiation (FAPAR) (Knorr et al., 2014) have been used as proxies for fuel

loads in global empirical fire models. As an alternative, satellite retrievals of vegetation optical depth (VOD) might be used as proxy for fuel loads. VOD is a vegetation variable that is derived from active or passive microwave satellite observations and is related to vegetation density and water content (Liu et al., 2011b, 2013a, Vreugdenhil et al., 2016a, 2016b). VOD has a higher sensitivity to forest biomass than FAPAR (Andela et al., 2013) and was used to estimate temporal changes in biomass (Liu et al., 2015). Thus VOD might be a valuable predictor variable for the biomass-driven variability in fire activity. Satellite

datasets of surface soil moisture might be valuable proxies for the moisture of surface fuels in empirical fire models (Krueger et al., 2015, 2016) because they represent the top ~3 cm of the soil (Dorigo et al., 2015). So far, it has not yet been systematically tested what variables provide the best information for empirical fire models to represent fuel loads, fuel moisture, or fire weather conditions.

In this study, we aim to identify what model structures and complexities are required to predict global spatial patterns and

temporal dynamics of burned area. For this purpose, we develop a new, simple empirical fire modelling concept, called SOFIA (Satellite Observations for FIre Activity) where observational datasets on land cover, climate conditions, soil moisture, vegetation state, and socioeconomics (population density and an indicator of socioeconomic development) are used to predict spatial patterns and temporal dynamics of burned area. Based on the philosophy of model-data integration, we generated several different candidate model structures,, and optimized and evaluated each model against observed burned area time

series. Additionally, we simulated global burned area with an alternative machine learning approach (random forest) and with a process-oriented vegetation/fire model (JSBACH-SPITFIRE) to compare the performance of the derived SOFIA models with two independent state-of-the art modelling approaches. The random forest approach was used to test if a modelling concept that allows more flexible model structures than SOFIA results in higher performances. However, SOFIA has the advantage over random forest that it could easily be transferred or implemented to global process-oriented vegetation/fire

models. The SPITFIRE fire module within the JSBACH (Jena scheme for biosphere-atmosphere coupling in Hamburg) land surface model (Lasslop et al., 2014; Rabin et al., 2016) was used to compare SOFIA results with a global vegetation/fire model and to provide suggestions for further model development.

We first describe the observational datasets and the derived variables that we used to develop SOFIA models (Sect. 2). Secondly, we describe the SOFIA modelling concept, and the JSBACH-SPITFIRE and random forest modelling approaches

(Sect. 3). In Section 4, we first present the global performance and complexity of SOFIA models (Sect. 4.1) and how several structural components contribute to model performance (Sect. 4.2). Then we compare the best performing SOFIA models globally against random forest and JSBACH-SPITFIRE (Sect. 4.3) and apply the best SOFIA model to explore spatial patterns of controlling factors for fire activity (Sect. 4.4). Finally, we discuss our results with respect to previous work about controls on fire (Sect. 5.1) and suggest the use of multiple datasets and model-data integration approaches to improve global process-

oriented vegetation/fire models (Sect. 5.2).

## 2 Datasets and derived variables for model development

We used various satellite and climate datasets to predict burned area using SOFIA. Therefore we used datasets of global monthly burned area as response variable and several datasets on land cover, climate, soil moisture, vegetation state, and socioeconomic factors as predictor variables in model development (Table 1). Several other predictor variables were derived

from each dataset such as pre-fire annual mean values of climate, soil moisture and vegetation state, or aggregated land cover properties because these variables might provide more predictive power for fire modelling than instantaneous values (Table 1). We based the analysis mostly on long-term harmonized or multi-satellite merged datasets in order to derive appropriate SOFIA model structures for long-term (i.e. decadal) variability in burned area that is covered for the period 1995–2015 of the




GFED burned area dataset (Giglio et al., 2013). Although state-of-the-art single satellite sensors may provide information in

higher quality, the use of such datasets would restrict the temporal coverage of the analysis. Given the common coverage of the used predictor datasets, the analysis was consequently performed for the period 1997–2011, on monthly time steps, and on a 0.25° spatial resolution. This is also comparable to common application domains of state-of-the art global process-oriented vegetation/fire models (Rabin et al., 2016). Datasets were temporally and spatially aggregated or interpolated if they originally differed from these temporal and spatial resolutions (details in the following sections for each dataset).


**Table 1: Description of used datasets and derived predictor variables.**

| Dataset | Derived variables | Description |
|---|---|---|
| **Burned area (response variable)** | | |
| GFED | GFED burned area version 4 (Giglio et al., 2013), http://www.globalfiredata.org | |
| | GFED.BA | Fractional burned area of a 0.25° grid cell, used for optimization of SOFIA models |
| Fire CCI | ESA Fire CCI burned area version 4.1 (Chuvieco et al., 2016), http://cci.esa.int/data | |
| | CCI.BA | Fractional burned area of a 0.25° grid cell, independent dataset in evaluation |
| **Predictor variables** | | |
| **Land cover / plant functional types (PFTs)** | | |
| Land cover CCI | ESA land cover_cci version 1.6.1, http://maps.elie.ucl.ac.be/CCI/viewer/index.php Land cover classes were translated to fractional coverages of plant functional types (PFTs) in 0.25° grid cells (Poulter et al., 2015a) (Table A 1). | |
| | CCI.LC.Tree.BE | Broadleaved evergreen trees |
| | CCI.LC.Tree.BD | Broadleaved deciduous trees |
| | CCI.LC.Tree.NE | Needle-leaved evergreen trees |
| | CCI.LC.Tree.ND | Needle-leaved deciduous trees |
| | CCI.LC.Shrub.BE | Broadleaved evergreen shrubs |
| | CCI.LC.Shrub.BD | Broadleaved deciduous shrubs |
| | CCI.LC.Shrub.NE | Needle-leaved evergreen shrubs |
| | CCI.LC.Herb | Natural grass and herbaceous vegetation |
| | CCI.LC.Crop | Cropland and managed grass |
| | CCI.LC.HrbCrp | Natural and managed grass and croplands = Herb + Crop |
| | CCI.LC.Tree | Coverage of trees = Tree.BE + Tree.BD + Tree.NE + Tree.ND |
| | CCI.LC.Shrub | Coverage of shrubs = Shrub.BE + Shrub.BD + Shrub.NE |
| | CCI.LC.Broadleaf | Coverage of broadleaved vegetation = Tree.BE + Tree.BD + Shrub.BE + Shrub.BD |
| | CCI.LC.Needleleaf | Coverage of needle-leaved vegetation = Tree.NE + Tree.ND + Shrub.NE |
| **Climate and soil moisture** | | |
| CRU | CRU TS3.23 climate data (Harris et al., 2014), https://crudata.uea.ac.uk/cru/data/hrg/cru_ts_3.23 | |
| | CRU.T.orig | Mean monthly air temperature (°C) |
| | CRU.T.annual | Mean air temperature in the actual month and the 12 months before a fire |
| | CRU.WET.orig | Monthly number of wet day |
| | CRU.WET.annual | Mean number of wet days in the actual month and the 12 months before a fire |
| | CRU.DTR.orig | Mean monthly diurnal temperature range (K) |
| GPCC | GPCC precipitation version 7, http://dx.doi.org/10.5676/DWD_GPCC/FD_M_V7_050 | |
| | GPCC.P.orig | Monthly total precipitation (mm) |
| | GPCC.P.annual | Total precipitation in the actual month and the 12 months before a fire |
| Soil moisture CCI | ESA soil moisture_cci version 02.3, http://cci.esa.int/data | |
| | CCI.SM.orig | Mean monthly surface soil moisture |
| | CCI.SM.annual | Mean surface soil moisture in the actual month and the 12 months before a fire |
| **Vegetation state** | | |
| GIMMS FAPAR | GIMMS fraction of absorbed photosynthetic active radiation version 3g (Zhu et al., 2013), http://cliveg.bu.edu/modismisr/lai3g-fpar3g.html | |
| | GIMMS.FAPAR.orig | Mean monthly FAPAR |
| | GIMMS.FAPAR.pre | FAPAR in the month before a fire |
| | GIMMS.FAPAR.annual | Mean FAPAR in the 12 months before a fire |
| VOD | Multi-sensor harmonized vegetation optical depth (Liu et al., 2011b, 2015), provided by Y. Liu | |
| | Liu.VOD.orig | Mean monthly VOD |
| | Liu.VOD.pre | VOD in the month before a fire |
| | Liu.VOD.annual | Mean VOD in the 12 months before a fire |
| **Socioeconomics** | | |
| PD | GRUMP population density version 1 (years 1990, 1995, 2000) (Balk et al., 2006), http://dx.doi.org/10.7927/H4R20Z93 | |
| | PD.med | Population density (individuals km-2), median estimate of three methods for temporal inter- and extrapolation (spline interpolation, linear interpolation, interpolation with last value as constant) |
| NLDI | Night light development index (year 2006) (Elvidge et al., 2012), http://ngdc.noaa.gov/eog/dmsp/download_nldi.html | |
| | NLDI | Night light development index, but grid cells without night lights or population set to 1.01 |




### 2.1 Burned area

Global monthly burned area data was taken from the Global Fire Emissions Database (GFED) (Giglio et al., 2013) and the
ESA Fire CCI datasets (Chuvieco et al., 2016). GFED version 4 provides monthly burned area time series on a 0.25° spatial
resolution for the period 1995-2015 based on a combination of the MODIS burned area product (from 2000 onwards) with
active fire observations from VIRS (Visible and Infrared Scanner) and ATSR (Along-Track Scanning Radiometer) (before
2000) (Giglio et al., 2013). Fire CCI version 4.1 provides burned area time series on 0.25° spatial resolution for the period
2005-2011 based on a combination of MERIS data and MODIS thermal anomalies (Alonso-Canas and Chuvieco, 2015;
Chuvieco et al., 2016). Because of the longer temporal coverage, the GFED dataset was used as the response variable in model
development and for model evaluation. The Fire CCI dataset was used as an independent burned area dataset in model
evaluation. Differences between the two datasets reflect the uncertainty in satellite-derived burned area. For both datasets
burned area is expressed as the fractional burned area of a 0.25° grid cell.

### 2.2 Land cover

Land cover data was taken from the ESA land cover CCI product which provides three global land cover maps at 300 m spatial
resolution covering the epochs 1998-2002, 2003-2007, and 2008-2012. We did not use the original land cover classification
of the maps but translated land cover classes into plant functional types (PFTs) to be comparable with the classification used
in global vegetation models (Poulter et al., 2011). The translation followed largely the rules by Poulter et al. (2015a) with some
modifications to avoid coverage of broad-leaved evergreen trees and shrubs in boreal and arctic regions (Table A 1). The
following nine PFTs were derived: broadleaved evergreen tree and shrub (Tree.BE, Shrub.BE), broadleaved deciduous tree
and shrub (Tree.BD, Shrub.BD), needle-leaved evergreen tree and shrub (Tree.NE, Shrub.NE), needle-leaved deciduous tree
(Tree.ND), natural grass or herbaceous vegetation (Herb), and managed grasslands or crops (Crop). The land cover maps were
spatially aggregated and expressed as the fractional coverage of PFTs within a 0.25° grid cell.
We further aggregated the coverage of PFTs within each 0.25° grid cell to the total coverages of trees (Tree = sum of all tree
PFTs, Table 1), shrubs (Shrub), and herbaceous vegetation including croplands (HrbCrp = Herb + Crop). To potentially
characterise fuel types based on the dominant leaf type, PFTs were further aggregated into needle-leaved (Needleleaf) and
broadleaved vegetation (Broadleaf) vegetation.
As land cover distribution is affected by fires, the land cover maps may regionally contain effects of past fires. Consequently,
it can happen that fire activity is explained with the impact of the actual fire activity already present in a land cover map. We
tried to reduce this effect by shifting the land cover maps by 2 years. This means that the map for the epoch 1998–2002 is used
for the years ≤ 2004, the map for the epoch 2003–2007 for the period 2005–2009, and the map for the period 2008–2012 for
the years ≥ 2010. However the three maps have only marginal temporal differences so that the impact of assigning land cover
maps to certain years is rather small.

### 2.3 Climate

Monthly data of mean air temperature, diurnal temperature range (DTR), and monthly number of wet days was taken from the
Climate Research Unit (CRU) TS3.2 dataset (Harris et al., 2014). These datasets provide monthly climate time series at 0.5°
resolution based on spatially interpolated weather station observations. Precipitation was taken from the Global Precipitation
Climatology Center (GPCC) version 7 dataset (Schneider et al., 2015). All climate datasets were resampled to 0.25° using the
nearest neighbour method in order to avoid smoothing of climate anomalies through alternative resampling methods such as
bilinear interpolation.
We used the monthly values and long-term conditions of climate datasets as predictor variables (Table 1). As long-term
conditions, we computed the mean temperature, mean diurnal temperature range, mean number of wet days, and the total
precipitation of the actual month and the 12 months before a fire.



### 2.4 Soil moisture

Surface soil moisture was taken from the ESA CCI soil moisture dataset (version 02.3 COMBINED) which is based on a merging of soil moisture products from various active and passive satellite sensors (Dorigo et al., 2015; Liu et al., 2011a, 2012). The dataset represents the upper soil layer (~ 2cm) and is available on a 0.25° spatial resolution and daily time step for the period 1979-2015. The long-term dynamic of the soil moisture dataset is consistent and environmentally plausible as demonstrated in a comparison with precipitation, soil moisture, and normalized difference vegetation index trends from

independent datasets or land surface models (Albergel et al., 2013; Dorigo et al., 2012).

As soil moisture cannot be accurately retrieved underneath dense (tropical) forests, estimates are not available in all regions and thus the dataset has spatial gaps. We excluded such grid cells in the full analysis. Soil moisture time series were aggregated to monthly mean values. Temporal gaps in soil moisture time series were filled using a season-trend regression model as described in Forkel et al. (2013) and based on Verbesselt et al. (2010a, 2010b) but without accounting for breakpoints.

However, some years in some grid cells were excluded from the entire analysis if soil moisture estimates were only available for less than 3 months within this year.

We used the monthly soil moisture values and long-term soil moisture conditions as predictor variables (Table 1). Long-term soil moisture conditions were computed as the mean soil moisture of the actual month and the 12 months before a fire.

### 2.5 Vegetation state

To account for effects of vegetation phenology, biomass, or vegetation water content on fire activity, we used the GIMMS3g FAPAR (Zhu et al., 2013) and a VOD dataset (Liu et al., 2011b). GIMMS3g FAPAR is a long-term multi-sensor merged dataset of FAPAR and is based on the GIMMS3g NDVI (Normalized Difference Vegetation Index) dataset with a spatial resolution of 1/12° and a temporal resolution of 16 days for the period 1981 to 2012 (Pinzon and Tucker, 2014). GIMMS3g FAPAR was aggregated to 0.25° spatial resolution and averaged to monthly time steps. VOD by Liu et al. (2011b) is a long-

term harmonized dataset from several passive microwave sensors. The VOD dataset has a spatial resolution of 0.25° and a monthly temporal resolution for the period 1988-2012.

Permanent gaps in FAPAR or VOD time series (mostly gaps occurring in winter in northern latitudes) were filled with the minimum value of each time series (Forkel et al., 2015) and remaining gaps were filled using the season-trend regression model (Forkel et al., 2013).

We used the monthly FAPAR or VOD values of the month before a fire as predictor variables because vegetation of the actual month is likely affected by the fire event which we aim to explain. Additionally, we computed mean FAPAR and VOD of the 12 months before a fire as long-term vegetation state predictor variables.

### 2.6 Socioeconomic variables

We used satellite-based datasets on population density and socioeconomic development as predictor variables for burned area.

Population density (PD) was taken from the Global Rural-Urban Mapping Project (GRUMP) V1 dataset (Balk et al., 2006). This dataset is based on (sub-)national population statistics, satellite observations of night-time lights, and the spatial distribution of cities to provides estimates of population density on a 1 km grid for the years 1990, 1995, and 2000. The dataset was aggregated to 0.25°. The dataset was temporally interpolated between 1990 and 2000 and extrapolated between 2000 and 2011 for each grid cell to achieve a full coverage for the period 1997-2011. The interpolated time series is the median estimate

from three interpolation methods (repeating last value as a constant, linear interpolation, spline interpolation). This allowed to make use of the temporal information of the population density dataset.

As an indicator for socioeconomic development, we used the night light development index (NLDI) (Elvidge et al., 2012). NLDI is derived from satellite observations of light emissions during night and an independent estimate of population density. NLDI ranges between 0 (light emissions equally distributed among people, highest development) and 1 (light emissions



concentrated at one person, lowest development). NLDI is highly correlated with electrification rates and the human development index (Elvidge et al., 2012). The dataset is available on a 0.25° spatial resolution for the year 2006. NLDI is not available for grid cells without population or without detected night lights, which introduces gaps in the global NLDI map. We filled these gaps with a value of 1.01 (indicating very low development or natural ecosystems) in order to not introduce spatial gaps of the NLDI dataset in the empirical modelling of burned area.

## 3 Modelling approaches and model-data analysis

### 3.1 SOFIA modelling concept

SOFIA is an empirical modelling concept that allows to test several alternative model structures to predict fractional burned area. The basic structure of SOFIA fire models is inspired by SIMFIRE (simple fire model) which uses empirical relationships to estimate fire frequency from vegetation (i.e. FAPAR), fire weather conditions, and socioeconomic variables (Knorr et al.,

2014). In SOFIA we generalize the SIMFIRE approach by using and testing several alternative predictor variables as controls for fire activity. Each SOFIA model structure is based on the assumption that potentially the entire vegetated area can burn but burning is actually suppressed by several controlling factors:

$$BA_t = \sum_{g=1}^{g=N} A_g * f_{g,t} \tag{1}$$

where $BA$ is the fractional burned area of a grid cell at time step $t$, $A_g$ is the fractional coverage of land cover group $g$, and $f_g$ is

a factor that controls fire spread [0 = fully suppressed burning and 1 = un-constrained burning] for a specific land cover group. Land cover groups $g$ can for example be classified according to growth forms (trees, shrubs, grasses, crops), plant functional types (PFTs), or any other potentially meaningful separation of land cover. The factor $f_g$ is a product of individual functions that represent climatic, environmental, and socioeconomic controls on fire:

$$f_g = \prod_{i=1}^{i=N} f(x_{i,g}) \tag{2}$$

$$f(x_{i,g}) = \min\left[1, \frac{max_{g,i}}{1+e^{\left(-sl_{i,g} \times \left(x - x0_{i,g}\right)\right)}}\right] \tag{3}$$

where $x$ is the value of an environmental or socioeconomic variable $i$; and $max$, $sl$ and $x0$ are parameters of a logistic function. We used the minimum value from 1 and the logistic function, and included $max$ as a free parameter to allow the representation of exponential relationships within the basic structure of logistic functions. Parameters of the logistic functions can be either defined per vegetation cover group or as global parameters. Variables $x$ can be for example vegetation state variables such as

FAPAR or VOD to represent fuel loads, climate variables such as the number of wet days or diurnal temperature range to represent fire weather conditions, and socioeconomic variables such as population density or NLDI to represent human effects on fire activity. Consequently, the development of an actual SOFIA model requires two steps, namely the definition of a model structure (i.e. selection of potentially appropriate predictor variables, Sect. 3.2) and the estimation of the model parameters (Sect. 3.3).

SOFIA models allows to reproduce the typical right-tailed distribution of burned area (i.e. many grid cells and months with no burned area in comparison to relatively few grid cells and months with fire activity) and to explore underlying response functions of fire activity to environmental or socioeconomic variables that can take step-wise, linear, sigmoidal, or exponential shapes depending on the parameters of the logistic functions (Figure 1). Similar model structures like SOFIA where a response variable is controlled by a product of several functions have been previously applied in environmental modelling for example

in light-use efficiency models to simulate NPP (Cai et al., 2014; Nemani et al., 2003) or in phenology models to simulate leaf development (Forkel et al., 2014; Jolly et al., 2005; Stöckli et al., 2011). The values of the control functions can also be used to map the spatial covariation of burned area with environmental or socioeconomic variables. Such a mapping of controls was previously done for plant productivity (Nemani et al., 2003) and phenology (Forkel et al., 2014; Jolly et al., 2005) based on



red-green-blue (RGB) composite maps. Here we will demonstrate how this approach can be used to investigate spatial patterns

of climatic, environmental and socioeconomic controls on fire activity.

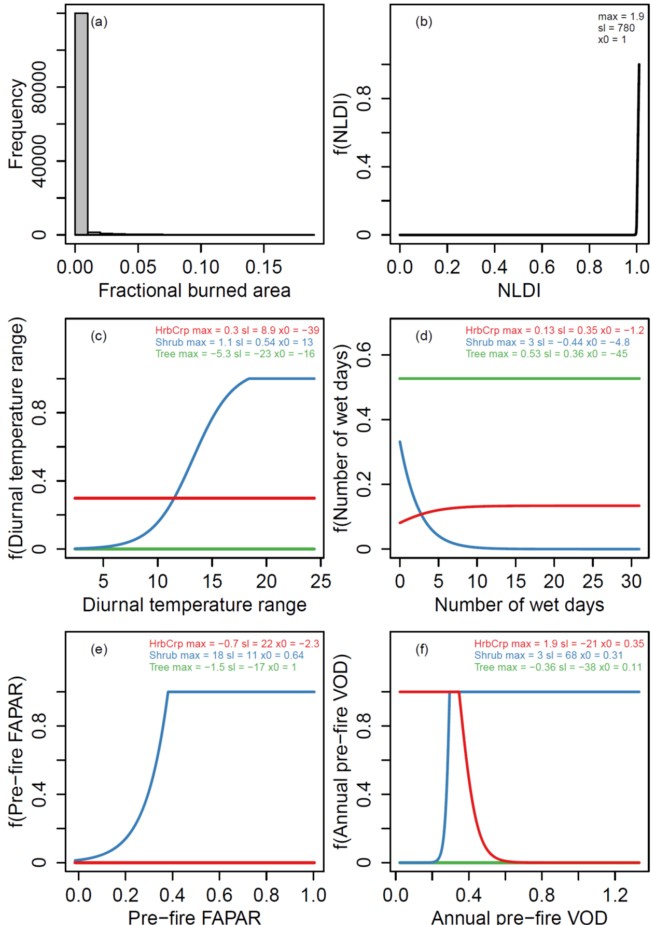

**Figure 1: Example of a SOFIA model structure with three land cover groups (i.e. herbaceous vegetation and crops, shrubs, trees) and five controls on fire activity. The example is taken from the SOFIA model SF.124421 (Table 2). (a) Histogram of the simulated**
**fractional burned area. Response functions of fractional burned area on (b) night light development index, (c) diurnal temperature range, (d) number of wet days, (e) fraction of absorbed photosynthetic active radiation in the month before a fire, and (f) mean vegetation optical depth in the 12 months before a fire. max, sl, and x0 are parameters of the logistic functions.**

### 3.2 Structural components for SOFIA models

Several alternative model structures need to be defined in order to identify appropriate components of SOFIA models to predict

burned area. Each SOFIA model uses a specific land cover grouping scheme and several controlling factors for fire activity.

We tested different land cover grouping schemes to assess the required complexity of SOFIA models to regionalize model

parameters. As grouping schemes we either used growth forms ("GrowthForm" including the variables *Tree*, *Shrub*, and

*HrbCrp*; Table 1), growth forms with crops separated from herbaceous vegetation ("GrowthFormCrop" including *Tree*, *Shrub*,

*Herb*, *Crop*), leaf types ("LeafType" including *Needleleaf*, *Broadleaf*, *Herb*, *Crop*), or PFTs ("PFT" using the nine PFTs).

Differences between GrowthForm and GrowthFormCrop will allow to assess if a separation of croplands from herbaceous

vegetation is necessary to explain fire activity. The LeafType grouping scheme may potentially be useful because needles



usually decompose slower than broadleaves and thus form larger pools of litter fuel. Differences between GrowthFormCrop and LeafType allow to assess if model parameters should be separated rather by growth form or by leaf type. The PFT land

cover grouping scheme is finally used to assess if the interaction of growth forms and leaf types is required to regionalize model parameters.

We defined five controlling factors on fire activity and assigned several corresponding predictor variables to each controlling factor to evaluate required structural components of SOFIA models:

1.  *Human influences* represent potential relations between socioeconomic indicators and burned area. As predictor variables

we used either population density with a global parameter set (PD), NLDI with a global parameter set (NLDI), or NLDI with parameters that vary per land cover group (NLDI.g).

2.  *Temperature effects* represent potential relations between diurnal temperature range (CRU.DTR.orig) or long-term air temperature (CRU.T.annual) and burned area.

3.  *Direct wetness effects* represent the obvious suppression of fire activity by wet conditions. We included either the actual-

month number of wet days (CRU.WET.orig), precipitation (GPCC.P.orig), or surface soil moisture (CCI.SM.orig).

4.  *Direct vegetation effects* represent potential relations between the pre-fire vegetation state and burned area. Therefore we either used previous-month FAPAR (GIMMS.FAPAR.pre) or VOD (Liu.VOD.pre) as predictor variables.

5.  *Long-term wetness or vegetation effects* represent potential relations between pre-fire averaged conditions of wetness or vegetation variables and burned area. Several reasons exist to test long-term controlling factors as structural components

of SOFIA models. Firstly, long-term conditions of precipitation and soil moisture are strongly linked to plant productivity especially in semi-arid ecosystems and thus might represent variations in vegetation and fuel loads. Secondly, long-term conditions of FAPAR and VOD are more closely related to vegetation coverage or biomass and thus might better represent fuel loads than the actual monthly values. As predictor variables for long-term conditions, we used aggregated values from the 12 months before a fire for the number of wet days (CRU.WET.annual), precipitation (GPCC.P.annual), soil moisture

(CCI.SM.annual), FAPAR (GIMMS.FAPAR.annual), or VOD (Liu.VOD.annual).

We also allowed that a certain controlling factor is not included in a SOFIA model to test if this controlling factor is generally needed in the SOFIA model. This setup of controlling factors and associated predictor variables allows the definition of several candidate model structures (Table A 2). For example, the SOFIA model SF.124421 (the coding is described in Table A 2) used growth forms as land cover grouping scheme, NLDI for human influences, diurnal temperature range as temperature effect,

the number of wet days as direct wetness effect, previous-month FAPAR as direct vegetation effect, and long-term pre-fire VOD as long-term vegetation effect (Figure 1). The model structure determines the complexity which we assess here based on the number of controlling factors within a SOFIA model and on the number of parameters $N$ in a model ($N$ = number of controlling factors * number of land cover groups * 3 parameters). We required that SOFIA models included at least 3 controlling factors and have less than 100 parameters. This results in 2712 candidate SOFIA models. We optimized and

evaluated 95 randomly selected models from the set of candidate models (Table A 2). Although this selection does not allow a full factorial assessment of controlling factors and predictor variables in SOFIA models, it is a trade-off between computational feasibility and an assessment of the tendency of a factor regarding model performance.

### 3.3 Optimization and evaluation of SOFIA models

#### 3.3.1 Model optimization

After the definition of candidate structures for SOFIA models, parameters for each controlling function need to be estimated for each model to achieve an optimal performance. The parameters $p$ of the logistic functions of each controlling factor were estimated by minimizing the sum-of-squared error (SSE) between the monthly observed (*obs*) and simulated (*sim*) fractional burned area:

$$SSE = \sum_{i=1}^{i=N}(sim_i - obs_i)^2 \hspace{4cm} (4)$$



where *i* is an index over grid cells and months. We also tested alternative cost functions in the optimization which transform
      burned area data, explicitly account for variance or which were based on burned area anomalies instead of absolute area in
      order to potentially better predict the variability of observed burned area (Table A 3).

      The minimization of SSE was performed by applying a genetic optimization algorithm. The used algorithm (GENOUD, genetic
      optimization using derivatives) combines a global search algorithm (i.e. genetic optimization) with a local search algorithm
(i.e. BFGS) (Mebane and Sekhon, 2011). GENOUD was already previously used to estimate parameters in a dynamic global
      vegetation model (Forkel et al., 2014). Here we applied GENOUD by using 500 individuals (i.e. parameter sets) per generation,
      and allowed the algorithm to run for maximum 30 generations. The parameter sets of the first generation were generated
      randomly. The second generation is generated by using several operators to clone, mutate, and crossover the best parameter
      sets of the first generation (Mebane and Sekhon, 2011). The BFGS local search algorithm was first used starting from the best
parameter set that evolved in the 28[th] generation in order to avoid a too fast convergence of the algorithm towards a local
      optimum.

### 3.3.2 Model selection and evaluation

We selected the best performing SOFIA models from all optimized candidate models based on the Akaike Information
Criterion (AIC) (Burnham and Anderson, 2002). AIC is a metric to empirically infer appropriate model structures from several
candidate models based on performance (in terms of SSE) and by penalizing for model complexity (in terms of the number of
      model parameters *N*):

$$AIC = 2 \times N - 2 \times log(e^{-SSE}) \tag{5}$$

Given a certain performance threshold, the best model has the lowest AIC value (Burnham and Anderson, 2002).

To evaluate the simulated spatial-temporal patterns and temporal dynamics of fractional burned area, we used the index of
agreement (IoA) and the fractional variance (FV) (Janssen and Heuberger, 1995):

$$IoA = 1 - \frac{\sum_{i=1}^{i=N}(obs_i - sim_i)^2}{\sum_{i=1}^{i=N}(|sim_i - \overline{obs}| + |obs_i - \overline{obs}|)^2} \tag{6}$$

$$FV = \frac{\sigma_{sim} - \sigma_{obs}}{0.5 \times (\sigma_{sim} + \sigma_{obs})} \tag{7}$$

where $\overline{obs}$, $\overline{sim}$ and $\sigma_{obs}$, $\sigma_{sim}$ are the means and variances of the observations and simulations, respectively. IoA ranges
between 0 (worst fit) and 1 (best fit) and is an overall efficiency metric that is sensitive to correlation and bias. FV ranges
between -2 and 2 (best agreement at 0) where negative values indicate an underestimation and positive values an overestimation
      of the observed variance.

### 3.3.3 Data sampling for model optimization and evaluation

We sampled several grid cells from the global datasets (0.25° resolution) to optimize and evaluate all candidate SOFIA models.
A sampling of grid cells is necessary to retain enough independent data for evaluation of SOFIA models and because
optimization of all SOFIA models on the entire global datasets with 0.25° spatial resolution, monthly time steps, and 15 years
      was computationally not feasible. However the sampling needs to represent the global spatial patterns and the entire statistical
      distribution of burned area, including extreme fire events. Therefore we performed a sampling of grid cells stratified by regions
      (representing biomes) and by the maximum annual burned area in each grid cell (representing extreme fire years) (Figure A
      1). For each region, we computed quantiles of the maximum annual burned area to cover the regional statistical distribution of
      fire activity. For each regional quantile class (e.g. quantiles 0.01 to 0.02), we sampled randomly grid cells while maximizing
the spatial distance between sampled grid cells to cover a wide geographic range. In total, 3161 grid cells were sampled with
      most of the cells in savannahs and tropical croplands (n = 953, largest region) and fewest cells in boreal needle-leaved
      deciduous forests (n = 135, smallest region) (Figure A 1 b). Consequently, the sampled grid cells are representative for the
      global statistical distributions (Figure A 1 c-e) and for spatial patterns of fire activity (Figure A 1 f).





The sampled grid cells were further divided into a subset for optimization (60% of the sampled grid cells) and for evaluation (40% of the sampled grid cells). The time periods in both subsets was further divided according to years for which the monthly data was used for optimization (even years in 1998 to 2010) and for which the monthly data was used for evaluation (uneven years in 1997 to 2011). We used every second year for optimization or evaluation to avoid that potential temporal changes in the quality of multi-sensor satellite datasets (e.g. burned area, soil moisture, FAPAR, and VOD) affect the evaluation of model

results. Based on this sampling scheme, 1817 grid cells (= 152.628 monthly observations in even years) were used for optimization and 1212 grid cells (= 116.352 monthly observations in uneven years) were used for evaluation. Note that fewer observations were used in the optimization and evaluation subsets for the comparison against the Fire CCI burned area dataset because this dataset starts only in 2005.

    We applied the best-performing SOFIA models to all global 0.25° grid cells to compare them globally with the GFED and

CCI burned area datasets and with JSBACH-SPIFIRE. From these global results, we compared maps of mean annual burned area and regional statistical distributions and temporal dynamics of annual burned area for the period 2005 to 2011. Therefor we aggregated burned area from the datasets and from the best SOFIA models to the coarse spatial resolution of JSBACH (1.875*1.875°).

### 3.4 Data-driven fire modelling with random forest

We used the random forest machine learning approach to evaluate if the basic structure of SOFIA models is flexible enough to predict burned area or if a more flexible modelling approach can reach higher performances. Random forest is a regression approach that can consider non-linear, non-monotonic and abrupt, and non-additive relations between multiple predictor variables and a response variable (Breiman, 2001). Random forest is an ensemble of multiple regression trees that are trained based on the response variable. Each tree uses a randomly selected set of predictor variables and data points (Breiman, 2001).

Random forest was already previously applied to identify controls on vegetation dynamics and on fire activity (Aldersley et al., 2011; Archibald et al., 2009). We used 500 trees per random forest. For the training of the random forest, we used the same data subset that was also used to optimize SOFIA models (Sect. 3.3.3). The analysis was performed using the randomForest package in R (Liaw and Wiener, 2002).

    We performed three different random forest model experiments. The model experiment RF1 used all predictor variables from

Table 1 to explore the potential performance of the used datasets to predict burned area. The model experiment RF2 used all predictor variables except the variables from the soil moisture dataset in order to apply random forest globally and to compare the results with SOFIA independent of the spatial gaps of the soil moisture dataset. The model experiment RF.124421 uses the same predictor variables as the SOFIA model SF.124421 (i.e. CCI.LC.Tree/Shrub/HrbCrp, NLDI, CRU.WET.orig, Liu.VOD.annual, GIMMS.FAPAR.pre, CRU.DTR.orig) in order to compare the performance of the two model approaches

based on the same predictor variables.

### 3.5 Process-oriented fire modelling with JSBACH-SPITFIRE

    We simulated burned area with the SPITFIRE (spread and intensity of fire) fire module within the JSBACH (Jena Scheme for Biosphere-Atmosphere Coupling in Hamburg) land surface model in order to compare the performance of SOFIA models to a state-of-the art global vegetation/fire model. This comparison potentially allows us to provide suggestions for the further

development of global vegetation/fire models.

    JSBACH is the land component of the MPI (Max Planck Institute for Meteorology) Earth system model (Raddatz et al., 2007). SPITFIRE is a physically based fire module that simulates fire ignitions (based on lightning and population density), fire spread, and fire effects depending on weather conditions, vegetation type and structure, fuel moisture, and fuel size (Thonicke et al., 2010). SPITFIRE was originally developed for the LPJ (Lund-Potsdam-Jena) dynamic global vegetation model

(Thonicke et al., 2010). For the implementation of SPITFIRE in JSBACH, two parameters in SPITFIRE were adjusted, one




related to human ignitions and the other related to the drying of fuels (Lasslop et al., 2014). Additionally, the relation between wind speed and the rate of fire spread was modified (Lasslop et al., 2015) and a decrease of fire duration with increasing population density was implemented (Hantson et al., 2015a).

JSBACH was applied on a spatial resolution of 1.875°*1.875°. JSBACH runs on a half-hourly time step, while the SPITFIRE

module is called at daily time steps. A detailed description of the simulation setup is given in the FireMIP (fire model inter-comparison project) protocol from which we use the JSBACH baseline simulation SF1 (Rabin et al., 2016). Following a spinup period to equilibrate carbon pools (continued until the slow carbon pool varied less than 1% between consecutive 50-year periods), a transient simulation was started in 1700. Data on land use (Hurtt et al., 2011) and population density (Goldewijk et al., 2010) were used starting in 1700 and interpolated to annual resolution. The $CO_2$ concentration of the atmosphere was

provided starting from 1750 at annual resolution (Le Quéré et al., 2014). $CO_2$ concentration before 1750 was set to the value of 1750. Climate forcing is based on the CRUNCEPv5 dataset (1901-2013) (Wei et al., 2014). Climate data was recycled over the years 1901-1920 before 1901.

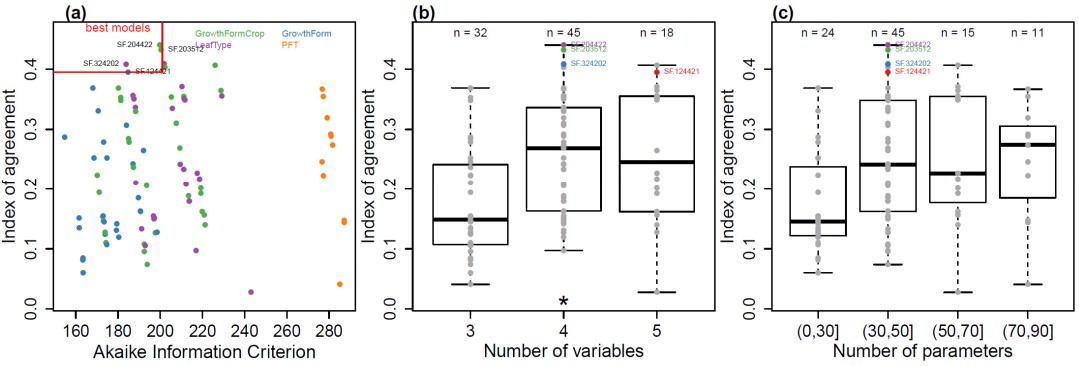

**Figure 2: Effects of model complexity on the performance of SOFIA models. Model performance is expressed as the index of agreement between simulated and observed (GFED) monthly burned area time series in the optimization data subset. (a) Scatterplot of the index of agreement against AIC classified by the used land cover grouping scheme. (b and c) Effect of the number of variables and parameters on model performance, respectively. The star symbol in (b) indicates a significantly higher IoA of models with 4 variables than the other groups (Wilcoxon rank sum test, p ≤ 0.05). The four best SOFIA models (IoA ≥ 0.395 and AIC ≤ 200) are**
**highlighted by a red box in (a) and by coloured points in (b) and (c).**

## 4 Results

### 4.1 Performance and complexity of SOFIA models

The optimized candidate SOFIA models covered wide ranges of complexities and performances (Figure 2, Table 2). The best SOFIA models explained reasonable the monthly spatial-temporal patterns of fractional burned area (i.e. up to IoA = 0.45 for

SF.230512, Table 2) but underestimated the observed variance (i.e. negative FV, best FV = -1.44 for SF.204422). Although the comparison of the GFED and Fire CCI burned area datasets showed only a moderate agreement (IoA = 0.85 and FV = 0.06), the performance of SOFIA models was similar for both datasets (Table 2). The performance of SOFIA models was very similar for the optimization and evaluation data subsets which shows that SOFIA models can be robustly applied to different spatial and temporal domains. The SOFIA model with the lowest AIC considered only three controlling factors and had 21

parameters (SF.124002, Table A 2). However, this model reached only a poor performance (IoA = 0.29 and FV = -1.68 in the optimization subset). Consequently, this model is not suited to simulate global fire activity. Therefore we selected the best SOFIA models according to both performance (IoA ≥ 0.4) and AIC (AIC ≤ 200) (Figure 2 a). The four best SOFIA models had different combinations of predictor variables which indicates the equifinality in predicting global burned area. However




the results show that SOFIA models were robust enough to predict global monthly fractional burned area for different spatial

and temporal domains and using different datasets.

**Table 2: Performance of the best SOFIA and of random forest models in predicting global distributed monthly burned area time series in the optimization and evaluation data subsets, respectively. Results for all SOFIA models are provided in Table A2. Please note that results for JSBACH-SPITFIRE are not included in this table because of its coarser spatial resolution.**

| Name | Model structure and included variables | GFED.BA as reference (1997-2011) | | | | | | CCI.BA as reference (2005-2011) | | | |
|---|---|---|---|---|---|---|---|---|---|---|---|
| | | Optimization subset (1817 cells, even years) (data used for optimization) | | | | Evaluation subset (1212 cells, uneven years) | | Optimization subset (even years) | | Evaluation subset (uneven years) | |
| | | SSE | AIC | IoA | FV | IoA | FV | IoA | FV | IoA | FV |
| GFED | Comparison of GFED.BA with CCI.BA | - | - | - | - | - | - | 0.78 | -0.19 | 0.85 | 0.06 |
| **Best SOFIA models** | | | | | | | | | | | |
| SF.204422 | GrowthFormCrop, CRU.WET.orig, Liu.VOD.annual, GIMMS.FAPAR.pre, CRU.T.annual | 51.88 | 199.8 | 0.44 | -1.44 | 0.39 | -1.55 | 0.42 | -1.53 | 0.41 | -1.53 |
| SF.203512 | GrowthFormCrop, GPCC.P.orig, GIMMS.FAPAR.annual, Liu.VOD.pre, CRU.T.annual | 52.17 | 200.3 | 0.43 | -1.45 | 0.42 | -1.54 | 0.45 | -1.49 | 0.45 | -1.51 |
| SF.324202 | LeafType, NLDI, CRU.WET.orig, GPCC.P.annual, CRU.T.annual | 52.92 | 183.8 | 0.41 | -1.49 | 0.37 | -1.65 | 0.39 | -1.59 | 0.35 | -1.65 |
| SF.124421 | GrowthForm, NLDI, CRU.WET.orig, Liu.VOD.annual, GIMMS.FAPAR.pre, CRU.DTR.orig | 53.40 | 184.8 | 0.40 | -1.51 | 0.39 | -1.51 | 0.39 | -1.59 | 0.41 | -1.51 |
| **Random forest models** | | | | | | | | | | | |
| RF1 | Random forest based on all variables as in Table 1 | 8.36 | - | 0.95 | -0.59 | 0.58 | -1.24 | 0.77 | -0.76 | 0.58 | -1.24 |
| RF2 | Like RF1 but without CCI.SM variables | 8.58 | - | 0.95 | -0.60 | 0.58 | -1.26 | 0.77 | -0.77 | 0.58 | -1.26 |
| RF.124421 | Random forest using the same variables as the SOFIA model SF.124421 | 24.05 | - | 0.81 | -1.23 | 0.41 | -1.69 | 0.65 | -1.35 | 0.40 | -1.70 |


We also tested if alternative cost functions in the optimization of SOFIA models would reduce the underestimation of the observed variance of burned area. The tested alternative cost functions explicitly accounted for variance, burned area anomalies, or were based on transformed burned area values (Table A 3). Although a cost function based on IoA and FV reached better performances in terms of IoA (best IoA = 0.45 against CCI.BA in the evaluation subset) and reproduced the

observed variance of burned area (FV = 0 against GFED in the training subset), the resulting model overestimated mean fractional burned area which is reflected by a high SSE (Table A 3). Other alternative cost functions resulted in weaker performances than the default SSE cost function. Consequently, we used the SSE-based cost function for the optimization of all SOFIA models.

The performance of SOFIA models varied with model complexity. SOFIA models that used a higher number of controlling

factors (n = 4 or 5) had in average a better performance than models with only three factors (Figure 2b). However, very complex SOFIA models with a high number of parameters (n = 70-90) did not necessarily result in higher performances than models with an average number of parameters (n = 30-70, Figure 2c). Models with a low number of parameters (n < 30) had in average low performances but we also found some SOFIA models with few parameters that reached good performances (e.g. SF.124021 with only 30 parameters, Table A 2). The four best SOFIA models had between 30 and 50 parameters. The number

of parameters in SOFIA models was mostly affected by the choice of a certain land cover grouping scheme to regionalize model parameters. Models that used the GrowthForm (3 groups), GrowthFormCrop or LeafType (both 4 groups) grouping schemes reached much lower AIC values than models that used the PFT grouping scheme (with 9 PFTs) (Figure 2a). These results demonstrate that SOFIA models with a higher number of predictor variables but a medium amount of model parameters reached the best performances in predicting global monthly spatial-temporal patterns of burned area.

Random forest models reached slightly better performances than the best performing SOFIA models. The random forest model based on all variables reached very good performance in training (IoA = 0.95 for RF1) and moderate performances in the evaluation subset (IoA = 0.58 for RF1, Table 2). Similar as for the SOFIA models, the employed random forests underestimated the observed variance. The random forest models with (RF1) and without soil moisture variables (RF2) reached similar performances. The random forest model RF.124421 reached slightly weaker performances (IoA = 0.4, FV= -1.7 in evaluation

against CCI burned area) than the corresponding SOFIA model SF.124421 with the same predictor variables. Thus the highly





flexible model structure of the random forest machine learning approach did not necessarily result in a much better performance than the best-performing SOFIA models. Consequently, the SOFIA modelling concept offers enough flexibility to assess different model structures to predict burned area.

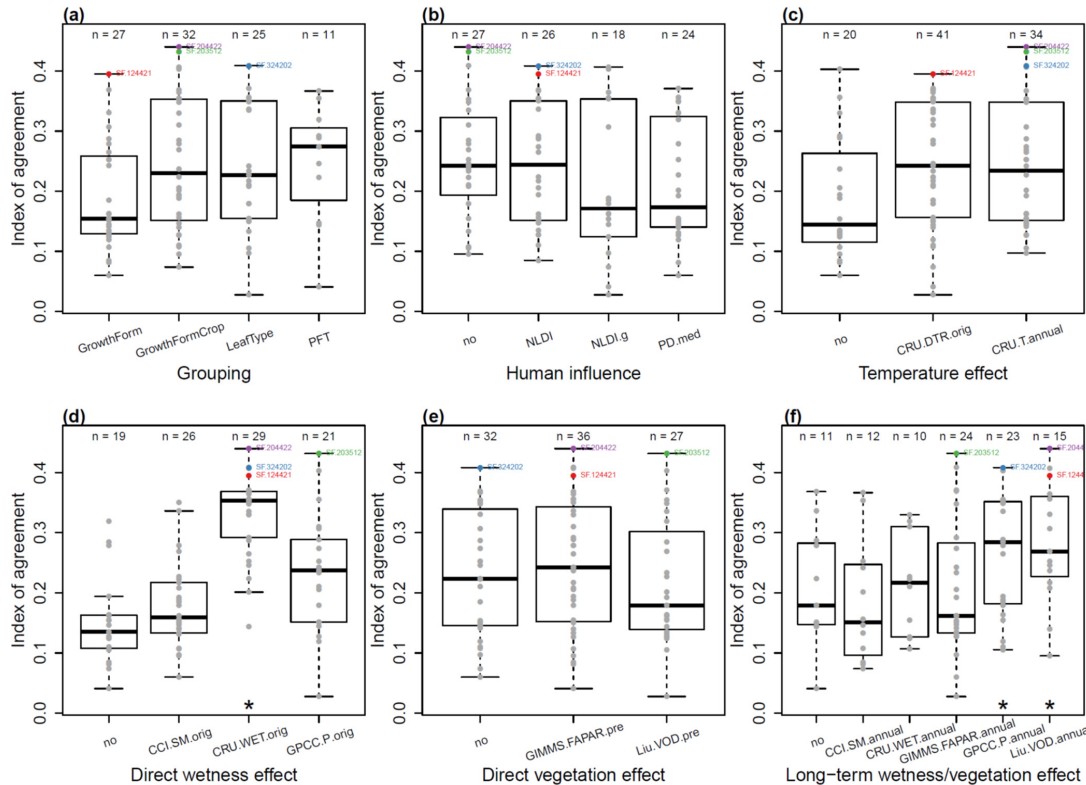

**Figure 3: Effect of structural components of SOFIA models on the performance in simulating global monthly burned area dynamics. Performance is expressed as the index of agreement between simulated and observed (GFED) monthly burned area for the training data subset. Boxplots show the distribution of IoA based on all SOFIA model experiments that include the respective variable. Star symbols indicate a significantly higher IoA of a variable in comparison to the "no" group of each controlling factor (Wilcoxon rank sum test, p ≤ 0.05). Distribution of IoA depending on the used (a) land cover grouping scheme; and variables to account for (b) human influence; (c) temperature effects; (d) direct wetness effects; (e) direct vegetation effects; and (f) long-term wetness or vegetation effects. The best models (IoA > 0.4 and AIC < 200) are highlighted with coloured dots.**

## 4.2 Required structural components of SOFIA models

The performance of SOFIA models depended on the predictor variables that were used as components in model structures (Figure 3). The choice of a certain land cover grouping scheme in SOFIA models to regionalize model parameters had only weak effects on model performance (Figure 3a). Models based on the GrowthForm scheme had on average weaker performances than models based on land cover grouping schemes that separately considered croplands. Models based on PFTs had on average higher performances than models with less land cover groups. However, the best SOFIA models were not related to a certain land cover grouping scheme, i.e. any land cover grouping scheme can potentially be used to regionalize model parameters. These results demonstrate that a PFT-based parameterization provides on average better results but some models based on growth forms can also reach best performances in predicting global burned area.

Including human influences in SOFIA model structures did not improve model performance (Figure 3b). The best models either did not consider human influences or considered human influences through NLDI as global controlling function.



However, NLDI did in average not contribute to higher performances. SOFIA models that used population density had on average weaker performance than SOFIA models that used NLDI or that did not consider human influences. The weaker performance of population density as component in SOFIA models could be caused by the general model structure in which potential burned area equals the total vegetated area: As highly populated areas are usually associated with low vegetation cover, potential burned area is low as well, and thus population density does not provide further information. Although two of

the best SOFIA models did not contain any variable for human influences (SF.204422, SF.203512), they however considered the fractional coverage of croplands in the used land cover grouping scheme. Consequently, these two models considered human influence on fire indirectly through the coverage of croplands. These results suggest that human influences on fire activity can be relatively interchangeably described by the coverage of croplands, NLDI, or population density, but the combination of these factors does not result in a further improvement in model performance. These results also suggest that

more appropriate information and metrics about human fire usage and management are required to significantly improve global fire models.

Considering temperature variables in SOFIA models caused on average better model performances than model structures without temperature variables (Figure 3c). However, we also found one model without a temperature control that reached good performance (SF.233210, Table A 2). All of the best performing models included diurnal temperature range or pre-fire annual

mean temperature as controlling factors. These results show that temperature-related variables are important predictors in SOFIA.

The consideration of direct wetness effects in SOFIA models had the largest positive impact on model performance (Figure 3d). Models that did not consider direct wetness effects had lower performances than models that used soil moisture, precipitation, or the number of wet days. Especially models based on the number of wet days reached significant higher IoA

than models without direct wetness effects (Wilcoxon rank sum test, $p \leq 0.05$). Consequently, direct wetness effects on fire activity were a required component of SOFIA models to predict burned area.

Whether or not including direct vegetation controls did not lead to a significant change in performance of the SOFIA models (Figure 3e). The best models either did not consider direct vegetation effects (SF.324202), used pre-fire FAPAR (SF.204422, SF.124421), or pre-fire VOD (SF.203512). This suggests that FAPAR and VOD conditions of the month before a fire did not

provide additional information to predict burned area in SOFIA models.

On the contrary, considering long-term wetness or vegetation effects in SOFIA models caused significantly higher model performances than not considering these effects (Figure 3f). Especially SOFIA models that used pre-fire annual precipitation or VOD reached significantly higher IoA. Models with long-term effects based on soil moisture, the number of wet days, or FAPAR had on average similar performances as models without long-term effects. However, we also found some good models

that used long-term conditions of FAPAR (e.g. SF.203512). These results demonstrate that long-term conditions in vegetation productivity (reflected by annual precipitation) or vegetation structure (reflected by VOD or FAPAR) were required components of SOFIA models to predict burned area.

Based on the performances of the different controlling factors and associated variables, the ideal SOFIA model should include NLDI as human influence, one variable to account for temperature effects, the number of wet days as direct wetness effect,

and pre-fire annual conditions of precipitation or VOD as long-term wetness/vegetation effects. This ideal model structure is realized in two of the best performing SOFIA models (SF.124421 and SF.324202, Figure 3). The choice of a certain land cover grouping scheme or of a direct vegetation effect are secondary components of SOFIA model structures.

### 4.3 Global evaluation of burned area from different modelling approaches

#### 4.3.1 Global spatial patterns

The best SOFIA models were applied globally to assess their performance in simulating global and regional spatial-temporal patterns of annual total burned area with respect to random forest models and JSBACH-SPITFIRE. All three model approaches




reproduced well the global spatial pattern of mean annual burned area with large amounts of burned area in Africa, Australia, and Tropical South America, and smaller amounts of burned area in the rest of the world ($0.663 \geq$ IoA $\leq 0.841$, Figure 4). However models were often biased in comparison to the observational datasets. The global mean annual burned area was 341

Mha for the GFED dataset, 346 Mha for the CCI dataset, and is estimated much higher (464 Mha) based on assumptions about undetected small fires (Randerson et al., 2012). Although JSBACH-SPITFIRE overestimated global burned area (~32%) in comparison to the GFED and CCI datasets it was however tuned (by adjusting ignitions) to reproduce the burned area estimates including small fires. Results from the SOFIA and random forest models cannot be directly compared to these global burned values because they have gaps both in space and time depending on the missing values in the used predictor variables.

Therefore, we masked the GFED and CCI datasets with the spatial-temporal distribution of gaps in all SOFIA and random forest models and recomputed the global mean annual burned area (Figure 4). All SOFIA models underestimated global mean annual burned area (-24 to -40 %, Figure 4). The random forest model RF2 overestimated (~60 %) and the random forest model RF.124421 reached a realistic (3-5% overestimation) global mean annual burned area. Despite the fact that all models reproduced well the global spatial pattern of annual burned area, the maps indicate regional differences especially in extra-

tropical regions.

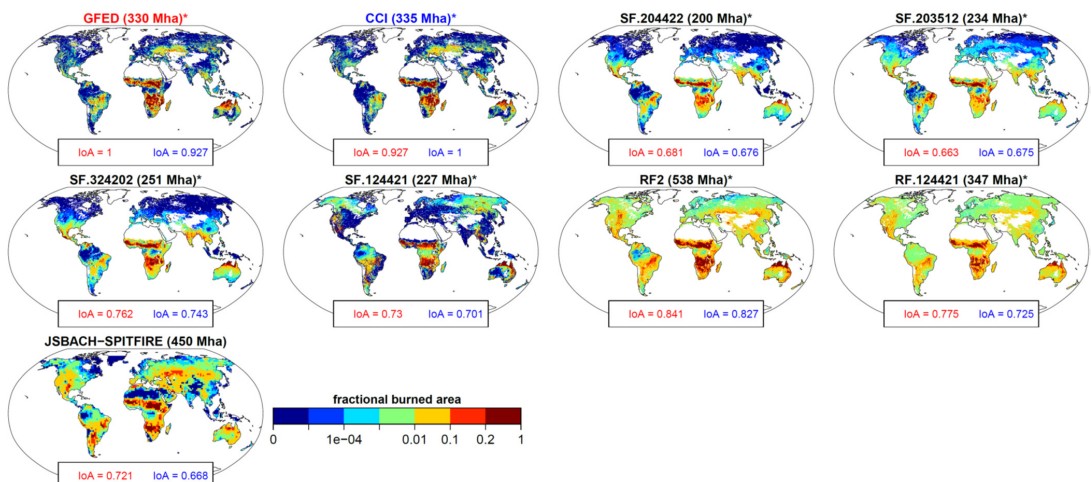

Figure 4: Mean annual fractional burned area in 2005-2011 from observational datasets and global fire models. Numbers in brackets
are the global mean annual burned area. In case of * symbol, the computation of global total annual burned area considered the common spatial-temporal occurrence of missing values in all SOFIA and random forest models on the 0.25° grid cells. IoA is shown with respect to GFED (red) and CCI (blue), respectively. All maps were aggregated to the coarsest common spatial resolution (i.e. JSBACH, ~1.875*1.875°) for the computation of IoA and total burned area.

### 4.3.2 Variability in tundra and boreal forests

Regionally, we found varying performances of SOFIA models, random forest, and JSBACH-SPITFIRE in simulating spatial-temporal and statistical distributions of annual total burned area (Figure 5). In northern regions (boreal forests and tundra), differences between all datasets and models were large: Whereas three SOFIA models produced almost no fire activity and thus had very poor performances, the model SF.124421 reached medium performances (IoA = 0.48 vs. CCI in boreal needleleaf

deciduous forests, Figure 5 c). The main difference between these SOFIA models is that SF.124421 used diurnal temperature range and the other three SOFIA models used annual pre-fire temperature as temperature effects on fire activity. Thus the




results suggest that mean annual temperature is not an appropriate predictor variable to represent boreal fire activity within a global fire model. Random forest models strongly overestimated mean annual burned area in northern regions.

In the tundra, all models had very low performances but SF.124421 reproduced at least the mean annual burned area from the

GFED dataset. However, also the GFED and CCI datasets strongly disagree in the tundra (IoA = 0.17 and FV = -1.91 for CCI vs. GFED, Figure 5 a) while only moderately agreeing in boreal forests. We found that SOFIA and random forest models agreed slightly better with the CCI dataset than with the GFED dataset in northern regions although the GFED dataset was used for training. In boreal needle-leaved evergreen forests, SF.124421 reproduced mean annual burned area and reached the highest IoA of all models (Figure 5 b).

In boreal needle-leaved deciduous forests, the random forest models reached the highest performance (IoA = 0.52 for RF2 against CCI) but overestimated mean annual burned area. SF.124421 and JSBACH-SPITFIRE only slightly overestimated mean annual burned and reached medium performances (IoA = 0.47 for SF.124421 vs. CCI, IoA = 0.31 for JSBACH-SPITFIRE vs. CCI) (Figure 5 c). In summary, although SF.124421 had only moderate performances in northern regions, it reached slightly better performances than random forest models and JSBACH-SPITFIRE. However, these results demonstrate

the need to further investigate fire activity in tundra and boreal forests by improving the agreement of satellite datasets and by developing more appropriate empirical and process-oriented fire models.





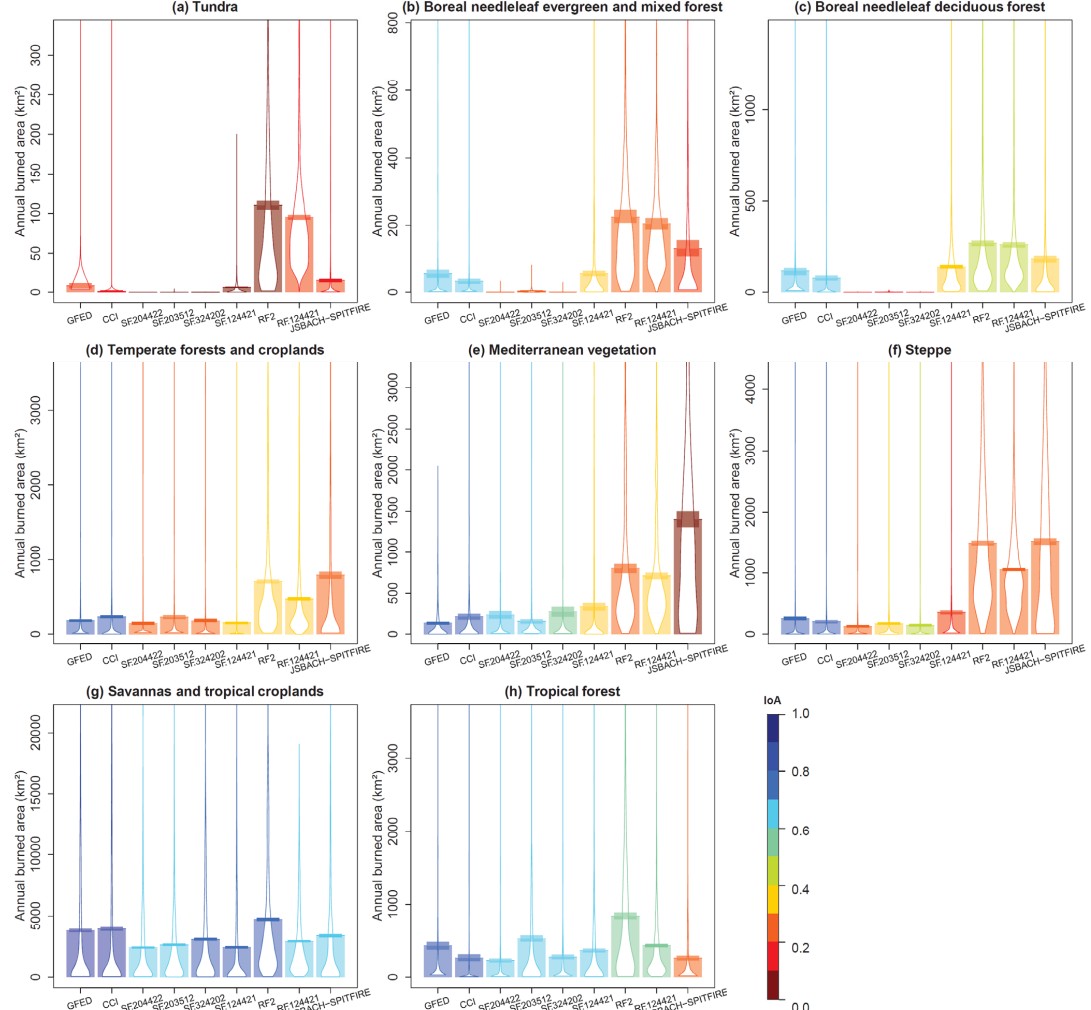

**Figure 5: Regional distributions of annual total burned area per 1.875° grid cells from datasets and global fire models for the years 2005-2011. Bars show the mean of the distribution. Horizontal bands at the top of each bar are error estimates for the mean value (i.e. 95% highest density intervals). Violins show the distribution of values. Colours represent the index of agreement between a model and both (i.e. GFED and CCI) datasets. For GFED and CCI, the index of agreement was computed only with respect to the other observational burned area dataset. The extent of regions is shown in Figure A1 a.**

### 4.3.3 Variability in temperate regions and the Mediterranean

In temperate regions, SOFIA models generally outperformed random forest models and JSBACH-SPITFIRE in reproducing the observed spatial-temporal and statistical distributions of annual total burned area (Figure 5d-f). The random forest models and JSBACH-SPITFIRE overestimated mean annual burned area in all temperate regions.

In temperate forests and croplands, SF.124421 reached the best performance of all models (IoA = 0.43 and FV = -0.2 vs. GFED), whereas the other three SOFIA models had weaker performances (Figure 5 d). Random forest models reached medium IoA (up to 0.4 for RF.124421 vs. GFED) but overestimated mean annual burned area. JSBACH-SPITFIRE had medium IoA and overestimated mean annual burned area in comparison to GFED and CCI.

In the Mediterranean, all SOFIA models had medium to good performances (0.28 ≤ IoA ≤ 0.75) and outperformed JSBACH-SPITFIRE and random forest models (Figure 5 e). The performance was usually higher in comparison to the CCI dataset than





in comparison to the GFED dataset because GFED contained much fewer very large burned areas and thus had also on average a smaller burned area than the CCI dataset. The models SF.204422 and SF.203512 (both using the GrowthFormCrop scheme and no human influence) had better performances than the models SF.324202 and SF.124421 (both using NLDI). This indicates that the better performance is related to how croplands and human influences are represented in these models.

In the steppes, all SOFIA models reproduced the observed mean annual burned area and some reached medium performances
(IoA = 0.48 for SF.324202 vs. CCI, Figure 5 f). These results for temperate regions and the Mediterranean demonstrate that SOFIA models can realistically reproduce observed fire activity.

### 4.3.4 Variability in tropical regions

In tropical regions, SOFIA models had good performances in reproducing the observed spatial-temporal and statistical distributions of annual total burned area and had comparable or better performances than the random forest models and
JSBACH-SPITFIRE (Figure 5g-h). In savannahs and tropical croplands, all SOFIA and random forest models and JSBACH-SPITFIRE had good performances in reproducing the spatial-temporal distribution of annual total burned area ($0.63 \leq$ IoA $\leq$ 0.78) but underestimated the variance and extreme fire years ($-1.2 \leq$ FV $\leq -0.4$). This underestimation of very large burned areas in savannahs is the main cause for the underestimation of the mean annual burned area in this region and of the global total burned area by SOFIA models. SF.324202 and SF.124421 had slighter better performances than the other two SOFIA
models.

In tropical forests, all SOFIA models had medium to good performances in reproducing the spatial-temporal distribution of annual total burned area ($0.61 \leq$ IoA $\leq 0.68$) but also underestimated the variance and extreme fire years ($-1.16 \leq$ FV $\leq -0.36$, Figure 5 h). However, the FV of all models was usually better in comparison to the CCI dataset than for the GFED dataset. The CCI dataset had less very large burned areas and thus a smaller variance than the GFED dataset in tropical forests (FV =
-0.22 for CCI vs. GFED). Random forest models reached moderate but weaker performances than SOFIA models. JSBACH-SPITFIRE had a low performance in reproducing the spatial-temporal variability (IoA = 0.33 vs. GFED) but reproduced mean annual burned area. These results demonstrate that SOFIA models better reproduce observed fire activity in tropical regions than random forests or JSBACH-SPITFIRE.

In summary, we found that all modelling approaches (SOFIA, random forest, JSBACH-SPITFIRE) had relatively good
performances in savannahs and tropical croplands. All SOFIA models had relatively good performances in tropical forests and the Mediterranean. Only some SOFIA models reached good performances in temperate forests and croplands (SF.124421) and in steppes (SF.324202). Random forest models and JSBACH-SPITFIRE had generally weaker performances than SOFIA models. The model SF.124421 (Figure 1) had the best performance from all SOFIA models in the tundra, boreal forests, temperate forests and croplands; it had very good performance in savannahs and tropical forests; and it outperformed random
forest and JSBACH-SPITFIRE in steppes and the Mediterranean. Consequently, we finally identified SF.124421 as the globally best performing SOFIA model and provide in the following chapter a demonstration how to apply this model to identify controls on fire activity.

### 4.4 Application of SOFIA to identify controls on fire activity

The underlying controlling functions of SOFIA models allow to investigate the role of human, vegetation, and climate variables
on spatial patterns of fire activity. To demonstrate such a potential application of a SOFIA model, we mapped mean values of each controlling function for the period 1997-2011 from the SOFIA model SF.124421 (Figure 6). Based on this model, human influences (i.e. NLDI) suppressed fire activity in most parts of Europe and southern Russia, east and south-east Asia, India, central and eastern North America, south-east South America, south Australia and New Zealand (Figure 6 a). These regions correspond to the most populated and developed regions of the world. This pattern was caused by the underlying controlling
function of SF.124421 where NLDI < 1 (i.e. developed regions) suppressed and NLDI > 1 (i.e. unpopulated regions or natural




ecosystems) allowed fire activity (Figure 1 b). These results indicate a predominant suppressing effect of humans on fire activity.

Temperature effects, expressed as diurnal temperature range, allowed fire activity mostly in the semi-deserts of western North America, in the Sahel, Australia, and had a moderate suppression effect in tropical forests and the tundra (Figure 6 b). These

spatial patterns were caused by the controlling function that had a strong sigmoidal increase of fire activity with diurnal temperature range in shrublands and allowed moderate fire activity in herbaceous vegetation and croplands (Figure 1 c).

Direct wetness effects, expressed as the number of wet days, generally allowed fire activity in all forest regions and moderately suppressed fire activity in the rest of the world (Figure 6 c). The underlying controlling function showed no sensitivity for forests, a weak positive relation in herbaceous vegetation and croplands, and a strong exponential decrease of fire activity with

increasing number of wet days in shrublands (Figure 1 d).

As direct vegetation effect, pre-fire FAPAR suppressed fire activity in herbaceous vegetation and croplands of central North America, central Asia, in the northern Sahel, the Kalahari, central Australia, and in parts of South America (Figure 6 d). On the other hand, pre-fire FAPAR supported fire activity mostly in the southern Sahel and northern and eastern Australia. These patterns were caused by a general strong suppression of fire activity with pre-fire FAPAR in herbaceous vegetation and

croplands and an exponential increase of fire activity with increasing pre-fire FAPAR in shrublands (Figure 1 e).

As long-term vegetation effect, mean vegetation optical depth in the 12 months before fire strongly supported fire activity in central North America, central Asia, the Tibetan plateau, the Sahel, parts of India, the Kalahari, in Australia (except interior), and in northern Patagonia (Figure 6 e). In all other regions, annual VOD had a moderate effect on fire activity. The underlying controlling function showed an exponential increase of fire activity with annual VOD in shrublands, an exponential decrease

with annual VOD in herbaceous vegetation and croplands and a strong suppression across all VOD ranges for trees (Figure 1 f). The diverging responses with annual VOD in shrublands and herbaceous vegetation indicate that fire activity increases with higher vegetation density or biomass in shrublands but decreases with increasing vegetation water content in herbaceous vegetation, respectively. Additionally, the general suppression of fire activity with VOD for trees indicates that fire activity is suppressed by vegetation density or high vegetation water content in forests.

We further combined the controlling functions of SF.124421 to investigate combined controls on fire activity. Therefore we created a red-green-blue composite map in which the red channel contains the NLDI control function, the green channel contains the mean of the direct (pre-fire FAPAR) and long-term vegetation (pre-fire annual VOD) effect, and the blue channel contains the climate effects (mean of control functions on number of wet days and diurnal temperature range) (Figure 6f). Generally, bright colours in this map indicate a strong suppression of fire activity with small burned areas and dark colours

indicate that fire activity is allowed, resulting in large burned areas. Regionally, different combinations of socioeconomic, vegetation and climate factors controlled fire activity. Socioeconomic development dominantly suppressed fire activity in western North America, and in populated regions of boreal forests (red colours). Vegetation predominantly supressed fire activity in southern boreal and tropical forests (green colours). Primarily climate conditions and secondly socioeconomic development suppressed fire activity in semi-deserts of the northern Sahel, central Asia, the Kalahari, and south-western

Australia (purple colours). Socioeconomic development and climate equally supressed fire activity in the Mediterranean, India, eastern Asia, and east South America (pink colour). Both socioeconomic development and vegetation conditions supressed fire activity in most parts of Europe, central and eastern North America, and eastern China (yellow/orange colours). Both climate and vegetation conditions supressed fire activity in the tundra and in central Australia (cyan colours). All factors moderately supported fire activity in boreal forests and strongly support fire activity in large parts of the Sahel, southern Africa,

northern Australia, and western North America (dark colours). In summary, fire activity is controlled by regionally diverse and complex interactions of human, vegetation and climate factors.





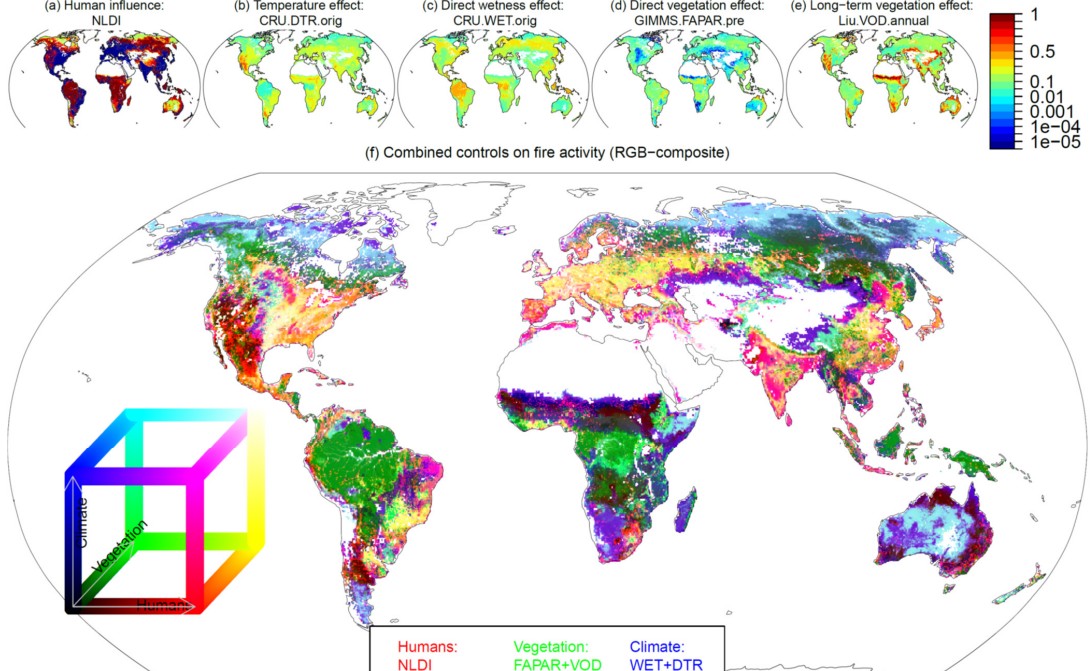

**Figure 6: Combined climate, vegetation, and human controls on fire activity based on the SOFIA model SF.124421. The maps in (a-e) show the average value for each controlling function for the period 1997-2011. High values (1, red) indicate that this factor allows unlimited burning and low values (0, blue) indicate that this factor suppresses burning. The map in (f) is a red-green-blue composite of the human influence (map in (a), red channel), the combined direct and long-term vegetation effect (mean of (d) and (e), green channel), and the climate effect (mean of (b) and (c), blue channel). Bright and dark colours indicate a strong suppression and allowance of fire activity, respectively.**

## 5 Discussion and conclusions

### 5.1 Controls on fire activity and required model components

We developed the SOFIA modelling concept in order to identify required model structures to predict spatial-temporal patterns of burned area. The best SOFIA models reached globally average performances but outperformed the state-of-the art process-oriented vegetation/fire model JSBACH-SPITFIRE. We interpret the globally medium and regionally varying performances as current upper limits that can be reached with the used predictor datasets and variables because the more flexible and highly adaptive machine learning algorithm random forest did not achieve much higher performance in the evaluation data subset. These upper limits in model performance might be due to several reasons:

1. Uncertainties in the observations for the predictor and response variables inhibit the development of models with high performance. For example, we found regionally partly large differences between the two burned area datasets, especially in northern regions. These uncertainties originate from differences in sensor characteristics and in the ability of the used algorithms to detect small fires.

2. Other processes and variables are important for the spread of fires but cannot be resolved at the used spatial and temporal resolution. For example, on local to regional scales the spread of fire is controlled by landscape structure and topography whereas climatic controls are usually more important on larger scales (Archibald et al., 2009; Liu et al., 2013b; Parisien et al., 2010). Most of the regional controls can likely not be resolved at the used spatial resolution (0.25°) although this resolution is already higher than the resolution of most global vegetation/fire models. Also wind speed and direction is an important control on the spread of fires on short temporal scales but this effect cannot accurately represented based on monthly data (Bistinas et al., 2014).



3. There is a lack of global observations that directly represent fuel loads, fuel moisture, or modes of human fire usage. For example, all of the used predictor variables are only proxies for fuel loads (FAPAR or VOD) or fuel moisture (surface soil moisture) but do not directly represent fuel such conditions. Similarly, data on population density or socioeconomic development are used as proxies for human effects on fire but cannot represent the complex social, economic, and cultural practices, and policies of human fire use and management.

However, we demonstrated from the ensemble of all candidate SOFIA models that model components on human influences, direct wetness effects, and long-term vegetation effects on fire activity are most important and all need to be accounted for to improve global vegetation/fire models.

Globally, burned area decreases with increasing socioeconomic development and is mainly supressed in croplands and developed regions of the world (Figure 6). Hence our results confirm previous studies that also assign a predominantly suppressing effect of humans in African croplands (Andela and van der Werf, 2014) and a global decrease of burned area with increasing population density (Archibald et al., 2013; Bistinas et al., 2014; Chuvieco and Justice, 2010; Knorr et al., 2014). On the other hand, datasets on population density or socioeconomic development did not improve the performances of the SOFIA models if croplands were already included in the model structure. Consequently, a more detailed analysis and modelling of human-fire interactions is necessary but is currently hampered by the availability of globally available, temporally resolved accurate information on human fire usage and management.

Direct wetness effects, especially based on the number of wet days, were the component of SOFIA models that contributed most to model performance (Figure 3). These result are in agreement with previous results that identified the number of dry days (the inverse of the number of wet days) as an important variable to predict fire activity (Bistinas et al., 2014). Especially for shrublands, we identified strong exponential relations with the number of wet days and diurnal temperature range. Currently, shrubs are not considered in all ecosystem models (e.g. not in models of the LPJ family, Sitch et al. (2003)) which suggest the need to implement and parameterize shrub PFTs to improve simulations of fire activity. The number of wet days and diurnal temperature range are also used in process-oriented fire models like SPITFIRE to compute the Nesterov index (a fire weather index) and fuel moisture content (Thonicke et al., 2010). Here we confirm that the use of diurnal temperature range and the number of wet days are appropriate predictor variables to simulate fuel moisture conditions and thus fire activity. However, while the Nesterov index is used as fire weather index in many fire modules of global vegetation models (Lasslop et al., 2014; Prentice et al., 2011; Thonicke et al., 2010; Venevsky et al., 2002; Yue et al., 2014), studies on forest fire management rely more often on alternative fire weather indices such as from the Canadian Forest Fire Weather Index (FWI) (Bedia et al., 2012; Stocks et al., 1989). We also show that direct wetness effects can be represented by satellite-derived surface soil moisture. Additionally, several other indices have been derived from satellite data to estimate fuel moisture conditions (Yebra et al., 2013). Consequently, it is necessary to systematically compare the predictive power of fire weather indices, satellite-derived and reanalysis-based surface soil moisture data , and soil moisture schemes of ecosystem models to potentially improve the direct effect of wet conditions on fire activity in global vegetation/fire models.

Long-term vegetation effects contributed strongly to the performance of SOFIA models and thus indicate an important role of vegetation dynamics on the spatial-temporal variability of fire activity. Consequently, global vegetation models require a good representation of vegetation distribution and dynamics to realistically simulate fire activity. Vegetation distribution can be improved either through the prescription of high quality land cover maps in land surface models, or by improving model structures and by constraining model parameters that affect vegetation dynamics in DGVMs. For both approaches, time-variant, e.g. annually resolved, land cover maps would be very valuable to realistically reflect vegetation dynamics. However, it is currently unclear how realistic land cover dynamics are represented for example by the three epochs of the ESA CCI land cover maps or by annual maps of the MODIS land cover product. Hence intensified efforts are required to check the plausibility of land cover changes in current and upcoming time-variant land cover maps.





SOFIA models with a long-term effect of VOD had better performances than models without this effect. The good performance of SOFIA models with VOD as predictor variable likely reflects variability in fuel loads because VOD is sensitive to vegetation density and biomass (Andela et al., 2013; Liu et al., 2015). The importance of VOD suggests that processes such as carbon allocation, turnover and vegetation mortality that control the dynamics of biomass need to be carefully assessed in global

vegetation models in order to accurately simulate fuel loads and hence fire activity. The finding of a strong suppression of fire activity with VOD in forests corresponds to previous findings that show that woody vegetation tends to suppress burned area either because moist wood is more difficult to ignite than dry grass or litter, or because forests provide generally more moist conditions (Kelley and Harrison, 2014). Fire activity increases with biomass at low vegetation densities and strongly decreases with increasing biomass and very high vegetation densities but the actual fire activity is enhanced or suppressed by moisture

conditions (Krawchuk and Moritz, 2011; Murphy et al., 2011). Consequently, the SOFIA concept and the identified sensitivities of fire activity with direct wetness effects and with VOD confirm and implement previous conceptual models where fire activity follows a biomass gradient and is modulated by moisture conditions (Krawchuk and Moritz, 2011; Murphy et al., 2011).

**5.2 From satellite data to improved global vegetation/fire models**

The better performance of SOFIA models compared to JSBACH-SPITFIRE and the generally good performance especially in temperate and tropical regions demonstrates the potential of SOFIA models to improve global vegetation/fire models. The SOFIA modelling concept can be potentially applied to more complex global vegetation/fire models such as SPITFIRE in order to identify appropriate model structures. Thereby the SOFIA controlling functions should rely on forcing datasets (e.g. temperature, precipitation) and simulated state variables (e.g. litter and soil moisture, biomass compartments, litter stocks,

vegetation structure) of the vegetation models. This allows also to represent feedbacks of changing vegetation conditions on fire activity. By applying the SOFIA concept to forcing and state variables of a process-oriented vegetation model, a best model structure could be identified in order to better reproduce observed spatial-temporal dynamics of burned area.

In order to represent realistic vegetation/fire interactions, vegetation models need to satisfactorily reproduce observed patterns and dynamics of fuel moisture and vegetation state variables. Consequently, it is necessary to test and improve global

vegetation/fire models against multiple observational datasets that cover various aspects of vegetation/fire interactions: For example, satellite datasets on land cover, FAPAR, VOD, and biomass (Avitabile et al., 2016; Pettinari and Chuvieco, 2016; Saatchi et al., 2011; Thurner et al., 2014) may be useful to constrain vegetation dynamics, biomass allocation, and fuel loads; datasets on surface soil moisture, VOD, and evapotranspiration (Tramontana et al., 2016) may be useful to test hydrological schemes and to constrain fuel moisture; and datasets on burned area, fire size (Hantson et al., 2015b), fire radiative power, or

fuel consumption (Andela et al., 2016; van Leeuwen et al., 2014) might be useful to constrain fire behaviour. Such datasets are currently under-exploited in the development of global vegetation/fire models because #1 they were still missing at the time of model development (Thonicke et al., 2001), #2 there is only little experience of applying formal model-data integration approaches within global fire modelling, or #3 no appropriate model components or observation operators exist that link for example modelled fuel moisture with satellite-derived surface soil moisture or modelled biomass compartments with VOD.

For example, it is currently unclear which physiological processes, morphological plant components, and ecosystem structures contribute to a certain VOD signal (Vreugdenhil et al., 2016a). Consequently, it is necessary to better understand the plant and ecosystem controls on VOD to improve global vegetation/fire models.

Previously developed global fire models commonly used observed data for model evaluation but did not undertake a formal model-data integration cycle from the definition of model structures, model parameter estimation, to model evaluation, and

potentially back to a re-formulation of model structures by using observational data. In our study we firstly applied the full model-data integration cycle to derive an optimal structure for an empirical global fire model to predict global burned area. However, in order to apply model-data integration for global process-oriented vegetation/fire models, multiple datasets on



vegetation, hydrological, and fire-related variables should be used to realistically constrain vegetation/fire interactions. Hence there is a need to develop appropriate observation operators and to extent currently existing model-data integration frameworks

of global vegetation models (Forkel et al., 2014; Kaminski et al., 2013; MacBean et al., 2016; Schürmann et al., 2016) to the corresponding fire modules in order to formally assess model structures and to constrain model parameters. In summary, model-data integration frameworks need to be developed that make use of multiple satellite datasets on vegetation and moisture proxies in order to improve the representation of fire in global vegetation models and thus to better understand interactions of fire with ecosystems and the atmosphere within the Earth system.

**Code availability**

The code for this study is organized in several R packages and is available from https://r-forge.r-project.org/R/?group_id=1612. Thereby the package *SOfireA* contains the basic SOFIA model structure and functions to optimize and plot SOFIA models, and the package *ModelDataComp* contains functions for model-data comparison such as model evaluation metrics and comparison plots. The R package *randomForest* was used for random forest fits (Liaw and Wiener, 2002).

**Data availability**

The used original data is available under the URLs, DOIs, or can be obtained from PIs as indicated in Table 1. The pre-processed (spatially and temporally interpolated) data for the optimization and evaluation data subsets is included as example dataset '*firedata*' in the *SOfireA* R package (https://r-forge.r-project.org/R/?group_id=1612).

**Appendix**


Table A 1: Land cover to plant functional type conversion table. The units are % coverage of each PFT per land cover class. The conversion factors are based on Poulter et al. (2015a) with some modifications that affect boreal and arctic regions, i.e. to avoid coverage of broadleaved evergreen PFTs in these regions and to reach a total tree cover that is comparable to the MODIS tree cover product (Hansen et al., 2003).

| ID | Land cover class | Plant functional types | | | | | | | | | | |
|----|------------------|---------|---------|---------|---------|----------|----------|----------|------|------|------|--------|
| | | Tree.BE | Tree.BD | Tree.NE | Tree.ND | Shrub.BE | Shrub.BD | Shrub.NE | Herb | Crop | Bare | NoLand |
| 0 | No data | | | | | | | | | | | 100 |
| 10 | Cropland, rainfed | | | | | | | | | 100 | | |
| 11 | Cropland, rainfed, Herbaceous cover | | | | | | | | | 100 | | |
| 12 | Cropland, rainfed, Tree or shrub cover | | | | | | 50 | | | 50 | | |
| 20 | Cropland, irrigated or post-flooding | | | | | | | | | 100 | | |
| 30 | Mosaic cropland (>50%) / natural vegetation (tree, shrub, herbaceous cover) (<50%) | 5 | 5 | | | 5 | 5 | 5 | 15 | 60 | | |
| 40 | Mosaic natural vegetation (tree, shrub, herbaceous cover) (>50%) / cropland (<50%) | 5 | 5 | | | 5 | 10 | 5 | 30 | 40 | | |
| 50 | Tree cover, broadleaved, evergreen, closed to open (>15%) | 90 | | | | 5 | 5 | | | | | |
| 60 | Tree cover, broadleaved, deciduous, closed to open (>15%) | | 70 | | | | 10 | | 20 | | | |
| 61 | Tree cover, broadleaved, deciduous, closed (>40%) | | 80 | | | | 10 | | 10 | | | |
| 62 | Tree cover, broadleaved, deciduous, open (15-40%) | | 30 | | | | 20 | | 40 | | 10 | |
| 70 | Tree cover, needleleaved, evergreen, closed to open (>15%) | | | 70 | 0 | | 5 | 5 | 20 | | | |
| 71 | Tree cover, needleleaved, evergreen, closed (>40%) | | | 75 | 0 | | 5 | 5 | 15 | | | |
| 72 | Tree cover, needleleaved, evergreen, open (15-40%) | | | 30 | | | 5 | 5 | 30 | | 30 | |
| 80 | Tree cover, needleleaved, deciduous, closed to open (>15%) | | | | 50 | 0 | 15 | 5 | 25 | | 5 | |
| 81 | Tree cover, needleleaved, deciduous, closed (>40%) | | | | 70 | 0 | 10 | 5 | 15 | | | |
| 82 | Tree cover, needleleaved, deciduous, open (15-40%) | | | | 30 | | 10 | 5 | 35 | | 20 | |
| 90 | Tree cover, mixed leaf type (broadleaved and needleleaved) | | 35 | 25 | 0 | 0 | 10 | 10 | 15 | | 5 | |
| 100 | Mosaic tree and shrub (>50%) / herbaceous cover (<50%) | 0 | 15 | 15 | 0 | 0 | 15 | 15 | 40 | | | |



| | | | | | | | | | | |
|---|---|---|---|---|---|---|---|---|---|---|
| 110 | Mosaic herbaceous cover (>50%) / tree and shrub (<50%) | 0 | 7.5 | 7.5 | | 0 | 10 | 10 | 60 | 5 | |
| 120 | Shrubland | | | | | 20 | 30 | 10 | 20 | 20 | |
| 121 | Shrubland, evergreen | | | | | 30 | | 30 | 20 | 20 | |
| 122 | Shrubland, deciduous | | | | | | 60 | | 20 | 20 | |
| 130 | Grassland | | | | | | | | 70 | 30 | |
| 140 | Lichens and mosses | | | | | | | | 60 | 40 | |
| 150 | Sparse vegetation (tree, shrub, herbaceous cover) (<15%) | 0 | 2 | 2 | | 0 | 4 | 2 | 5 | 85 | |
| 152 | Sparse shrub (<15%) | | | | | 0 | 6 | 4 | 5 | 85 | |
| 153 | Sparse herbaceous cover (<15%) | | | | | | | | 15 | 85 | |
| 160 | Tree cover, flooded, fresh or brakish water | 30 | 30 | | | | | | 20 | | 20 |
| 170 | Tree cover, flooded, saline water | 60 | | | | 20 | | | | | 20 |
| 180 | Shrub or herbaceous cover, flooded, fresh/saline/brakish water | | 5 | 5 | | | 10 | 10 | 40 | | 30 |
| 190 | Urban areas | | 2.5 | 2.5 | | | | | 15 | 75 | 5 |
| 200 | Bare areas | | | | | | | | | 100 | |
| 201 | Bare areas, consolidated | | | | | | | | | 100 | |
| 202 | Bare areas, unconsolidated | | | | | | | | | 100 | |
| 210 | Water bodies | | | | | | | | | | 100 |
| 220 | Permanent snow and ice | | | | | | | | | | 100 |


**Table A 2: Structure and performance of all tested candidate SOFIA models. N denotes the number of model parameters. SSE, AIC, IoA and FV are based on monthly burned area time series in the optimization and evaluation data subsets from the GFED and CCI datasets, respectively. Model experiments are ordered by SSE. The best SOFIA models (IoA ≥ 0.4 and AIC ≤ 200.5) are indicated in red.**

| Structure of SOFIA models: used control factors and associated variables | |
|---|---|
| Grouping scheme (groups)<br>  1 GrowthForm<br>  2 GrowthFormCrop<br>  3 LeafType<br>  4 PFT | Long-term wetness/productivity effect (wetveg.longterm)<br>  0 no<br>  1 CCI.SM.orig.filter13<br>  2 GPCC.P.orig.filter13<br>  3 CRU.WET.orig.filter13<br>  4 Liu.VOD.orig.filter13<br>  5 GIMMS.FAPAR.orig.filter13 |
| Human influence (human)<br>  0 no<br>  1 PD.med (global)<br>  2 NLDI (global)<br>  3 NLDI.g (per group) | Direct vegetation effect (veg.dir)<br>  0 no<br>  1 Liu.VOD.orig.lagneg1<br>  2 GIMMS.FAPAR.orig.lagneg1 |
| Direct wetness effect (wet.dir)<br>  0 no<br>  1 CCI.SM.orig<br>  2 (unused)<br>  3 GPCC.P.orig<br>  4 CRU.WET.orig | Temperature effect (temp)<br>  0 no<br>  1 CRU.DTR.orig<br>  2 CRU.T.orig.filter13 |
| Example: SF.204422 = (2) GrowthFormCrop + (0) no human influence + (4) CRU.WET.orig + (4) Liu.VOD.orig.filter13 + (2) GIMMS.FAPAR.orig.lagneg1 + (2) CRU.T.orig.filter13 | |

| | Model structure and included variables | | | | | | | Comparison against GFED.BA (1997-2011) | | | | | | Comparison against CCI.BA (2005-2011) | | | |
|---|---|---|---|---|---|---|---|---|---|---|---|---|---|---|---|---|---|
| Name | N | groups | human | wet.dir | wetveg.longterm | veg.dir | temp | Training (1817 cells, even years) Data used for parameter optimization | | | | Evaluation (1212 cells, uneven years) | | Training (even years) | | Evaluation (uneven years) | |
| | | | | | | | | SSE | AIC | IoA | FV | IoA | FV | IoA | FV | IoA | FV |
| SF.204422 | 48 | 2 | 0 | 4 | 4 | 2 | 2 | 51.88 | 199.8 | 0.44 | -1.44 | 0.39 | -1.55 | 0.42 | -1.53 | 0.41 | -1.53 |
| SF.203512 | 48 | 2 | 0 | 3 | 5 | 1 | 2 | 52.17 | 200.3 | 0.43 | -1.45 | 0.42 | -1.54 | 0.45 | -1.49 | 0.45 | -1.51 |
| SF.304522 | 48 | 3 | 0 | 4 | 5 | 2 | 2 | 52.92 | 201.8 | 0.41 | -1.49 | 0.40 | -1.58 | 0.40 | -1.57 | 0.42 | -1.59 |
| SF.324202 | 39 | 3 | 2 | 4 | 2 | 0 | 2 | 52.92 | 183.8 | 0.41 | -1.49 | 0.37 | -1.65 | 0.39 | -1.59 | 0.35 | -1.65 |
| SF.234422 | 60 | 2 | 3 | 4 | 4 | 2 | 2 | 52.99 | 226.0 | 0.41 | -1.49 | 0.35 | -1.63 | 0.40 | -1.58 | 0.33 | -1.64 |
| SF.233210 | 48 | 2 | 3 | 3 | 2 | 1 | 0 | 53.10 | 202.2 | 0.40 | -1.50 | 0.34 | -1.69 | 0.41 | -1.57 | 0.32 | -1.68 |
| SF.124421 | 39 | 1 | 2 | 4 | 4 | 2 | 1 | 53.40 | 184.8 | 0.40 | -1.51 | 0.39 | -1.51 | 0.39 | -1.59 | 0.41 | -1.51 |
| SF.124021 | 30 | 1 | 2 | 4 | 0 | 2 | 1 | 54.05 | 168.1 | 0.37 | -1.56 | 0.36 | -1.56 | 0.37 | -1.64 | 0.37 | -1.57 |
| SF.204501 | 36 | 2 | 0 | 4 | 5 | 0 | 1 | 54.09 | 180.2 | 0.37 | -1.56 | 0.31 | -1.76 | 0.36 | -1.63 | 0.29 | -1.76 |
| SF.314511 | 51 | 3 | 1 | 4 | 5 | 1 | 1 | 54.14 | 210.3 | 0.37 | -1.55 | 0.34 | -1.63 | 0.35 | -1.63 | 0.34 | -1.62 |
| SF.424102 | 84 | 4 | 2 | 4 | 1 | 0 | 2 | 54.37 | 276.7 | 0.37 | -1.55 | 0.35 | -1.63 | 0.36 | -1.63 | 0.36 | -1.63 |
| SF.234421 | 60 | 2 | 3 | 4 | 4 | 2 | 1 | 54.40 | 228.8 | 0.36 | -1.56 | 0.33 | -1.63 | 0.36 | -1.64 | 0.34 | -1.63 |
| SF.314420 | 39 | 3 | 1 | 4 | 4 | 2 | 0 | 54.55 | 187.1 | 0.36 | -1.58 | 0.32 | -1.68 | 0.33 | -1.65 | 0.32 | -1.68 |
| SF.333221 | 60 | 3 | 3 | 3 | 2 | 2 | 1 | 54.57 | 229.1 | 0.35 | -1.57 | 0.32 | -1.71 | 0.35 | -1.64 | 0.30 | -1.69 |
| SF.224211 | 51 | 2 | 2 | 4 | 2 | 1 | 1 | 54.59 | 211.2 | 0.35 | -1.58 | 0.36 | -1.55 | 0.36 | -1.65 | 0.37 | -1.56 |
| SF.204202 | 36 | 2 | 0 | 4 | 2 | 0 | 2 | 54.62 | 181.2 | 0.35 | -1.58 | 0.35 | -1.52 | 0.35 | -1.67 | 0.36 | -1.54 |
| SF.424201 | 84 | 4 | 2 | 4 | 2 | 0 | 1 | 54.62 | 277.2 | 0.35 | -1.58 | 0.34 | -1.61 | 0.35 | -1.66 | 0.34 | -1.61 |
| SF.234102 | 48 | 2 | 3 | 4 | 1 | 0 | 2 | 54.64 | 205.3 | 0.35 | -1.58 | 0.31 | -1.74 | 0.35 | -1.65 | 0.31 | -1.76 |
| SF.321221 | 51 | 3 | 2 | 1 | 2 | 2 | 1 | 54.66 | 211.3 | 0.35 | -1.58 | 0.32 | -1.73 | 0.35 | -1.64 | 0.29 | -1.72 |
| SF.204502 | 36 | 2 | 0 | 4 | 5 | 0 | 2 | 54.66 | 181.3 | 0.35 | -1.59 | 0.28 | -1.80 | 0.34 | -1.66 | 0.26 | -1.80 |
| SF.314201 | 39 | 3 | 1 | 4 | 2 | 0 | 1 | 54.77 | 187.5 | 0.35 | -1.58 | 0.32 | -1.67 | 0.35 | -1.65 | 0.31 | -1.67 |




| | | | | | | | | | | | | | | | | |
|---|---|---|---|---|---|---|---|---|---|---|---|---|---|---|---|---|
| SF.314211 | 51 | 3 | 1 | 4 | 2 | 1 | 1 | 54.85 | 211.7 | 0.35 | -1.58 | 0.30 | -1.75 | 0.34 | -1.64 | 0.30 | -1.75 |
| SF.304211 | 48 | 3 | 0 | 4 | 2 | 1 | 1 | 54.88 | 205.8 | 0.34 | -1.61 | 0.30 | -1.70 | 0.32 | -1.68 | 0.30 | -1.70 |
| SF.321021 | 39 | 3 | 2 | 1 | 0 | 2 | 1 | 55.12 | 188.2 | 0.34 | -1.61 | 0.31 | -1.70 | 0.32 | -1.68 | 0.30 | -1.70 |
| SF.214320 | 39 | 2 | 1 | 4 | 3 | 2 | 0 | 55.15 | 188.3 | 0.33 | -1.62 | 0.32 | -1.65 | 0.32 | -1.68 | 0.33 | -1.66 |
| SF.114401 | 30 | 1 | 1 | 4 | 4 | 0 | 1 | 55.31 | 170.6 | 0.33 | -1.61 | 0.29 | -1.69 | 0.33 | -1.68 | 0.30 | -1.68 |
| SF.410311 | 84 | 4 | 1 | 0 | 3 | 1 | 1 | 55.57 | 279.1 | 0.32 | -1.64 | 0.31 | -1.76 | 0.33 | -1.68 | 0.34 | -1.75 |
| SF.203321 | 48 | 2 | 0 | 3 | 3 | 2 | 1 | 55.81 | 207.6 | 0.31 | -1.64 | 0.31 | -1.69 | 0.32 | -1.68 | 0.34 | -1.68 |
| SF.133402 | 36 | 1 | 3 | 3 | 4 | 0 | 2 | 55.97 | 183.9 | 0.31 | -1.64 | 0.29 | -1.60 | 0.32 | -1.70 | 0.29 | -1.59 |
| SF.124002 | 21 | 1 | 2 | 4 | 0 | 0 | 2 | 56.35 | 154.7 | 0.29 | -1.68 | 0.29 | -1.72 | 0.27 | -1.75 | 0.29 | -1.73 |
| SF.424520 | 84 | 4 | 2 | 4 | 5 | 2 | 0 | 56.41 | 280.8 | 0.29 | -1.66 | 0.22 | -1.78 | 0.28 | -1.71 | 0.20 | -1.79 |
| SF.423220 | 84 | 4 | 2 | 3 | 2 | 2 | 0 | 56.48 | 281.0 | 0.29 | -1.67 | 0.28 | -1.71 | 0.30 | -1.72 | 0.28 | -1.72 |
| SF.200211 | 36 | 2 | 0 | 0 | 2 | 1 | 1 | 56.49 | 185.0 | 0.28 | -1.68 | 0.25 | -1.81 | 0.28 | -1.71 | 0.23 | -1.80 |
| SF.201021 | 36 | 2 | 0 | 1 | 0 | 2 | 1 | 56.60 | 185.2 | 0.28 | -1.69 | 0.28 | -1.74 | 0.28 | -1.73 | 0.30 | -1.73 |
| SF.110221 | 30 | 1 | 1 | 0 | 2 | 2 | 1 | 56.60 | 173.2 | 0.28 | -1.69 | 0.28 | -1.74 | 0.28 | -1.73 | 0.30 | -1.73 |
| SF.201412 | 48 | 2 | 0 | 1 | 4 | 1 | 2 | 56.65 | 209.3 | 0.27 | -1.72 | 0.31 | -1.66 | 0.25 | -1.78 | 0.35 | -1.66 |
| SF.303122 | 48 | 3 | 0 | 3 | 1 | 2 | 2 | 56.75 | 209.5 | 0.24 | -1.79 | 0.23 | -1.85 | 0.25 | -1.82 | 0.24 | -1.85 |
| SF.423502 | 84 | 4 | 2 | 3 | 5 | 0 | 2 | 56.84 | 281.7 | 0.27 | -1.69 | 0.27 | -1.73 | 0.27 | -1.75 | 0.28 | -1.73 |
| SF.124222 | 39 | 1 | 2 | 4 | 2 | 2 | 2 | 57.05 | 192.1 | 0.26 | -1.70 | 0.23 | -1.79 | 0.25 | -1.76 | 0.21 | -1.80 |
| SF.103402 | 27 | 1 | 0 | 3 | 4 | 0 | 2 | 57.31 | 168.6 | 0.25 | -1.72 | 0.24 | -1.74 | 0.26 | -1.76 | 0.25 | -1.73 |
| SF.404401 | 81 | 4 | 0 | 4 | 4 | 0 | 1 | 57.37 | 276.7 | 0.25 | -1.73 | 0.19 | -1.83 | 0.24 | -1.78 | 0.18 | -1.84 |
| SF.114102 | 30 | 1 | 1 | 4 | 1 | 0 | 2 | 57.37 | 174.7 | 0.25 | -1.72 | 0.25 | -1.76 | 0.25 | -1.76 | 0.27 | -1.76 |
| SF.103521 | 36 | 1 | 0 | 3 | 5 | 2 | 1 | 57.54 | 187.1 | 0.24 | -1.73 | 0.25 | -1.75 | 0.25 | -1.77 | 0.26 | -1.75 |
| SF.303511 | 48 | 3 | 0 | 3 | 5 | 1 | 1 | 57.60 | 211.2 | 0.23 | -1.76 | 0.22 | -1.82 | 0.23 | -1.78 | 0.22 | -1.82 |
| SF.401301 | 81 | 4 | 0 | 1 | 3 | 0 | 1 | 57.66 | 277.3 | 0.22 | -1.77 | 0.18 | -1.85 | 0.21 | -1.80 | 0.17 | -1.85 |
| SF.203420 | 36 | 2 | 0 | 3 | 4 | 2 | 0 | 57.68 | 187.4 | 0.24 | -1.74 | 0.24 | -1.76 | 0.24 | -1.78 | 0.24 | -1.76 |
| SF.311312 | 51 | 3 | 1 | 1 | 3 | 1 | 2 | 57.76 | 217.5 | 0.23 | -1.76 | 0.22 | -1.84 | 0.23 | -1.78 | 0.25 | -1.82 |
| SF.223520 | 39 | 2 | 2 | 3 | 5 | 2 | 0 | 57.77 | 193.5 | 0.21 | -1.82 | 0.21 | -1.83 | 0.21 | -1.85 | 0.22 | -1.83 |
| SF.224001 | 27 | 2 | 2 | 4 | 0 | 0 | 1 | 58.07 | 170.1 | 0.22 | -1.76 | 0.22 | -1.80 | 0.21 | -1.81 | 0.24 | -1.80 |
| SF.301421 | 48 | 3 | 0 | 1 | 4 | 2 | 1 | 58.13 | 212.3 | 0.21 | -1.79 | 0.18 | -1.86 | 0.21 | -1.81 | 0.18 | -1.85 |
| SF.303301 | 36 | 3 | 0 | 3 | 3 | 0 | 1 | 58.21 | 188.4 | 0.21 | -1.78 | 0.20 | -1.86 | 0.21 | -1.81 | 0.20 | -1.85 |
| SF.321421 | 51 | 3 | 2 | 1 | 4 | 2 | 1 | 58.29 | 218.6 | 0.22 | -1.76 | 0.17 | -1.83 | 0.22 | -1.80 | 0.16 | -1.83 |
| SF.220220 | 27 | 2 | 2 | 0 | 2 | 2 | 0 | 58.53 | 171.1 | 0.19 | -1.81 | 0.19 | -1.83 | 0.18 | -1.85 | 0.19 | -1.84 |
| SF.214112 | 51 | 2 | 1 | 4 | 1 | 1 | 2 | 58.62 | 219.2 | 0.20 | -1.78 | 0.15 | -1.85 | 0.20 | -1.82 | 0.14 | -1.84 |
| SF.211512 | 51 | 2 | 1 | 1 | 5 | 1 | 2 | 58.63 | 219.3 | 0.19 | -1.80 | 0.20 | -1.81 | 0.19 | -1.83 | 0.20 | -1.81 |
| SF.231220 | 48 | 2 | 3 | 1 | 2 | 2 | 0 | 58.71 | 213.4 | 0.19 | -1.80 | 0.20 | -1.76 | 0.18 | -1.86 | 0.20 | -1.77 |
| SF.131201 | 36 | 1 | 3 | 1 | 2 | 0 | 1 | 58.81 | 189.6 | 0.18 | -1.81 | 0.21 | -1.79 | 0.19 | -1.84 | 0.23 | -1.79 |
| SF.333021 | 48 | 3 | 3 | 3 | 0 | 2 | 1 | 58.87 | 213.7 | 0.18 | -1.83 | 0.16 | -1.88 | 0.18 | -1.86 | 0.17 | -1.88 |
| SF.301211 | 48 | 3 | 0 | 1 | 2 | 1 | 1 | 58.89 | 213.8 | 0.18 | -1.82 | 0.15 | -1.89 | 0.19 | -1.84 | 0.15 | -1.89 |
| SF.221512 | 51 | 2 | 2 | 1 | 5 | 1 | 2 | 58.92 | 219.8 | 0.16 | -1.87 | 0.16 | -1.88 | 0.16 | -1.89 | 0.17 | -1.88 |
| SF.130212 | 36 | 1 | 3 | 0 | 2 | 1 | 2 | 59.30 | 190.6 | 0.16 | -1.84 | 0.17 | -1.83 | 0.15 | -1.87 | 0.18 | -1.83 |
| SF.130512 | 36 | 1 | 3 | 0 | 5 | 1 | 2 | 59.32 | 190.6 | 0.16 | -1.84 | 0.17 | -1.84 | 0.16 | -1.87 | 0.18 | -1.85 |
| SF.131500 | 27 | 1 | 3 | 1 | 5 | 0 | 0 | 59.38 | 172.8 | 0.15 | -1.85 | 0.13 | -1.88 | 0.15 | -1.88 | 0.13 | -1.88 |
| SF.310212 | 39 | 3 | 1 | 0 | 2 | 1 | 2 | 59.42 | 196.8 | 0.15 | -1.85 | 0.13 | -1.91 | 0.14 | -1.88 | 0.13 | -1.91 |
| SF.221111 | 51 | 2 | 2 | 1 | 1 | 1 | 1 | 59.44 | 220.9 | 0.16 | -1.84 | 0.17 | -1.84 | 0.15 | -1.87 | 0.18 | -1.84 |
| SF.100322 | 27 | 1 | 0 | 0 | 3 | 2 | 2 | 59.47 | 172.9 | 0.15 | -1.85 | 0.17 | -1.85 | 0.14 | -1.88 | 0.17 | -1.86 |
| SF.311021 | 39 | 3 | 1 | 1 | 0 | 2 | 1 | 59.52 | 197.0 | 0.15 | -1.85 | 0.14 | -1.91 | 0.14 | -1.88 | 0.14 | -1.91 |
| SF.210102 | 27 | 2 | 1 | 0 | 1 | 0 | 2 | 59.54 | 173.1 | 0.15 | -1.86 | 0.15 | -1.90 | 0.13 | -1.89 | 0.15 | -1.90 |
| SF.301120 | 36 | 3 | 0 | 1 | 1 | 2 | 0 | 59.58 | 191.2 | 0.13 | -1.88 | 0.10 | -1.90 | 0.11 | -1.89 | 0.09 | -1.90 |
| SF.421502 | 84 | 4 | 2 | 1 | 5 | 0 | 2 | 59.59 | 287.2 | 0.15 | -1.86 | 0.15 | -1.90 | 0.14 | -1.88 | 0.16 | -1.90 |
| SF.414011 | 84 | 4 | 1 | 4 | 0 | 1 | 1 | 59.60 | 287.2 | 0.14 | -1.86 | 0.14 | -1.91 | 0.13 | -1.88 | 0.15 | -1.91 |
| SF.323502 | 39 | 3 | 2 | 3 | 5 | 0 | 2 | 59.61 | 197.2 | 0.15 | -1.84 | 0.17 | -1.80 | 0.16 | -1.87 | 0.18 | -1.79 |
| SF.111510 | 30 | 1 | 1 | 1 | 5 | 1 | 0 | 59.64 | 179.3 | 0.13 | -1.88 | 0.11 | -1.91 | 0.12 | -1.90 | 0.11 | -1.91 |
| SF.211421 | 51 | 2 | 1 | 1 | 4 | 2 | 1 | 59.64 | 221.3 | 0.14 | -1.86 | 0.11 | -1.92 | 0.13 | -1.88 | 0.09 | -1.92 |
| SF.111502 | 30 | 1 | 1 | 1 | 5 | 0 | 2 | 59.68 | 179.4 | 0.14 | -1.87 | 0.15 | -1.88 | 0.13 | -1.89 | 0.15 | -1.88 |
| SF.133002 | 27 | 1 | 3 | 3 | 0 | 0 | 2 | 59.72 | 173.4 | 0.14 | -1.86 | 0.15 | -1.88 | 0.14 | -1.88 | 0.16 | -1.87 |
| SF.113001 | 21 | 1 | 1 | 3 | 0 | 0 | 1 | 59.77 | 161.5 | 0.15 | -1.82 | 0.12 | -1.88 | 0.17 | -1.85 | 0.11 | -1.88 |



| SF.120510 | 21 | 1 | 2 | 0 | 5 | 1 | 0 | 59.81 | 161.6 | 0.14 | -1.88 | 0.13 | -1.89 | 0.13 | -1.90 | 0.14 | -1.89 |
| SF.210322 | 39 | 2 | 1 | 0 | 3 | 2 | 2 | 59.84 | 197.7 | 0.13 | -1.88 | 0.12 | -1.91 | 0.12 | -1.90 | 0.12 | -1.91 |
| SF.130310 | 27 | 1 | 3 | 0 | 3 | 1 | 0 | 59.93 | 173.9 | 0.12 | -1.89 | 0.13 | -1.91 | 0.12 | -1.91 | 0.13 | -1.92 |
| SF.220510 | 27 | 2 | 2 | 0 | 5 | 1 | 0 | 59.94 | 173.9 | 0.13 | -1.88 | 0.12 | -1.90 | 0.12 | -1.91 | 0.13 | -1.90 |
| SF.220201 | 27 | 2 | 2 | 0 | 2 | 0 | 1 | 60.14 | 174.3 | 0.11 | -1.91 | 0.12 | -1.91 | 0.10 | -1.93 | 0.12 | -1.91 |
| SF.113201 | 30 | 1 | 1 | 3 | 2 | 0 | 1 | 60.17 | 180.3 | 0.12 | -1.88 | 0.13 | -1.93 | 0.12 | -1.90 | 0.14 | -1.93 |
| SF.123512 | 39 | 1 | 2 | 3 | 5 | 1 | 2 | 60.23 | 198.5 | 0.13 | -1.88 | 0.10 | -1.92 | 0.13 | -1.90 | 0.10 | -1.92 |
| SF.201420 | 36 | 2 | 0 | 1 | 4 | 2 | 0 | 60.24 | 192.5 | 0.10 | -1.92 | 0.09 | -1.93 | 0.09 | -1.94 | 0.09 | -1.94 |
| SF.201101 | 36 | 2 | 0 | 1 | 1 | 0 | 1 | 60.24 | 192.5 | 0.11 | -1.90 | 0.09 | -1.92 | 0.10 | -1.91 | 0.08 | -1.92 |
| SF.101320 | 27 | 1 | 0 | 1 | 3 | 2 | 0 | 60.33 | 174.7 | 0.11 | -1.90 | 0.11 | -1.92 | 0.10 | -1.92 | 0.11 | -1.92 |
| SF.300212 | 36 | 3 | 0 | 0 | 2 | 1 | 2 | 60.45 | 192.9 | 0.11 | -1.91 | 0.14 | -1.91 | 0.11 | -1.92 | 0.16 | -1.91 |
| SF.331502 | 48 | 3 | 3 | 1 | 5 | 0 | 2 | 60.51 | 217.0 | 0.10 | -1.91 | 0.08 | -1.94 | 0.08 | -1.93 | 0.08 | -1.94 |
| SF.110120 | 21 | 1 | 1 | 0 | 1 | 2 | 0 | 60.67 | 163.3 | 0.08 | -1.94 | 0.09 | -1.95 | 0.07 | -1.95 | 0.09 | -1.95 |
| SF.120120 | 21 | 1 | 2 | 0 | 1 | 2 | 0 | 60.69 | 163.4 | 0.08 | -1.93 | 0.09 | -1.94 | 0.08 | -1.94 | 0.09 | -1.94 |
| SF.111500 | 21 | 1 | 1 | 1 | 5 | 0 | 0 | 60.76 | 163.5 | 0.06 | -1.96 | 0.06 | -1.97 | 0.05 | -1.97 | 0.06 | -1.97 |
| SF.230101 | 36 | 2 | 3 | 0 | 1 | 0 | 1 | 60.91 | 193.8 | 0.07 | -1.94 | 0.08 | -1.95 | 0.07 | -1.95 | 0.08 | -1.95 |
| SF.333511 | 60 | 3 | 3 | 3 | 5 | 1 | 1 | 61.51 | 243.0 | 0.03 | -1.98 | 0.02 | -1.99 | 0.02 | -1.98 | 0.02 | -1.99 |
| SF.430021 | 81 | 4 | 3 | 0 | 0 | 2 | 1 | 61.56 | 285.1 | 0.04 | -1.98 | 0.04 | -1.98 | 0.04 | -1.98 | 0.04 | -1.98 |


**Table A 3: Performance of the SOFIA model SF.124421 depending on the type of cost function that is used in optimization.**

| Name | SOFIA model SF.124421 with different cost functions in optimization: | Comparison against GFED.BA (1997-2011) | | | | | | Comparison against CCI.BA (2005-2011) | | | |
| | | Training (1817 cells, even years in 1998-2010) Data used for training of RF and for SF parameter optimization | | | | Evaluation (1212 cells, uneven years in 1997-2011) | | Training (even years in 2006-2010) | | Evaluation (uneven years in 2005-2011) | |
| | | SSE | AIC | IoA | FV | IoA | FV | IoA | FV | IoA | FV |
| SF.SSE (SF.124421 in Tab. S2) | Default cost function, sum of squared error $$Cost = \sum_{i=1}^{i=N}(sim_i - obs_i)^2$$ | 53.40 | 184.8 | 0.40 | -1.51 | 0.39 | -1.51 | 0.39 | -1.59 | 0.41 | -1.51 |
| SF.KGE | Kling-Gupta efficiency: Euclidean distance in a 3-dimensional space defined by components for correlation, variance, and bias (Gupta et al., 2009) $$Cost = \sqrt{(r-1)^2 + \left(\frac{\sigma_{sim}}{\sigma_{obs}}-1\right)^2 + \left(\frac{\overline{sim}}{\overline{obs}}-1\right)^2}$$ $r$ is the Pearson correlation coefficient between $sim$ and $obs$ | 91.28 | 260.6 | 0.30 | -0.25 | 0.31 | -0.50 | 0.31 | -0.48 | 0.33 | -0.50 |
| SF.IoA-FV | Analogously to KGE, the Euclidean distance in a 2-dimensional space defined by IoA and FV $$Cost = \sqrt{(IoA-1)^2 + FV^2}$$ | 90.43 | 258.9 | 0.44 | 0.00 | 0.45 | -0.25 | 0.45 | -0.22 | 0.46 | -0.29 |
| SF.SSE-sqrt | Sum of squared error based on square root-transformed fractional burned area $$Cost = \sum_{i=1}^{i=N}(\sqrt{sim_i} - \sqrt{obs_i})^2$$ | 58.15 | 194.3 | 0.15 | -1.94 | 0.13 | -1.96 | 0.15 | -1.95 | 0.13 | -1.96 |
| SF.SSE-anom | Sum of squared error but with anomalies $x'$ included as additional component. $$Cost = SSE(sim, obs) + SSE(sim', obs')$$ Anomalies defined as the difference to a rolling mean value with a window length of 121 months: $$x' = x - rollMean(x)$$ | 57.20 | 192.4 | 0.25 | -1.73 | 0.20 | -1.81 | 0.22 | -1.78 | 0.19 | -1.82 |



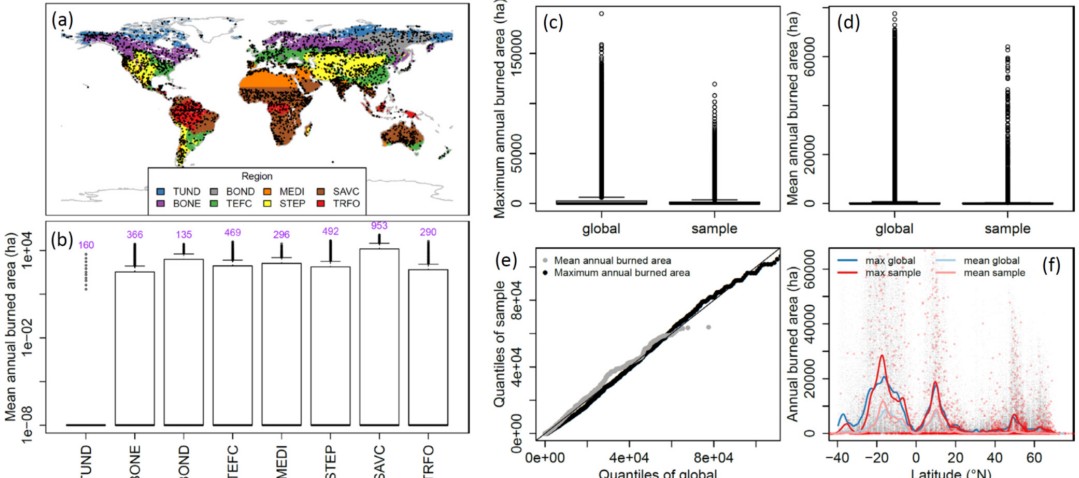

**Figure A 1: Representativeness of sampled 0.25° grid cells for global patterns of burned area (based on GFED burned data). (a) Spatial distribution of the grid cells of the optimization and evaluation data subsets and regions for regional analyses of results. Regions are TUND (tundra), BONE (boreal needle-leaved evergreen and mixed forests), BOND (boreal needle-leaved deciduous forests), TEFC (temperate forests and croplands), MEDI (Mediterranean regions), STEP (steppes), SAVC (savannahs and tropical croplands), and TRFO (tropical forests). (b) Distribution of mean annual burned area per region from the sampled grid cells. Numbers indicate the number of grid cells per regions. (c-f) Comparison of mean and maximum annual burned between all global grid cells and the sampled grid cells. (c) and (d) distribution of maximum and mean annual burned. (e) Quantiles of mean and maximum annual burned area. (f) Latitudinal gradients of annual burned area. Latitudinal gradients are smoothing splines fitted to the quantile 0.95 of mean and maximum annual burned area, respectively.**

**Author contribution**

M. Forkel and W. Dorigo designed the study and experimental setup. M. Forkel developed code, carried out the analysis, and mainly wrote the manuscript. I. Teubner contributed with data pre-processing. G. Lasslop performed JSBACH-SPITFIRE model runs. K. Thonicke and E. Chuvieco contributed with conceptual ideas and references. All co-authors discussed results and contributed to the manuscript.

**Competing interests**

The authors declare that they have no conflict of interest.

**Acknowledgements**

This work was supported by the European Space Agency through a Living Planet Fellowship for M. Forkel (CCI4SOFIE, CCI data for assessing soil moisture controls on fire emissions) and by the TU Wien Wissenschaftspreis 2015, a personal science award assigned to W. Dorigo from the Vienna University of Technology. We further thank the following organisations, projects, portals, and researchers for providing datasets: ESA CCI, GFED, CRU, GPCC, GIMMS, NASA SEDAC, NOAA EOG, and Y. Liu.



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
