# Peer review of "A data-driven approach to identify controls on global fire activity from satellite and climate observations (SOFIA V1)"

_Geoscientific Model Development, 2016_

## Short Comment (SC1) · 3 Jan 2017

Dear authors,

in my role as Executive editor of GMD, I would like to bring to your attention our Editorial version 1.1:

http://www.geosci-model-dev.net/8/3487/2015/gmd-8-3487-2015.html

This highlights some requirements of papers published in GMD, which is also available on the GMD website in the 'Manuscript Types' section:

http://www.geoscientific-model-development.net/submission/manuscript_types.html

In particular, please note that for your paper, the following requirements have not been met in the Discussions paper:

- "The main paper must give the model name and version number (or other unique identifier) in the title."

In order to simplify reference to your developments, please add the acronym "SOFIA model" and a version number in the title of your article in your revised submission to GMD.

Yours,

Astrid Kerkweg

———————————————————

---

## Author Comment (AC1) · 4 Jan 2017

Dear Astrid Kerkweg,

thank you for your short comment. We very much appreciate that you check that manuscripts meet the editorial requirements of GMD which ensures the high quality of the journal.

We were aware about the editorial requirements. Especially, we knew that #1 the model name and version number needs to be given in the title for "model description papers", and #2 that this rule applies for "development and technical papers" if the development is related to a single model or if a new development is shown with one specific model.

[Figure]

We did not include the model name and a version number in the title of our manuscript because we think that these rules do not apply to our manuscript. "SOFIA" is a more generic data-driven approach to model fire from which we created a large ensemble of different models with different types of forcing variables, different considered processes, and thus different model structures and complexities. This means that a single specific SOFIA model does not exist (although some specific models can be selected or rejected from the ensemble based on their performance). As the one single SOFIA model does not exist, we also cannot assign a version number. Based on these reasons, we did not include a model name and version in the title.

As an alternative, we suggest to include the term "SOFIA" (without a version number) in the title as the name of the modelling approach, e.g.: "Identifying required model structures to predict global fire activity from satellite and climate data: introducing the SOFIA modelling approach"

Please let us know if our reasoning and the suggested change of the title is in agreement with the editorial requirements of GMD.

Kind regards,

Matthias Forkel

---

## Short Comment (SC2) · 9 Jan 2017

Dear Matthias Forkel,

thanks for your readiness to adhere to the rules of GMD!

I agree, that in this case it is hard to figure out, what the correct procedure should be. I agree, that it does not make sense to assign a version number to the SOFIA models themselves.

What might be worth to label with a version number is the software which generates the SOFIA models. In your reply you write " "SOFIA" is a more generic data-driven approach to model fire from which we created a large ensemble of different models with

different types of forcing variables, different considered processes, and thus different model structures and complexities". From this I understand, that you use a software, which generates these ensembles. Thus, if this software changes (e.g., by further developments or bug fixes) the chosen ensembles will be different. Therefore it might be valuable to label this software (i.e., the SOFIA model approach) with a version number.

If you do not agree with this view, I think the compromise you are suggesting would be sufficient in this case.

Best regards, Astrid Kerkweg

---

## Referee Comment (RC1) · Anonymous Referee #1 · 13 Jul 2017

Dear Authors, First thank you for an extremely well-written manuscript about your exhaustive study of factors contributing to controls on burned area for climate models. I found the conclusions section especially well-written; it summarizes well the implications of your results and your conclusions are well-defended by the analysis you present.

This paper is entirely suitable for publication in GMD, I have only a few comments that perhaps can suggest where your account can be clarified.

As for the science itself, I have only a few relatively minor questions about your methodology. The strength of your conclusions is only moderate because the input datasets

you have used, which are with a few exceptions the best available, are not very good, and specifically lack skill at capturing the truly relevant properties of vegetation. This is not a flaw of your study, but merely the state of the science.

I did wonder about your choice of variables to represent climate/weather effects. First, you use a dataset based on statistical interpolation of weather station data. A reanalysis would be a much more appropriate choice: statistical interpolation of weather station data will have obvious consequences for your analysis: for instance, interpolation of (dense) coastal weather data into (data-sparse) inland areas will produce erroneous results in near-coastal interiors. Reanalysis data would not completely solve this problem, but would surely better capture the weather in fire-prone areas.

Second, the variables you use are "mean temperature, mean diurnal temperature range, mean number of wet days, and the total precipitation of the actual month and the 12 months before a fire."

1) What is the role of "diurnal temperature range" with regards to wildfire? It seems like a very loosely related quantity.

2) Would you not get better results by using temperature and rainfall anomalies, rather than absolute values? Or perhaps this would make no difference in your analysis.

Besides that question, I have only two other minor comments:

Line 350: "(e.g. quantiles 0.01 to 0.02)" this is not clear to me; generally when I hear "quantiles" I think "bottom 20%" or "top 25%" or things like that.

Line 419: "explained reasonable" -> "explained reasonably" This was the only typo I encountered in the entire manuscript!

---

## Referee Comment (RC2) · Anonymous Referee #1 · 13 Jul 2017

Thank you for the detailed response.

I recommend you include that Figure 2 showing the variables contributing to burned area in the random forest model and the associated discussion in the final manuscript, as supplementary material. That is very interesting analysis.

---

## Author Comment (AC2) · 13 Jul 2017

Referee comments are cited in *italics* and author's responses are written in normal font.

*Dear Authors, First thank you for an extremely well-written manuscript about your exhaustive study of factors contributing to controls on burned area for climate models. I found the conclusions section especially well-written; it summarizes well the implications of your results and your conclusions are well-defended by the analysis you present. This paper is entirely suitable for publication in GMD, I have only a few comments that perhaps can suggest where your account can be clarified.*

[Figure]

Dear Referee 1, we thank you for a very positive review and we are happy to respond to your questions.

*As for the science itself, I have only a few relatively minor questions about your methodology. The strength of your conclusions is only moderate because the input datasets you have used, which are with a few exceptions the best available, are not very good, and specifically lack skill at capturing the truly relevant properties of vegetation. This is not a flaw of your study, but merely the state of the science.*

We completely agree to this point. We mentioned and discussed this issue already in chapters 2 and 5.1. We hypothesize that the use of datasets from newer satellite sensors with possibly higher quality retrievals might result in higher model performances. However, this needs to be demonstrated in a potential follow-up study. Moreover, we think that fire modelling could advance from more fire-relevant satellite products (e.g. time series of biomass and fuel loads instead of FAPAR or VOD; or fuel moisture instead of surface soil moisture) than the ones that we used here.

*I did wonder about your choice of variables to represent climate/weather effects. First, you use a dataset based on statistical interpolation of weather station data. A reanalysis would be a much more appropriate choice: statistical interpolation of weather station data will have obvious consequences for your analysis: for instance, interpolation of (dense) coastal weather data into (data-sparse) inland areas will produce erroneous results in near-coastal interiors. Reanalysis data would not completely solve this problem, but would surely better capture the weather in fire-prone areas.*

We agree that re-analysis data might better resolve fire weather conditions in regions were interpolation-based datasets rely on remote information. However we had two reasons for using interpolation-based climate data (CRU and GPCC datasets). Firstly, the CRU and GPCC datasets are also commonly used as forcing within global vegetation/fire models (e.g. (Schaphoff et al., 2013; Thonicke et al., 2010)) or re-analysis

datasets are corrected by such datasets like in the CRU-NCEP dataset that is used in several vegetation model-inter-comparison projects like TRENDY or FireMIP (Rabin et al., 2017). As our study aimed to provide suggestions for the development of global vegetation/fire models, we here relied on comparable forcing datasets. Secondly, we nevertheless tested beforehand the influence of using alternative climate datasets but found no generally strong effect on model performance. As differences in climate forcing datasets are usually more associated to differences in precipitation than in temperature, we tested the capability of several precipitation datasets to predict burned area within the random forest machine learning approach (Figure 1). Specifically, we compared the predictive capabilities of the number of wet days from CRU (WET, pink in Fig. 1) with precipitation from GPCC (used in paper, brown in Fig. 1), CRU (yellow in Fig. 1), and GPCP (violet in Fig. 1) (Huffman et al., 2009). In GPCP, satellite information are additionally used to rain gauge data. Thus GPCP should potentially better account for the spatial-temporal variability in precipitation. We found marginally better performances in predicting burned area when using GPCP than GPCC or CRU in boreal, temperate and tropical forests but slightly worse performance in steppes and the Mediterranean. All precipitation datasets and the number of wet days resulted in very similar performances at the global scale. In summary, at local to regional scales the prediction of burned area is sensitive to the chosen meteorological forcing dataset. However, we could not identify a precipitation datasets that would result in clearly better fire model performances at biome- to global scales.

*Second, the variables you use are "mean temperature, mean diurnal temperature range, mean number of wet days, and the total precipitation of the actual month and the 12 months before a fire." 1) What is the role of "diurnal temperature range" with regards to wildfire? It seems like a very loosely related quantity. 2) Would you not get better results by using temperature and rainfall anomalies, rather than absolute values? Or perhaps this would make no difference in your analysis.*

Diurnal temperature range (DTR) has been long used as predictor for fire weather con-
ditions. DTR is sensitive to stable weather conditions, i.e. DTR is usually high under high pressure systems with low cloud cover that allow high maximum temperature during day time but low temperatures at the morning because of the strong long-wave radiation loss during night-time (Lewis and Karoly, 2013). Such weather conditions are usually associated to low humidity and thus are supportive for fire activity. Therefore, DTR has been used as predictor variable in fire weather indices such as the Nesterov index (Nesterov, 1949; Venevsky et al., 2002) or is also used in fire models such as SPITFIRE (Thonicke et al., 2010). Also newer analyses based on satellite and climate data have shown that DTR shows a strong sensitivity to burned area (Bistinas et al., 2014). We also found in our preparatory analyses for this study that DTR is one of the most important (rank 5) predictor variables for global spatial-temporal dynamics of burned area (Figure 2).

We also initially thought that using anomalies of precipitation, temperature, soil moisture, or vegetation variables would be more relevant than using absolute values. To test the importance of several variables for the prediction of burned area, we computed several statistical properties of each variable and used random forest to quantify the importance of variables (Figure 2). Statistical properties were for example monthly anomalies relative to the mean seasonal cycle or averaged absolute values and anomalies over several pre-fire months (in total 132 variables were included in this analysis). Surprisingly, we found generally a higher importance of the absolute variables than of the anomalies. For example, short-term anomalies of precipitation (GPCC.P.anom) or soil moisture (CCI.SM.anom) had very low importance (below rank 53, not included in Fig. 2). We think that the lower importance of anomaly-based variables is caused by the fact that measurement noise is more prominently included in anomaly time series than in absolute-value time series. However, anomaly-based variables might have more predictive capabilities than absolute variables at regional scales which will strongly depend on the regional data quality. The most important anomaly-based variable was the average of the anomaly of wet days in the actual month and the 12 months before a fire (CRU.WET.anom.filter13, rank 3). However such a variable has only a limited phys-

ically meaning for fire activity but it likely represent an indirect effect of precipitation on vegetation productivity and thus fuel production. For the development of SOFIA models, we finally selected the set of predictor variables based on their importance (Figure 2), their interpretability, and based on how closely they are related to fire activity (by avoiding variables that account for indirect effects).

*Besides that question, I have only two other minor comments: Line 350: "(e.g. quantiles 0.01 to 0.02)" this is not clear to me; generally when I hear "quantiles" I think "bottom 20%" or "top 25%" or things like that.*

The quantiles 0.01 and 0.02 are the percentiles 1% and 2%, respectively. The terms quantiles and percentiles are often not precisely used. Here, we mean (for example) that we computed the quantiles 0.01 (i.e. 1% of values are below this value) and 0.02 (i.e. 2% of values are below this value) of annual burned in each region. Consequently, some regional grid cells have annual burned areas that fall between the quantiles 0.01 and 0.02. From these grid cells, we took a random sample to be used in the training dataset. We repeated this procedure for all 0.01 quantile ranges between minimum (quantile 0) and maximum (quantile 1) to include in the training dataset grid cells that represent the entire regional statistical distribution of burned area.

*Line 419: "explained reasonable" -> "explained reasonably" This was the only typo I encountered in the entire manuscript!*

We will change this typo (and a few others) in the revised manuscript.

**Figure captions**

Figure 1: Effect of different climate and vegetation datasets on the regional index of agreement between observed (GFED dataset) and predicted (by different random forest models) burned area. Shown is the index of agreement for the evaluation data subset.

Figure 2: Importance of several variables to predict monthly burned area using random forest. Importance is expressed as the percentage increment in mean squared error if a certain variable is not included in random forest. Thus, the most important variables cause the largest increment in MSE. Variables that include "orig" or "anom" indicates original absolute values and anomalies (relative to the mean seasonal cycle), respectively. "filterX" indicates mean values over the X months before the actual month for which burned area should be predicted. In total 132 variables were included in this analysis but variables below ran 53 are not shown in this figure).

**References**

Bistinas, I., Harrison, S. P., Prentice, I. C. and Pereira, J. M. C.: Causal relationships versus emergent patterns in the global controls of fire frequency, Biogeosciences, 11(18), 5087–5101, doi:10.5194/bg-11-5087-2014, 2014.

Huffman, G. J., Adler, R. F., Bolvin, D. T. and Gu, G.: Improving the global precipitation record: GPCP Version 2.1, Geophys. Res. Lett., 36(17), L17808, doi:10.1029/2009GL040000, 2009.

Lewis, S. C. and Karoly, D. J.: Evaluation of Historical Diurnal Temperature Range Trends in CMIP5 Models, J. Clim., 26(22), 9077–9089, doi:10.1175/JCLI-D-13-00032.1, 2013.

Nesterov, V. G.: Gorimost' lesa i metody eio opredelenia, Goslesbumaga, Moscow., 1949.

Rabin, S. S., Melton, J. R., Lasslop, G., Bachelet, D., Forrest, M., Hantson, S., Kaplan, J. O., Li, F., Mangeon, S., Ward, D. S., Yue, C., Arora, V. K., Hickler, T., Kloster, S., Knorr, W., Nieradzik, L., Spessa, A., Folberth, G. A., Sheehan, T., Voulgarakis, A., Kelley, D. I., Prentice, I. C., Sitch, S., Harrison, S. and Arneth, A.: The Fire Modeling Intercomparison Project (FireMIP), phase 1: experimental and analytical protocols with detailed model descriptions, Geosci Model Dev, 10(3), 1175–1197, doi:10.5194/gmd-

10-1175-2017, 2017.

Schaphoff, S., Heyder, U., Ostberg, S., Gerten, D., Heinke, J. and Lucht, W.: Contribution of permafrost soils to the global carbon budget, Environ. Res. Lett., 8(1), 014026, doi:10.1088/1748-9326/8/1/014026, 2013.

Thonicke, K., Spessa, A., Prentice, I. C., Harrison, S. P., Dong, L. and Carmona-Moreno, C.: The influence of vegetation, fire spread and fire behaviour on biomass burning and trace gas emissions: results from a process-based model, Biogeosciences, 7(6), 1991–2011, 2010.

Venevsky, S., Thonicke, K., Sitch, S. and Cramer, W.: Simulating fire regimes in human-dominated ecosystems: Iberian Peninsula case study, Glob. Change Biol., 8(10), 984–998, 2002.
* * *
**Fig. 1.** Effect of different climate and vegetation datasets on the regional index of agreement ...

[Figure]

**Fig. 2.** Importance of several variables to predict monthly burned area using random forest ...

---

## Referee Comment (RC3) · Anonymous Referee #2 · 31 Aug 2017

The authors simulate fraction burned area at the global scale using a flexible modelling approach that is fed by climate and satellite-derived data. The topic is relevant because fire is an important component of the Earth System, and improving our estimates of fire activity can improve our capability of the Earth System itsefl. Nevertheless, I have serious doubts that the way the study is designed will help move the body of knowledge of fire in the Earth System forward. At some point in the work it seems like the authors are adding variables to a box without clearly understanding their relation\interaction with fire activity. In my opinion, this does not move the body of knowledge forward. Instead this is an exploratory approach that could be used to identify important variables and relation forms (or structures), although I don't think it will ever "identify required

model structures". In my opinion the work would benefit by a more constrained analysis instead of adding too many things to a box and trying to make sense of it in the end.

Some major comments:

1) The equifinality in the modelling approach is clear. Choosing 1 model from a bunch of good candidate models and identifying the controls of fire activity just misses the whole point. It gets more critical if you think that not all combinations were tested. What would make sense in my opinion would be analyze all valid models and trying to extract relevant information on the most important variables and influential controlling factors. Not sure on how you could do that but perhaps it would be worthwhile to take a look at Keith Beven's work and the GLUE methodology (references below).

2) The spatial resolution is a problem, as you correctly mentioned (L693-698). You are modelling fire closer to the landscape scale than the fire regime scale. To model fire activity at such a fine scale you would need other type of information and confine your analysis to specific regions. I think the work would gain a lot by doing the analysis at coarser resolutions.

3) The work needs a more solid background on how environmental and human factors influence fire patterns. This is absolutely crucial, frames the entire work and I suggest you do this at the beggnining of the Introduction. The "effects" described in 3.2 need this background information. Your work assumptions should be clear and supported by the correct literature.

4) The work is lacking many important references of relevant past works. Please see the list at the bottom of this document.

5) A proper analysis on the importance of crop inclusion is not made. It seems that there is a compensation between adding explicitly "crops" and adding a "human-related variable". This is strange for several reasons, since fire is used as a tool in croplands

all over the globe (see Korontzi et al 2006) both in poor and rich countries, as well in dense or sparsely populated regions (see Li et al. 2013). Mmost importantly you need to better support and investigate this result. Additionally, as you mention in the Discussion, you do not mimic the shape of the function that relates pop. Density with ignitions (e.g. Pechony & Schindell 2009), therefore excluding pop. density is most likely the problem of the model rather the problem of the variable. This in fact troubles me because I can make the same question for several other "effects": is the lack of importance of an effect\variable the consequence of having bad variables and\or models?

6) The comparisons of the impact of controlling effects and predictor variables on model performance made along the work are troublesome. I highlight one clear example. The land cover grouping scheme\classes is especially relevant for your modelling approach, since it controls model performance and model complexity. The analysis that you made does not help the reader to reach a clear conclusion. In one hand, PFTs lead to the best IoAs, on the other hand the additional complexity is extremely penalized. Why don't you make a simple a robust comparison with the same (or similar) degree of complexity? Do you need all classes in all land cover schemes?

7) The way conclusions are drawn needs to be moderated. You have a "virtual lab" called SOFIA that can help make some experiments, it will not inform you about the truth, certaintly not about what really determines fire activity at the global scale. An example (there are many) is related with the way you interpret the importance or not of separating crops from herbaceous. The only implication is on model performance...not on reality itself. Fires occur in croplands and in herbaceous areas for different reasons, independently of what the importance of the variables or controlling functions points out.

8) Some contraditory results : RF is better globally although you state that SOFIA outperforms both RF and JBACH. Fig 2 shows clear improvements depending on land cover grouping scheme which is contradicted in L523-524.

9) The stratified sampling is necessary and interesting, but confusing. It looks like you stratified in space, per biome (which biomes??), and then on the level of fire activity. Finally you sample in time, correct? I am not sure I totally understood the methodology I suggest you improve the way it is written and take a look at Boschetti et al 2015.

10) The abstract needs to be completely rewriten, so that the reader can undersand the relevance of the problem, what limitations you identified and are going to tackle, how are you going to tackle these limitations (really missing this one), your major results and finally implications for the evolution of the body of knowledge on this area.

Some Minor Commnts

11) There are Several terms in the text need to be revised: - fire suppression: Moisture does not suppress fires, it contrains\reduces\restricts fire activity (see Bradstock et al 2010) Firefighters suppress fires. - You mention several times "before a fire". You are modeling the occurrence of a fire, if it doesn't happen you still need predecent conditions, so I suggest you drop the term.

12) NLDIg adds extra parameters depending on the number of land cover groups. Did you consider this in your 100 parameters threshold? Did you consider this in the AIC calculation? Is the calculation shown in L302-303 correct?

13) Figure 2: Number of variables or number of controlling factors? Maybe I missed something, but each controlling actor can have multiple variables? (e.g. temperature can have both DTR and annual temp?). If not, as I originally understood, then I don't understand that does "number of variables" mean in Figure 2.

Some suggestions: - A pairwise plot between estimated and GFED\CCI burned area - Maps of differences in estimated-observed burned area - A map showing disagreements between GFED and CCI would help frame the evaluation better. (low expectation on areas where both disagree)

At this stage I don't think it makes since to go into gramar details.

The references I mentioned previsouly:

Le Page et al. 2010. Seasonality of vegetation fires as modified by human action: observing the deviation from eco‐climatic fire regimes Magi et al 2012. Separating agricultural and non-agricultural fire seasonality at regional scales Le Page et al 2015. HESFIRE: a global fire model to explore the role of anthropogenic and weather drivers. Li et al 2013 Quantifying the role of fire in the Earth system – Part 1: Improved global fire modeling in the Community Earth System Model (CESM1),. Bradstock 2010. A biogeographic model of fire regimes in Australia: current and future implications (the "switches" that control fire); Boschetti et al 2016. A stratified random sampling design in space and time for regional to global scale burned area product validation Beven & Binley, 1992. The future of distributed models: model calibration and uncertainty prediction. Beven, 2002. Towards a coherent philosophy for modelling the environment. Beven & Binley, A., 2014. GLUE: 20 years on. Pettinari & Chuvieco 2016. Generation of a global fuel data set using the Fuel Characteristic Classification System Rabin et al. 2015 Quantifying regional, time-varying effects of cropland and pasture on vegetation fire (the effects of crops on burned area) Broxton et al. 2014. A Global Land Cover Climatology Using MODIS Data (for Discussion) Korontzi et al. 2006. Global distribution of agricultural fires in croplands from 3 years of Moderate Resolution Imaging Spectroradiometer (MODIS) data.

---

## Author Response (AR1)

**Responses and proposed changes to referees for gmd-2016-301**

Matthias Forkel, Wouter Dorigo, Gitta Lasslop, Irene Teubner, Emilio Chuvieco and Kirsten Thonicke

**Note to formatting:** *We repeat referee comments in italics and orange*. Author responses are in normal font. Our proposed changes to the manuscript are coloured blue. Cross-references to figures in the initial discussion manuscript and in the response letter are indicated as Figure # and Figure R#, respectively. References to line numbers refer to the initial discussion manuscript.

**1 Response to Anonymous Referee #1**

Our response to Anonymous Referee #1 is published online in the interactive discussion at https://www.geosci-model-dev-discuss.net/gmd-2016-301/.

In response to referee 1, we include Figure A1 in the appendix of the manuscript and add a paragraph on the pre-selection of predictor variables in chapter 2:

"We used datasets of global monthly burned area as response variable and several datasets on land cover, climate, soil moisture, vegetation state, and socioeconomic factors as predictor variables in model development. To make pre-selection of relevant predictor variables, we first tested the predictive performance of various candidate variables such as absolute values, anomalies, or long-term precedent mean values of precipitation, wet days, soil moisture, or vegetation state (Figure A1). We generally found a higher importance of the absolute variables than of the anomalies. For the development of SOFIA models, we finally selected the set of predictor variables based on their importance, their interpretability, and based on how closely they are related to fire activity (by avoiding variables that account for indirect effects) (Table 1)."

**2 Response to Anonymous Referee #2**

*The authors simulate fraction burned area at the global scale using a flexible modelling approach that is fed by climate and satellite-derived data. The topic is relevant because fire is an important component of the Earth System, and improving our estimates of fire activity can improve our capability of the Earth System itsefl. Nevertheless, I have serious doubts that the way the study is designed will help move the body of knowledge of fire in the Earth System forward.*

Dear Referee 2, we are grateful for the critical review that will help to improve the quality of the manuscript. Given the contrasting assessments of our study from referees 1 and 2, apparently the scientific community does not unanimously question the contribution of our approach to moving forward the body of knowledge of fire in the Earth system. However, we also acknowledge that we may have given the wrong impression that our approach aims to provide a complete and detailed assessment of controlling factors of fire activity in the Earth system. Such an assessment is beyond the scope of our study and also beyond the primary

scope of model description papers in GMD. Thus, to avoid overambitious claims and to make the aims of our study more clearly we propose to rewrite the sentence at line 94-98 as follows:

"Here we aim to describe and apply a flexible data-driven fire modelling approach, called SOFIA (Satellite Observations for FIre Activity). The SOFIA approach provides a framework to identify the importance and the functional relationships between observational datasets and the spatial and temporal variability of burned area while revealing model formulations that could be easily adapted for more complex vegetation-fire models. We test the approach using observational datasets of land cover, climate conditions, soil moisture, vegetation state, and socioeconomics."

*At some point in the work it seems like the authors are adding variables to a box without clearly understanding their relation \ interaction with fire activity. In my opinion, this does not move the body of knowledge forward. In- stead this is an exploratory approach that could be used to identify important variables and relation forms (or structures), although I don't think it will ever "identify required model structures". In my opinion the work would benefit by a more constrained anal- ysis instead of adding too many things to a box and trying to make sense of it in the end.*

We fully agree with the referee that our study did not "identify required model structures". Instead, our study proposes a data-driven modelling approach that can be used as a tool to reach this goal. Therefore, we intentionally used the term "identifying" instead of "identification" in the title. Indeed, we follow an exploratory or data-driven approach to identify important variables and their relationships with burned area. We would like to point out that we follow a different paradigm in model development than in classical model development. Classical model development follows phases of data acquisition (measurements or data collection), and the development of conceptual, mathematical, and computational models (Gupta et al., 2012). Thereby, the development of the mathematical and computational models are based on the conceptual model, i.e. based on assumptions or theories about how a system works. In contrary, we are here following a data-driven approach (also referred as reverse approach or reverse engineering) (Solomatine and Ostfeld, 2008). In the data-driven approach, the conceptual model is developed after adequate mathematical and computational models have been inferred from the data (Gupta et al. 2012). Adequate models as identified by data-driven approaches or in model optimization experiments are also referred to as "behavioural" models within the GLUE approach (Beven and Binley, 2014). Thus, the derived mathematical and computational models possibly allow generating new hypotheses about the system. Previously, data-driven approaches based on machine learning methods have been successfully used to identify controlling factors for fire (Aldersley et al., 2011; Archibald et al., 2009; Bistinas et al., 2014). Our work builds on these approaches but aims to go a step further by identifying relations between single variables and burned area (we call this model structures), which could be adopted within more complex fire models or which could form the basis for a new empirical fire model. Of course, such relations cannot be directly transferred to more complex models but likely require re-calibration because ecosystem state variables such as FAPAR or VOD follow different definitions than in the observations or are not available in vegetation models. Following the data-driven paradigm, we intentionally did not (further) constrain our analysis approach. However, we agree with the referee that fire

model development could potentially benefit from a hybrid approach, which includes expert knowledge or existing theories and models within the data-driven modelling procedure (Solomatine and Ostfeld, 2008). This is however beyond the scope of our study.

We propose to make clearer that we aim to describe and apply a data-driven approach of model development but that we don't aim to provide a complete understanding of controls and mechanisms for fire activity. First, we suggest changing the title of the manuscript:

"A data-driven approach to identify controls on global fire activity from satellite and climate observations (SOFIA V1)"

Secondly, we propose to rewrite lines 63-77 of the Introduction:

"Despite such recent model developments, it is not clear which functional relationships, complexity, and model parametrizations are most adequate to represent fire activity (Hantson et al., 2016).

Satellite observations of fire activity can be further integrated with fire models to estimate model parameters or to assess the adequacy of functional relationships (Knorr et al., 2014; Lasslop et al., 2015; Le Page et al., 2015). For example, parameters of empirical relations were optimized in SIMFIRE (simple fire model) to predict annual fire frequency from vegetation conditions, fire weather conditions, and population density (Knorr et al., 2014). Such parameter optimization approaches are one aspect of model-data integration or model-data fusion  that encompasses a continuous cycle from the definition of model structures (i.e. predictor variables and functional relationships), estimation of model parameters, generalization or upscaling of the model, evaluation of model results, to model application and potentially back to a reformulation of the model structure (Keenan et al., 2011; Williams et al., 2009). However, a full model-data integration cycle has been rarely applied in the development of global fire models.

In comparison to process-oriented global vegetation-fire models, data-driven approaches provide an alternative framework to understand and model climate, vegetation, and socioeconomic controls on fire activity. While the development of mathematical and computational process-oriented vegetation-fire models usually starts from a conceptual model (Gupta et al., 2012), data-driven approaches aim to derive mathematical and computational models directly from the data (Solomatine and Ostfeld, 2008). In data-driven approaches, algorithms from artificial intelligence (e.g. neural networks), machine learning (e.g. random forest), or evolutionary algorithms (e.g. genetic optimization) are applied to predict a response variable (here burned area, or fire counts) from a set of potential predictor variables (Solomatine and Ostfeld, 2008). If an adequate data-driven model has been derived, the importance of individual variables and the sensitivities of the response variable to the predictor variables allow to develop a conceptual model about the studied system (Solomatine and Ostfeld, 2008). In global fire modelling, data-driven fire models have been developed using machine learning algorithms such as generalized linear models (Bistinas et al., 2014), maximum entropy (Parisien et al., 2016), or random forest (Aldersley et al., 2011; Archibald et al., 2009) mainly to identify controls on fire activity. However, such machine learning models often have complex structures, are seen as "black boxes", and thus cannot be easily adapted or even implemented within process-oriented global vegetation/fire models. Alternatively,

empirical fire models like SIMFIRE (Knorr et al., 2014) could be generalized to integrate several different candidate predictor variables and to then assess the importance and functional relationships. Consequently, such a flexible data-driven but functional fire modelling approach would allow exploring different predictor variables similar as in machine learning algorithms while potentially revealing model structures that can be more easily adapted for process-oriented vegetation-fire models."

*Some major comments:*

*1) The equifinality in the modelling approach is clear. Choosing 1 model from a bunch of good candidate models and identifying the controls of fire activity just misses the whole point. It gets more critical if you think that not all combinations were tested. What would make sense in my opinion would be analyze all valid models and trying to extract relevant information on the most important variables and influential controlling factors. Not sure on how you could do that but perhaps it would be worthwhile to take a look at Keith Beven's work and the GLUE methodology (references below).*

Equifinality, i.e. the presence of multiple adequate models and parameter sets that result in very similar responses, is a general problem in environmental modelling (Beven, 2006). It is obvious that equifinality is relevant in our study where we optimize model structures to predict fire activity. Equifinal "behavioural" models can be identified by assessing and comparing the adequacy of model structures (Gupta et al., 2012) and by assessing parameter covariances (Beven and Binley, 2014). General approaches to identify behavioural models and to avoid equifinal models are the use of multiple datasets of the same variable to account for errors or uncertainties in model forcing or reference data, the testing of different cost functions to constrain certain parameters, the inclusion of prior parameter uncertainties in the cost function, or the application of models to new observational data or under different conditions (Beven, 2006; Beven and Binley, 2014; Williams et al., 2009). The GLUE methodology is a likelihood-based approach to identify behavioural models and to quantify uncertainty in model predictions. Thereby, uncertainty quantification usually depends on the optimization of model parameters using global search algorithms such as Monte Carlo or evolutionary algorithms. Similarly, we also used an evolutionary algorithm (i.e. genetic optimization using derivatives (Mebane and Sekhon, 2011)) to train each SOFIA candidate model. Within GLUE, prior parameter uncertainties are often used to constrain the likelihood of each model. Contrary, we intentionally did not use prior parameter uncertainties in the estimation of SOFIA model parameters because #1 we intentionally did not want to constrain the analysis and #2 because prior parameter values and uncertainties were not available for SOFIA models. However, our results allow extracting parameter values and ranges for each functional relationship that could be used as prior parameter values in a potentially more constrained analysis. For example, parameters that control the functional relationship to NLDI, to DTR and WET in shrublands, and to VOD were well constrained in the SOFIA model SF.124421 (Figure R 2).

Besides the use of prior parameter uncertainties, we indeed applied all the approaches to avoid equifinality: We first generated several candidate models and assessed their global adequacy using a sample of grid cells (Table A2). From this analysis, we identified four adequate models

that represent global fire dynamics ("best" models, Table 2). We then applied and evaluated these best models based on all global burned area data and by using both the GFED and CCI burned area datasets as reference (i.e. accounting for observational uncertainty) (Fig. 4 and 5). This analysis helped to identify only one model that is adequate in all regions. However, it is true that in savannas, tropical croplands and tropical forests the four best SOFIA models perform relatively equally, i.e. all are adequate models in these regions (Fig. 5). Therefore, controls on fire activity can be analysed from all four models in these regions. As an example, we additionally analysed the controls from all best SOFIA models for Africa and the Mediterranean (Figure R 1).

We propose to include a more detailed discussion on equifinality (from line 704 onwards) and to change the title of chapter 5.1:

"5.1 Performance and equifinality of SOFIA models"

"The four best SOFIA models reached similar performances in savannas and tropical croplands, and in tropical forests which demonstrates the equifinality in fire modelling. Equifinality, i.e. the presence of multiple adequate models and parameter sets that result in very similar responses, is a general problem in environmental modelling (Beven, 2006). General approaches to avoid equifinal models are the use of multiple datasets of the same variable to account for errors or uncertainties in model forcing or reference data, the testing of different cost functions to potentially constrain certain parameters, the inclusion of prior parameter uncertainties in the cost function, or the application of models to new observational data or under different conditions (Beven, 2006; Beven and Binley, 2014; Williams et al., 2009). In our analysis, we were able to rule out three from four initially equifinal SOFIA models based on the application of these models to the full global data and by regional comparisons against two burned area datasets. The presence of equifinality in SOFIA model structures suggests to also include prior parameter uncertainties for each functional relationship to better constrain SOFIA models. However, prior parameter estimates and uncertainties were not available as we here applied the SOFIA approach for the first time. The results from the optimized SOFIA models allow extracting parameter values and ranges for each functional relationship. For example, parameters that control the functional relationship to socioeconomic development (NLDI), to diurnal temperature range and to the number of wet days in shrublands, and to VOD were well constrained in the SOFIA model SF.124421 (Figure A3). These parameters could be potentially used as prior parameter values in a more constrained analysis."

We will include Figure R 2 as Figure A 3 in the main text.

[Figure]

*Figure R 1: Equifinality in controls on fire activity from the four best SOFIA models. The maps are RGB composites of the human influence (red), the vegetation effects (green, mean of direct and long-term vegetation effect), and the climate effect (blue, mean of wetness and temperature effect). The spatial variability of each controlling factor is shown as latitudinal gradient where f(x) = 0 and f(x) = 1 indicate fully constrained and maximum fire activity, respectively.*

[Figure]

*Figure R 2: Uncertainty in parameters of SF.124421 after genetic optimization. Shown are distributions (outlines), mean values (= 1), and confidence intervals (bars) for mean values for each parameter in SF.124421. Plotted are parameters from equally good-performing parameter sets (i.e. > 0.8 \* normalized likelihood NLL, NLL = LL / max(LL) with LL = exp(-cost)).*

*2) The spatial resolution is a problem, as you correctly mentioned (L693-698). You are modelling fire closer to the landscape scale than the fire regime scale. To model fire activity at such a fine scale you would need other type of information and confine your analysis to specific regions. I think the work would gain a lot by doing the analysis at coarser resolutions.*

Here we used a spatial resolution of 0.25°. This resolution is common to many of the used datasets (e.g. GFED.BA, CCI.SM, VOD) and only slightly higher than that of the highest-resolution global vegetation/fire models (0.5°). We intentionally applied the analysis at the 0.25° resolution because predictor variables need to be spatially averaged for coarser resolution and thereby the variability in land cover and temporal dynamics of weather or vegetation conditions will be reduced as well. Thus using higher resolution data potentially helps to better identify underlying relationships than using coarse resolution data.

*3) The work needs a more solid background on how environmental and human factors influence fire patterns. This is absolutely crucial, frames the entire work and I suggest you do this at the beggning of the Introduction. The "effects" described in 3.2 need this background information. Your work assumptions should be clear and supported by the correct literature.*

First, we would like to point out again that, according to the data-driven approach of model development we intentionally aim to avoid too many assumptions. We shortly summarized the factors for fire at the beginning of the Introduction (lines 35-39) and describe how different satellite datasets can potentially represent controls on fire (lines 78-93). We hope that with

explaining our motivation better, e.g. the main purpose is the description of the SOFIA model approach, this concern is also partly addressed.

*4) The work is lacking many important references of relevant past works. Please see the list at the bottom of this document.*

Many thanks for these references. Although we were aware about some of these references, we did not include them in the first version of the manuscript to focus on the most relevant for global process-oriented and data-driven fire modelling. We will include the reference at appropriate places. See also our reply to comment #1 where we now refer to the GLUE approach and to comment #5 were we now include references for the importance of agriculture.

*5) A proper analysis on the importance of crop inclusion is not made. It seems that there is a compensation between adding explicitly "crops" and adding a "human-related variable". This is strange for several reasons, since fire is used as a tool in croplands all over the globe (see Korontzi et al 2006) both in poor and rich countries, as well in dense or sparsely populated regions (see Li et al. 2013). Mmost importantly you need to better support and investigate this result.*

Fires in croplands are mainly used to remove harvest residues or to fertilize soils and thus are indeed an important tool in agriculture. However, 89% to 92% of all global fires occur outside of croplands (Korontzi et al., 2006). Burned area decreased globally with increasing cropland area (Bistinas et al., 2014) and croplands have rather small fire sizes than non-agricultural regions (Hantson et al., 2015). Thus our approach was maybe not sufficiently able to discover the role of agricultural fires because we were using the GFED4 datasets without the correction for small fires. Many global vegetation/fire models do not account for agricultural fires (Hantson et al., 2016; Rabin et al., 2015) because regional practices of agricultural fire use are diverse and cannot be easily parameterized, and because agricultural fires do not directly affect natural vegetation and carbon cycle dynamics (unless agricultural fires escape to nearby forests). Accordingly, we never had the aim to describe mechanisms of fire use within agricultural stands but used the fractional coverage of croplands as predictor for the overall burned area within a grid cell. As the referee correctly states, in this sense the use of either croplands in the land cover grouping scheme or the use of NLDI as human effect on fire activity result in similar model performances. To further investigate the relation between cropland cover and burned area, we analysed the partial dependency of the burned area on the coverage of croplands as predicted by a random forest model (Figure R 5). We found that the relationships between cropland cover and burned area is non-monotonic, non-linear, and varies per biome (Figure R 5, right panel). These patterns are the reason why cropland cover does not add to model performance in SOFIA models that rely on monotonic functional relationships.

We propose to improve the discussion on the role of agricultural fires (lines 707-714):

"The derived SOFIA models and the spatial patterns of sensitivities show a sharp decline of burned area with increasing socioeconomic development or population density and thus agree

with previous studies that show a primarily negative effect of human activities, population density, or croplands on burned area (Andela et al., 2017; Archibald et al., 2013; Bistinas et al., 2014; Chuvieco and Justice, 2010; Knorr et al., 2014). Strikingly, our results suggest that human effects on global burned area can be expressed by either cropland area, NLDI, or population density but the combination of these factors did not improve the performances of SOFIA models. These variables serve all as proxies for the negative relationship between humans and burned area but do not directly describe human activities of fire use or suppression. For example, regional studies have shown that various information on infrastructure, land use, and other relevant socioeconomic indicators are important to predict fire activity (Archibald et al., 2009; Arndt et al., 2013; Parisien et al., 2016). However such spatially- and temporally-resolved datasets and assessments are missing for the global scale. Certainly, our results do not imply that croplands are unimportant for the global variability of burned area. Agricultural fires account for around 10% of all global fires (Korontzi et al., 2006) and for around 5% of global burned area (Giglio et al., 2013) and are used to remove harvest residues or to fertilize soils. However croplands show more small fires than large fires (Hantson et al., 2015b). As we here used the GFED burned area datasets that was not corrected for small fires (Giglio et al., 2013), small agricultural fires are likely misrepresented in this dataset and thus cannot be accurately analysed within the SOFIA approach. The representation of agricultural fires in a global fire model needs to account for various land use patterns and practices that go far beyond natural climate-vegetation relationships (Le Page et al., 2015; Magi et al., 2012; Rabin et al., 2015). By taking into account this complexity, agricultural fires are often not represented in global vegetation-fire models because they do not directly affect natural vegetation and carbon cycle dynamics (Hantson et al., 2016), unless agricultural fires escape to nearby forests (Cano-Crespo et al., 2015). In summary, an improved representation of human effects on fire in global vegetation-fire models is currently lacking since globally consistent, temporally and spatial resolved, relevant information on infrastructure and socioeconomics is not available."

*Additionally, as you mention in the Discussion, you do not mimic the shape of the function that relates pop. Density with ignitions (e.g. Pechony & Schindell 2009), therefore excluding pop. density is most likely the problem of the model rather the problem of the variable. This in fact trou- bles me because I can make the same question for several other "effects": is the lack of importance of an effect \ variable the consequence of having bad variables and \ or models?*

We read the reviewer's statement carefully, but unfortunately cannot agree and see the constellation rather different. In fact, we did not mention this in the Discussion. Indeed, we identified a global decrease of burned area with increasing population density which is in agreement with independent studies (Andela et al., 2017; Bistinas et al., 2014; Knorr et al., 2014) (Figure R 3). Please also note that the SOFIA approach does not explicitly separate fire ignition.

The importance of a variable in our analysis based on SOFIA models is affected by the usage of a variable within a certain model structure, by the observational error of this variable, and by the real-world relevance of this variable for fire activity. The effect of model structures on variable importance is compensated by comparing an ensemble of different model structures. The effects of observational error and real-world relevance of a certain variable on the

importance of a variable for model performance is a general challenge in environmental modelling (Beven, 2007; Gupta et al., 2012; Jakeman et al., 2006; Williams et al., 2009) and is not specific to our analysis. We referred to these issues already in our previously proposed changes in response to comments #1 and #5.

[Figure]

*Figure R 3: The response function from the best SOFIA model including population density (SF.314511) globally shows a decline of fire activity with increasing population density, a finding which is in agreement with independent studies (Andela et al., 2017; Bistinas et al., 2014; Knorr et al., 2014).*

*6) The comparisons of the impact of controlling effects and predictor variables on model performance made along the work are troublesome. I highlight one clear example. The land cover grouping scheme \ classes is especially relevant for your modelling approach, since it controls model performance and model complexity. The analysis that you made does not help the reader to reach a clear conclusion. In one hand, PFTs lead to the best IoAs, on the other hand the additional complexity is extremely penalized. Why don't you make a simple a robust comparison with the same (or similar) degree of complexity? Do you need all classes in all land cover schemes?*

Unfortunately, we do not fully understand this comment and the provided example. Contrary to the statement of the referee, our results show that IoA does not depend on the used land cover grouping scheme but we found good performing models in all land cover grouping schemes (Fig. 2 a, Fig. 3 a).

However, the referee is right that our analysis does currently not provide much insight on the question how land cover should be grouped or what land cover classes are required to accurately estimate fire activity. To address this question, we computed another random forest (RF) model only based on the PFT grouping scheme (RF.PFT, i.e. without any variables that account for weather conditions, vegetation conditions, or human influences). With RF.PFT we aim to predict spatial patterns (but not temporal dynamics) of burned area from the spatial distribution of vegetation types. The importance of variables in this RF model provides insight which land cover classes are most important to predict global patterns of burned area. Globally, the most important land cover variables were the coverage of broad-leaved deciduous shrubs (Shrub.BD), broad-leaved evergreen trees (Tree.BE), and broad-leaved

deciduous trees (Tree.BD) (Figure R 4). Of lowest importance were the coverages of crops (Crop), needle-leave deciduous trees (Tree.ND), and the ratio of croplands to other vegetation (CropRatio). Again, the low importance of croplands in the RF model might be surprising given the real importance of fire use in croplands but it is a result of the complex, non-monotonic, relations between cropland cover and burned area (Figure R 5). On the contrary, the dependency of burned area on the coverage of broad-leaved deciduous shrubs (which were of highest importance) was mostly monotonic and similar among biomes (Figure R 5). These patterns suggest that cropland cover is an insufficient predictor variable to represent agricultural fire use.

[Figure]

*Figure R 4: Importance of variables in a random forest model to predict burned area only from the spatial distribution of land cover classes as expressed by PFTs (RF.PFT). Variables with a higher percentage increment in mean squared error are of higher importance.*

[Figure]

*Figure R 5: Partial dependencies of burned area to the fractional coverage of broad-leaved deciduous shrubs (left) and croplands (right) based on a random forest model that is only based on PFTs (RF.PFT). Colours refer to biome-based regions as shown in Figure A1 of the manuscript.*

*7) The way conclusions are drawn needs to be moderated. You have a "virtual lab" called SOFIA that can help make some experiments, it will not inform you about the truth, certainly not about what really determines fire activity at the global scale. An example (there are many) is related with the way you interpret the importance or not of separating crops from herbaceous. The only implication is on model performance...not on reality itself. Fires occur in croplands and in herbaceous areas for different reasons, independently of what the importance of the variables or controlling functions points out.*

The term "virtual lab" is a very nice description of the SOFIA approach. Sometimes variables might be a good proxy for something that is not well observed. The cropland fraction for instance has been suggested in a linear modelling approach to be an indicator of landscape fragmentation and therefore increases in cropland fraction can reduce fire occurrence (Bistinas et al., 2014). Although such relations are maybe not "true", as it is not the cropland itself that is reducing the fire, it can still be valuable for improving the model's performance. However, our results modify this finding because we show the complexity of cropland-burned area relations by using non-linear modelling approaches. As models are always a simplification of reality, we fully agree that the SOFIA approach or specific SOFIA models cannot inform us about the reality of fire activity. However, we also never claimed that our SOFIA-based results are a description of fire in reality. Especially throughout chapter 4.4 and 5.1, we mention that results are based on the SOFIA model SF.124421 and thus model-dependent.

*8) Some contradictory results : RF is better globally although you state that SOFIA outperforms both RF and JBACH. Fig 2 shows clear improvements depending on land cover grouping scheme which is contradicted in L523-524.*

The referee likely misunderstood our results. With respect to the first example, RF is only better than SOFIA models because it can use a lot of variables or datasets. However only the RF model RF.124421 is comparable with a SOFIA model because they use the same set of predictor variables (RF.124421 and SF.124421). In this comparison, RF.124421 and SF.124421 reached similar IoA but SF.124421 reached slightly better FV in optimization (Tab. 2). SF.124421 also better reproduced spatial-temporal patterns of annual total burned area at regional scales than RF.124421 (Figure 5). Moreover, the regional comparisons also demonstrate that SOFIA models better reproduce means, distributions, and had higher IoA than more complex RF models (RF2) in many regions (Figure 5). Based on these results, we indeed conclude that SOFIA outperforms the RF models considered in this study.

Also the reviewer's comment on Figure 2 is not quite correct. We cannot see any clear dependency of the SOFIA performance on the used land cover grouping scheme (Fig. 2a).

*9) The stratified sampling is necessary and interesting, but confusing. It looks like you stratified in space, per biome (which biomes??), and then on the level of fire activity. Finally you sample in time, correct? I am not sure I totally understood the methodology I suggest you improve the way it is written and take a look at Boschetti et al 2015.*

We first computed for all grid cells the maximum annual burned area for all grid cells to stratify the sampling. We then used the regions (biomes) as shown in Fig. A1 (a) to further stratify the sampling. For each region, we determined a number of grid cells that we want to sample based on the area of the region. Then, we computed quantiles ((minimum = q0) < q0.01 < q0.02 < … < q0.99 < (q1 = maximum), increment by 0.01) of the regional annual maximum burned area. We assigned a quantile class (100 classes, e.g. first class for q0 to q0.01) to each grid cell of a region. We then randomly sampled grid cells for each quantile class. These sampled grid cell were then divided into subsets for model training and evaluation as described at lines 355-363. We rewrote the paragraph at lines 343-354:

"We sampled several grid cells from the global datasets (0.25° resolution) to optimize and evaluate all candidate SOFIA models. A sampling of grid cells is necessary to retain enough independent data for evaluation of SOFIA models and because optimization of all SOFIA models on the entire global datasets with 0.25° spatial resolution, monthly time steps, and 15 years was computationally not feasible. However the sampling needs to represent the global spatial patterns and the entire statistical distribution of burned area, including extreme fire events. Therefore, we performed a sampling of grid cells stratified by regions (representing biomes) and by the statistical distribution of burned area. We first computed the maximum annual burned area for all grid cells in 1997-2011 to represent the spatial distribution of extreme fire years. Regions were defined based on land cover and climate zone (Kottek et al., 2006) (Figure A 2). For each region, we classified the annual maximum burned area of each 0.25° grid cell into 100 classes according to regional quantiles of the maximum annual burned area (e.g. class 1 covers quantile 0 (minimum) to quantile 0.01 and the last class covers quantile 0.99 to 1 (maximum) of regional annual maximum burned area). We then randomly sampled grid cells for each regional quantile class. In total, 3161 grid cells were sampled with most of the cells in savannahs and tropical croplands (n = 953, largest region) and fewest cells in boreal

needle-leaved deciduous forests (n = 135, smallest region) (Figure A 2 b). Consequently, the sampled grid cells are representative for the global statistical distributions (Figure A 2 c-e) and for spatial patterns of fire activity (Figure A 2 f)."

*10) The abstract needs to be completely rewriten, so that the reader can undersand the relevance of the problem, what limitations you identified and are going to tackle, how are you going to tackle these limitations (really missing this one), your major results and finally implications for the evolution of the body of knowledge on this area.*

We acknowledge that the abstract is not written in a style that is common for most research articles and therefore may be a bit confusing. This is because our study is a model description paper, where we describe the SOFIA modelling approach, compare it to other models and show a particular application. An increase in the scientific knowledge is not required for articles in GMD. However, we propose to rewrite the abstract in the following way to make the aims of our study more clear:

"Vegetation fires affect human infrastructures, ecosystems, global vegetation distribution, and atmospheric composition. However, the climatic, environmental and socioeconomic factors that control global fire activity in vegetation are only poorly understood and in various complexities and formulations represented in global process-oriented vegetation-fire models. Data-driven model approaches such as machine learning algorithms have successfully been used to identify and better understand controlling factors for fire activity. However, such machine learning models cannot be easily adapted or even implemented within process-oriented global vegetation/fire models. To overcome this gap between machine learning-based approaches and process-oriented global fire models, we here introduce a new flexible data-driven fire modelling approach (Satellite Observations to predict FIre Activity, SOFIA approach version 1). SOFIA models can use several functional relationships between predictor variables and burned area that can be easily adapted with more complex process-oriented vegetation-fire models. We created an ensemble of SOFIA models to test the importance of several predictor variables. Models result in the highest performance in predicting burned area if they account for a direct restriction of fire activity at wet conditions and if they include a land cover-dependent restriction or allowance of fire activity by vegetation density and biomass. The use of vegetation optical depth data from microwave satellite observations, a proxy for vegetation biomass and water content, reaches higher model performance than commonly used vegetation variables from optical sensors. We further analyse spatial patterns of the sensitivity between human, climate, and vegetation predictor variables and burned area . We finally discuss how multiple observational datasets on climate, hydrological, vegetation, and socioeconomic variables together with data-driven modelling and model-data integration approaches can guide the future development of global process-oriented vegetation-fire models."

*Some Minor Commnts*

*11) There are Several terms in the text need to be revised: - fire suppression: Moisture does not suppress fires, it contrains \ reduces \ restricts fire activity (see Bradstock et al 2010) Firefighters suppress fires. - You mention several times "before a fire". You are modeling the occurrence of a fire, if it doesn't happen you still need predecent conditions, so I suggest you drop the term.*

We will use the terms "precedent months" and "restriction".

*12) NLDIg adds extra parameters depending on the number of land cover groups. Did you consider this in your 100 parameters threshold? Did you consider this in the AIC calculation? Is the calculation shown in L302-303 correct?*

The number of extra parameters in NLDI.g were considered in the threshold and in the computation of AIC. The calculation of the number of parameters is correct.

*13) Figure 2: Number of variables or number of controlling factors? Maybe I missed something, but each controlling actor can have multiple variables? (e.g. temperature can have both DTR and annual temp?). If not, as I originally understood, then I don't understand that does "number of variables" mean in Figure 2.*

Each controlling factor can have only one predictor variable. This implies that the "number of variables" equals the "number of controlling factors".

*Some suggestions:*

*- A pairwise plot between estimated and GFED \ CCI burned area*

Pairwise or scatterplots are an intuitive approach for the evaluation of a single model against a reference dataset. However, we need to compare two burned area datasets with four best-performing SOFIA models, two random forest models, and with JSBACH-SPITFIRE. This already results in 14 scatterplots. Additionally, we also want to compare model performance for different regions which further increases the number of scatterplots (14 scatterplots * 7 regions = 98 scatterplots). Thus scatterplots are not an adequate approach for model evaluation given this complexity. Therefore we developed the plot type as in Figure 5 (based on (Phillips, 2017)) which allows us to compare regional statistical distributions, mean values, and the IoA.

*- Maps of differences in estimated-observed burned area - A map showing disagreements between GFED and CCI would help frame the evaluation better. (low expectation on areas where both disagree)*

Maps can either visualize disagreements in spatial patterns (i.e. difference in mean annual burned area between two datasets), temporal dynamics (i.e. correlation or IoA between two datasets), or statistical distributions (e.g. Kolmogorov-Smirnov statistic or FV between two datasets). We already combined all of this information in Figure 5 that does not only serve for the evaluation of SOFIA models but also clearly highlights regional disagreements between the GFED and CCI datasets.

*At this stage I don't think it makes since to go into gramar details.*

Copernicus provides a professional language correction service.

*The references I mentioned previsouly:*

*Le Page et al. 2010. Seasonality of vegetation fires as modified by human action: observing the deviation from ecoãAˇRclimatic fire regimes*

*Magi et al 2012. Separating agricultural and non-agricultural fire seasonality at regional scales*

*Le Page et al 2015. HESFIRE: a global fire model to explore the role of anthropogenic and weather drivers.*

*Li et al 2013 Quantifying the role of fire in the Earth system – Part 1: Improved global fire modeling in the Community Earth System Model (CESM1),.*

*Bradstock 2010. A biogeographic model of fire regimes in Australia: current and future implications (the "switches" that control fire);*

*Boschetti et al 2016. A stratified random sampling design in space and time for regional to global scale burned area product validation*

*Beven & Binley, 1992. The future of distributed models: model calibration and uncertainty pre- diction.*

*Beven, 2002. Towards a coherent philosophy for modelling the environment.*

*Beven & Binley, A., 2014. GLUE: 20 years on.*

*Pettinari & Chuvieco 2016. Generation of a global fuel data set using the Fuel Characteristic Classification System*

*Rabin et al. 2015 Quantifying regional, time-varying effects of cropland and pasture on vegetation fire (the effects of crops on burned area)*

*Broxton et al. 2014. A Global Land Cover Climatology Using MODIS Data (for Discussion)*

*Korontzi et al. 2006. Global distri- bution of agricultural fires in croplands from 3 years of Moderate Resolution Imaging Spectroradiometer (MODIS) data.*

**A data-driven approach to identify controls on global fire activity from satellite and climate observations (SOFIA V1)**

[revised manuscript text omitted]

Satellite observations of fire activity can be further integrated with fire models to estimate model parameters or to assess the adequacy of functional relationships (Knorr et al., 2014; Lasslop et al., 2015; Le Page et al., 2015). For example, parameters of empirical relations were optimized in SIMFIRE (simple fire model) to predict annual fire frequency from vegetation conditions, fire weather conditions, and population density (Knorr et al., 2014). Such parameter optimization approaches are one aspect of model-data integration or model-data fusion that encompasses a continuous cycle from the definition of model structures (i.e. predictor variables and functional relationships), estimation of model parameters, generalization or upscaling of the model, evaluation of model results, to model application and potentially back to a reformulation of the model structure (Keenan et al., 2011; Williams et al., 2009). However, a full model-data integration cycle has been rarely applied in the development of global fire models.

In comparison to process-oriented global vegetation-fire models, data-driven approaches provide an alternative framework to understand and model climate, vegetation, and socioeconomic controls on fire activity. While the development of mathematical and computational process-oriented vegetation-fire models usually starts from a conceptual model (Gupta et al., 2012), data-driven approaches aim to derive mathematical and computational models directly from the data (Solomatine and Ostfeld, 2008). In data-driven approaches, algorithms from artificial intelligence (e.g. neural networks), machine learning (e.g. random forest), or evolutionary algorithms (e.g. genetic optimization) are applied to predict a response variable (here burned area, or fire counts) from a set of potential predictor variables (Solomatine and Ostfeld, 2008). If an adequate data-driven model has been derived, the importance of individual variables and the sensitivities of the response variable to the predictor variables allow to develop a conceptual model about the studied system (Solomatine and Ostfeld, 2008). In global fire modelling, data-driven fire models have been developed using machine learning algorithms such as generalized linear models (Bistinas et al., 2014), maximum entropy (Parisien et al., 2016), or random forest (Aldersley et al., 2011; Archibald et al., 2009) mainly to

identify controls on fire activity. However, such machine learning models often have complex structures, are seen as "black boxes", and thus cannot be easily adapted or even implemented within process-oriented global vegetation/fire models. Alternatively, empirical fire models like SIMFIRE (Knorr et al., 2014) could be generalized to integrate several different candidate predictor variables and to then assess the importance and functional relationships. Consequently, such a flexible data-driven but functional fire modelling approach would allow exploring different predictor variables similar as in machine learning algorithms while potentially revealing model structures that can be more easily adapted for process-oriented vegetation-fire models.

Satellite observations provide several datasets on vegetation and moisture conditions that can be used as predictor variables in data-driven fire models. Time-variant biomass datasets would be the first choice to represent fuel loads in empirical fire models because the availability of fuel is a prerequisite for fire activity (Krawchuk and Moritz, 2011). However, current global biomass maps are static (Avitabile et al., 2016; Saatchi et al., 2011; Thurner et al., 2014) and thus provide only limited information for fire modelling. Consequently, other proxies of vegetation biomass such as model-based net primary production (NPP) (Bistinas et al., 2014; Moritz et al., 2012), satellite-derived vegetation cover (Bistinas et al., 2014; Lehsten et al., 2010), or the fraction of absorbed photosynthetic active radiation (FAPAR) (Knorr et al., 2014) have been used as proxies for fuel loads in global empirical fire models. As an alternative, satellite retrievals of vegetation optical depth (VOD) might be used as proxy for fuel loads. VOD is a vegetation variable that is derived from active or passive microwave satellite observations and is related to vegetation density and water content (Liu et al., 2011b, 2013a, Vreugdenhil et al., 2016a, 2016b). VOD has a higher sensitivity to forest biomass than FAPAR (Andela et al., 2013) and was used to estimate temporal changes in biomass (Liu et al., 2015). Thus VOD might be a valuable predictor variable for the biomass-driven variability in fire activity. Satellite datasets of surface soil moisture might be valuable proxies for the moisture of surface fuels in empirical fire models (Krueger et al., 2015, 2016) because they represent the top ~3 cm of the soil (Dorigo et al., 2015). Such datasets might potentially provide useful information for empirical fire models to represent fuel loads, fuel moisture, or fire weather conditions.

Here, we aim to describe and apply a flexible data-driven fire modelling approach called SOFIA (Satellite Observations for FIre Activity). The SOFIA approach provides a framework to identify the importance and the functional relationships between observational datasets and the spatial and temporal variability of burned area while revealing model formulations that could be easily adapted for more complex vegetation-fire models. We test the approach using observational datasets of land cover, climate conditions, soil moisture, vegetation state, and socioeconomics. Based on the philosophy of model-data integration, we generated several different candidate model structures, and optimized and evaluated each model against observed burned area time series. Additionally, we simulated global burned area with the random forest machine learning approach and with a process-oriented vegetation-fire model (JSBACH-SPITFIRE) to compare the performance of the derived SOFIA models with two independent state-of-the art data-driven and process-oriented modelling approaches, respectively. We used random forest to test if a more flexible modelling approach than SOFIA results in higher performances. In comparison to random forest, SOFIA has the advantage that it could be easily transferred to or implemented in global process-oriented vegetation-fire models. The SPITFIRE fire module within the JSBACH (Jena scheme for biosphere-atmosphere coupling in Hamburg) land surface model (Lasslop et al., 2014; Rabin et al., 2016) was used to compare SOFIA results with a global process-oriented vegetation-fire model.

We first describe the observational datasets and the derived variables that we used to develop SOFIA models (Sect. 2). Secondly, we describe the SOFIA approach and the JSBACH-SPITFIRE and random forest modelling approaches (Sect. 3). In Section 4, we first present the global performance and complexity of SOFIA models (Sect. 4.1) and how several predictor variables contribute to model performance (Sect. 4.2). Then we compare the best performing SOFIA models globally against random forest and JSBACH-SPITFIRE (Sect. 4.3) and apply the best SOFIA model to explore spatial patterns of the sensitivity between predictor variables and burned area (Sect. 4.4). Finally, we discuss the performance and equifinality of our results (Sect. 5.1), the importance of certain predictor variables for global fire modelling (Sect 5.2), and suggest the use of multiple

datasets, data-driven modelling and model-data integration approaches to improve global process-oriented vegetation-fire
250 models (Sect. 5.3).

**2 Datasets and predictor variables for model development**

We used datasets of global monthly burned area as response variable and several datasets on land cover, climate, soil moisture, vegetation state, and socioeconomic factors as predictor variables in model development. To make a pre-selection of relevant predictor variables, we first tested the predictive performance of various candidate variables such as absolute values,
255 anomalies, or long-term precedent mean values of precipitation, wet days, soil moisture, or vegetation state using random forest (Figure A1). We generally found a higher importance of the absolute variables than of the anomalies. For the development of SOFIA models, we finally selected a set of candidate predictor variables based on their importance, their interpretability, and based on how closely they are related to fire activity (by avoiding variables that account for indirect effects) (
[revised manuscript text omitted]
., 2014). DTR has been long used as predictor for fire weather conditions because it is sensitive to stable weather conditions that are usually associated to low humidity and are supportive for fire activity (Bistinas et al., 2014; Venevsky et al., 2002). These datasets provide monthly climate time series at 0.5° resolution based on spatially interpolated weather station observations. Precipitation was taken from the Global Precipitation Climatology Center (GPCC) version 7 dataset (Schneider et al., 2015). All climate datasets were resampled to 0.25° using the nearest neighbour method in order to avoid smoothing of climate anomalies through alternative resampling methods such as bilinear interpolation.

[revised manuscript text omitted]

430    can take step-wise, linear, sigmoidal, or exponential shapes depending on the parameters of the logistic functions (Figure 1). Similar model structures like SOFIA where a response variable is controlled by a product of several functions have been previously applied in environmental modelling for example in light-use efficiency models to simulate NPP (Cai et al., 2014; Nemani et al., 2003) or in phenology models to simulate leaf development (Forkel et al., 2014; Jolly et al., 2005; Stöckli et al., 2011). The response value of the functional relationship can also be used to map sensitivities of burned area to environmental

435    or socioeconomic variables. Such a mapping of controls was previously done for plant productivity (Nemani et al., 2003) and phenology (Forkel et al., 2014; Jolly et al., 2005) based on red-green-blue (RGB) composite maps. Here we will demonstrate how this approach can be used to investigate spatial patterns of sensitivities between burned area and climatic, environmental and socioeconomic controls on fire activity.

| Deleted: | potentially appropriate |
| Deleted: | allows |
| Deleted: | ) and to explore |
| Deleted: | response functions of fire activity to environmental or socioeconomic variables that |
| Deleted: | (Cai et al., 2014; Nemani et al., 2003) |
| Deleted: | values |
| Deleted: | control functions |
| Deleted: | the spatial covariation |
| Deleted: | with |
| Deleted: | (Nemani et al., 2003) |
| Formatted: | Danish |
| Field Code Changed | |
| Formatted: | Danish |
| Formatted: | Danish |

[Figure]

**Figure 1: Example of a SOFIA model structure with three land cover groups (i.e. herbaceous vegetation and crops, shrubs, trees) and five controlling factors on fire activity. The example is taken from the SOFIA model SF.124421 (Table 2). (a) Histogram of the simulated fractional burned area. Response functions of fractional burned area on (b) night light development index, (c) diurnal temperature range, (d) number of wet days, (e) fraction of absorbed photosynthetic active radiation in the month before a fire, and (f) mean vegetation optical depth in the 12 precedent months. max, sl, and x0 are parameters of the logistic functions.**

**3.2 Testing controlling factors and predictor variables in SOFIA models**

To test appropriate controlling factors and related predictor variables in SOFIA models, we defined several alternative model structures. 
[revised manuscript text omitted]
 statistical distribution of burned area. We first computed the maximum annual burned area
560 for all grid cells in 1997-2011 to represent the spatial distribution of extreme fire years. Regions were defined based on land
cover and climate zone (Kottek et al., 2006) (Figure A 2). For each region, we classified the annual maximum burned area of
each 0.25° grid cell into 100 classes according to regional quantiles of the maximum annual burned area (e.g. class 1 covers
quantile 0 (minimum) to quantile 0.01 and the last class covers quantile 0.99 to 1 (maximum) of regional annual maximum
burned area). We then randomly sampled grid cells for each regional quantile class. 
[revised manuscript text omitted]

than the corresponding SOFIA model. Thus the highly flexible structure of the random forest machine learning approach did not necessarily result in a much better performance than the best-performing SOFIA models. Consequently, the SOFIA approach offers enough flexibility to assess different controlling factors and its functional relationships to predict burned area.

[Figure]

**Figure 3: Effect of** controlling factors and associated predictor variables in **SOFIA models on the performance in simulating global monthly burned area dynamics. Performance is expressed as the index of agreement between simulated and observed (GFED) monthly burned area for the training data subset. Boxplots show the distribution of IoA based on all SOFIA model experiments that include the respective variable. Star symbols indicate a significantly higher IoA of a variable in comparison to the "no" group of each controlling factor (Wilcoxon rank sum test, p ≤ 0.05). Distribution of IoA depending on the used (a) land cover grouping scheme; and variables to account for (b) human influence; (c) temperature effects; (d) direct wetness effects; (e) direct vegetation effects; and (f) long-term wetness or vegetation effects. The best models (IoA > 0.4 and AIC < 200) are highlighted with coloured dots.**

**4.2 Required** controlling factors and adequate predictor variables in **SOFIA models**

The performance of SOFIA models depended on the controlling factor and associated predictor variables that were used in model structures (Figure 3). The choice of a certain land cover grouping scheme in SOFIA models to regionalize model parameters had only weak effects on model performance (Figure 3a). Although models based on the GrowthForm scheme had on average weaker performances than models based on land cover grouping schemes with croplands, the best SOFIA models were not related to a certain land cover grouping scheme.

Including human influences as controlling factors in SOFIA models did not improve model performance (Figure 3b). The best models either did not consider human influences or considered human influences through NLDI as global controlling function. However, NLDI did in average not contribute to higher performances. SOFIA models that used population density had on average weaker performance than SOFIA models that used NLDI or that did not consider human influences. The weaker performance of population density as component in SOFIA models could be caused by the general model structure in which

potential burned area equals the total vegetated area: As highly populated areas are usually associated with low vegetation cover, potential burned area is low as well, and thus population density does not provide further information. Although two of the best SOFIA models did not contain any variable for human influences (SF.204422, SF.203512), they however considered the fractional coverage of croplands in the used land cover grouping scheme. Consequently, these two models considered human influence on fire indirectly through the coverage of croplands. These results suggest that human influences on fire activity can be relatively interchangeably described in SOFIA models by the coverage of croplands, NLDI, or population density.

Considering temperature variables in SOFIA models caused on average better model performances than model structures without temperature variables (Figure 3c). However, we also found one model without a temperature control that reached good performance (SF.233210, Table A 2). All of the best performing models included diurnal temperature range or pre-fire annual mean temperature as controlling factors. These results show that temperature-related variables are important predictors in SOFIA.

The consideration of direct wetness effects in SOFIA models had the largest positive impact on model performance (Figure 3d). Models that did not consider direct wetness effects had lower performances than models that used soil moisture, precipitation, or the number of wet days. Especially models based on the number of wet days reached significant higher IoA than models without direct wetness effects (Wilcoxon rank sum test, $p \leq 0.05$). Consequently, direct wetness effects on fire activity were a required component of SOFIA models to predict burned area.

Whether or not including direct vegetation controls did not lead to a significant change in performance of the SOFIA models (Figure 3e). The best models either did not consider direct vegetation effects (SF.324202), used pre-fire FAPAR (SF.204422, SF.124421), or pre-fire VOD (SF.203512). This suggests that precedent FAPAR and VOD conditions did not provide additional information to predict burned area in SOFIA models.

On the contrary, considering long-term wetness or vegetation effects in SOFIA models caused significantly higher model performances than not considering these effects (Figure 3f). Especially SOFIA models that used pre-fire annual precipitation or VOD reached significantly higher IoA. Models with long-term effects based on soil moisture, the number of wet days, or FAPAR had on average similar performances as models without long-term effects. However, we also found some good models that used long-term conditions of FAPAR (e.g. SF.203512). These results demonstrate that long-term conditions in vegetation productivity (reflected by annual precipitation) or vegetation structure (reflected by VOD or FAPAR) were required components of SOFIA models to predict burned area.

Based on the performances of the different controlling factors and associated predictor variables, the ideal SOFIA model should include NLDI as human influence, one variable to account for temperature effects, the number of wet days as direct wetness effect, and pre-fire annual conditions of precipitation or VOD as long-term wetness/vegetation effects. This ideal model structure is realized in two of the best performing SOFIA models (SF.124421 and SF.324202, Figure 3). The choice of a certain land cover grouping scheme or of a direct vegetation effect are secondary components of SOFIA model structures. The distribution of model parameters in SF.124421 after optimization reflects that parameters for the functional relationships with NLDI, the number of wet days, and VOD were well constrained and thus were the most sensitive parameters within this model to estimate global monthly burned area dynamics. These parameter estimates and distributions could be potentially used as prior parameter estimates to further constrain SOFIA models.

[revised manuscript text omitted]

**4.4 Sensitivity of burned area to climate, vegetation, and human predictor variables**

The underlying functional relationships in SOFIA models allow to map the sensitivities of burned area to human, vegetation, and climate variables. To demonstrate such a potential application of a SOFIA model, we mapped mean responses from each functional relationships for the period 1997-2011 from the SOFIA model SF.124421 (Figure 6). Based on this model, human influences (i.e. NLDI) restricted burned area in most parts of Europe and southern Russia, east and south-east Asia, India, central and eastern North America, south-east South America, south Australia and New Zealand (Figure 6 a). These regions correspond to the most populated and developed regions of the world. This pattern was caused by the underlying functional relationship of SF.124421 where NLDI < 1 (i.e. developed regions) restricted and NLDI > 1 (i.e. unpopulated regions or natural ecosystems) allowed fire activity (Figure 1 b). These results indicate a predominant restricting effect of humans on fire activity. Temperature effects in SF.124421, expressed as diurnal temperature range, allowed fire activity mostly in the semi-deserts of western North America, in the Sahel, Australia, and had a moderate restriction effect in tropical forests and the tundra (Figure 6 b). These spatial patterns were caused by the controlling function that had a strong sigmoidal increase of fire activity with diurnal temperature range in shrublands and allowed moderate fire activity in herbaceous vegetation and croplands (Figure 1 c).

Direct wetness effects, expressed as the number of wet days, generally allowed fire activity in all forest regions and moderately restricted fire activity in the rest of the world (Figure 6 c). The underlying controlling function in SF.124421 showed no sensitivity for forests, a weak positive relation in herbaceous vegetation and croplands, and a strong exponential decrease of fire activity with increasing number of wet days in shrublands (Figure 1 d).

As direct vegetation effect, pre-fire FAPAR restricted fire activity in herbaceous vegetation and croplands of central North America, central Asia, in the northern Sahel, the Kalahari, central Australia, and in parts of South America (Figure 6 d). On the other hand, pre-fire FAPAR supported fire activity mostly in the southern Sahel and northern and eastern Australia. These patterns were caused by a general strong restriction of fire activity with pre-fire FAPAR in herbaceous vegetation and croplands and an exponential increase of fire activity with increasing pre-fire FAPAR in shrublands in SF.124421 (Figure 1 e).

As long-term vegetation effect, 12-month precedent mean vegetation optical depth strongly supported fire activity in central North America, central Asia, the Tibetan plateau, the Sahel, parts of India, the Kalahari, in Australia (except interior), and in northern Patagonia (Figure 6 e). In all other regions, annual VOD had a moderate effect on fire activity in SF.124421. The underlying controlling function in SF.124421 showed an exponential increase of fire activity with annual VOD in shrublands, an exponential decrease with annual VOD in herbaceous vegetation and croplands and a strong restriction across all VOD ranges for trees (Figure 1 f). The diverging responses with annual VOD in shrublands and herbaceous vegetation indicate that fire activity increases with higher vegetation density or biomass in shrublands but decreases with increasing vegetation water content in herbaceous vegetation, respectively. Additionally, the general restriction of fire activity with VOD for trees indicates that fire activity is restricted by vegetation density or high vegetation water content in forests.

We further combined the controlling functions of SF.124421 to investigate combined controls on fire activity. Therefore we created a red-green-blue composite map in which the red channel contains the NLDI functional relationship, the green channel contains the mean of the direct (precedent month FAPAR) and long-term vegetation (12 month precedent VOD) effect, and the blue channel contains the climate effects (mean response of functional relationships to number of wet days and diurnal temperature range) from SF.124421 (Figure 6f). Generally, bright colours in this map indicate a strong restriction of fire activity (small burned area) and dark colours indicate that fire activity is allowed (large burned area). Regionally, different

combinations of socioeconomic, vegetation and climate factors controlled fire activity. Socioeconomic development dominantly restricted fire activity in western North America, and in populated regions of boreal forests (red colours). Vegetation predominantly supressed fire activity in southern boreal and tropical forests (green colours). Primarily climate conditions and secondly socioeconomic development restricted fire activity in semi-deserts of the northern Sahel, central Asia, the Kalahari, and south-western Australia (purple colours). Socioeconomic development and climate equally supressed fire activity in the Mediterranean, India, eastern Asia, and east South America (pink colour). Both socioeconomic development and vegetation conditions supressed fire activity in most parts of Europe, central and eastern North America, and eastern China (yellow/orange colours). Both climate and vegetation conditions supressed fire activity in the tundra and in central Australia (cyan colours). All factors moderately supported fire activity in boreal forests and strongly support fire activity in large parts of the Sahel, southern Africa, northern Australia, and western North America (dark colours). We want to point out that these sensitivities might look different if SOFIA models with alternative but adequate model structures would be applied for such an analysis. However the results highlight that fire activity is controlled by regionally diverse and complex interactions of human, vegetation and climate factors.

[Figure]

**Figure 6: Example of combined climate, vegetation, and human controls on fire activity based on the SOFIA model SF.124421. The maps in (a-e) show the average response value for each functional relationship for the period 1997-2011. High values (1, red) indicate that this factor allows unlimited burning and low values (0, blue) indicate that this factor restricts burning. The map in (f) is a red-green-blue composite of the human influence (map in (a), red channel), the combined direct and long-term vegetation effect (mean of (d) and (e), green channel), and the climate effect (mean of (b) and (c), blue channel). Bright and dark colours indicate a strong restriction and allowance of fire activity, respectively.**

**5 Discussion and conclusions**

**5.1 Performance and equifinality of SOFIA models**

We developed the SOFIA modelling approach as a framework to explore the importance and the functional relationships between different predictor variables and burned area while relying on relatively simple model structures. The best SOFIA models reached globally average performances but outperformed the state-of-the art process-oriented vegetation-fire model

JSBACH-SPITFIRE. We interpret the globally medium and regionally varying performances as current upper limits that can be reached with the used predictor datasets and variables because the more flexible and highly adaptive machine learning algorithm random forest did not achieve much higher performance in the evaluation data subset. These upper limits in model performance might be due to several reasons:

1. Uncertainties in the observations for the predictor and response variables inhibit the development of models with high performance. For example, we found regionally partly large differences between the two burned area datasets, especially in northern regions. These uncertainties originate from differences in sensor characteristics and in the ability of the used algorithms to detect small fires.

2. Other processes and variables are important for the spread of fires but cannot be resolved at the used spatial and temporal resolution. For example, on local to regional scales the spread of fire is controlled by landscape structure and topography whereas climatic controls are usually more important on larger scales (Archibald et al., 2009; Liu et al., 2013b; Parisien et al., 2010). Most of the regional controls can likely not be resolved at the used spatial resolution (0.25°) although this resolution is already higher than the resolution of most global vegetation-fire models. Also wind speed and direction is an important control on the spread of fires on short temporal scales but this effect cannot accurately be represented based on monthly data (Bistinas et al., 2014).

3. There is a lack of global observations that directly represent fuel loads, fuel moisture, or modes of human fire usage. For example, all of the used predictor variables are only proxies for fuel loads (FAPAR or VOD) or fuel moisture (surface soil moisture) but do not directly represent such conditions. Similarly, data on population density or socioeconomic development are used as proxies for human effects on fire but cannot represent the complex social, economic, and cultural practices, and policies of human fire use and management.

The four best SOFIA models reached similar performances in savannas and tropical croplands, and in tropical forests which demonstrates the equifinality in fire modelling. Equifinality, i.e. the presence of multiple adequate models and parameter sets that result in very similar responses, is a general problem in environmental modelling (Beven, 2006). General approaches to avoid equifinal models are the use of multiple datasets of the same variable to account for errors or uncertainties in model forcing or reference data, the testing of different cost functions to constrain certain parameters, the inclusion of prior parameter uncertainties in the cost function, or the application of models to new observational data or under different conditions (Beven, 2006; Beven and Binley, 2014; Williams et al., 2009). In our analysis, we were able to rule out three of four initially equifinal SOFIA models based on the application of these models to the global data and by regional comparisons against two burned area datasets. The results from the optimized SOFIA models allow extracting parameter values and ranges for each functional relationship. To give an example, parameters that control the functional relationship to 1) socioeconomic development (NLDI), 2) diurnal temperature range and to the number of wet days in shrublands, and 3) to VOD were well constrained in the SOFIA model SF.124421 (Figure A3). These parameters could be potentially used as prior parameter values in a more constrained analysis in the future. The presence of equifinality in SOFIA model structures suggests to include such prior parameter uncertainties for each functional relationship to better constrain individual SOFIA models. This technique can be applied in future generation of individual SOFIA models by using the current versions as prior parameter estimates and uncertainties.

**5.2 Importance of predictor variables and implications for global fire modelling**

The derived SOFIA models and the spatial patterns of sensitivities show a sharp decline of burned area with increasing socioeconomic development or population density and thus agree with previous studies that show a primarily negative effect of human activities, population density, or croplands on burned area (Andela et al., 2017; Archibald et al., 2013; Bistinas et al., 2014; Chuvieco and Justice, 2010; Knorr et al., 2014). Strikingly, our results suggest that human effects on global burned area can be expressed by either cropland area, NLDI, or population density but the combination of these factors did not improve

the performances of SOFIA models. These variables serve all as proxies for the negative relationship between humans and burned area but do not directly describe human activities of fire use or suppression. For example, regional studies have shown that various information on infrastructure, land use, and other relevant socioeconomic indicators are important to predict fire activity (Archibald et al., 2009; Arndt et al., 2013; Parisien et al., 2016). However such spatially- and temporally-resolved datasets and assessments are missing for the global scale. Certainly, our results do not imply that croplands are unimportant for the global variability of burned area. Agricultural fires account for around 10% of all global fires (Korontzi et al., 2006) and for around 5% of global burned area (Giglio et al., 2013) and are used to remove harvest residues or to fertilize soils. However croplands show more small fires than large fires (Hantson et al., 2015b). As we here used the GFED burned area datasets that was not corrected for small fires (Giglio et al., 2013), small agricultural fires are likely misrepresented in this dataset and thus cannot be accurately analysed within the SOFIA approach. The representation of agricultural fires in a global fire model needs to account for various land use patterns and practices that go far beyond natural climate-vegetation relationships (Le Page et al., 2015; Magi et al., 2012; Rabin et al., 2015). By taking into account this complexity, agricultural fires are often not represented in global vegetation-fire models because they do not directly affect natural vegetation and carbon cycle dynamics (Hantson et al., 2016), unless agricultural fires escape to nearby forests (Cano-Crespo et al., 2015). In summary, an improved representation of human effects on fire in global vegetation-fire models is currently lacking since globally consistent, temporally and spatial resolved, relevant information on infrastructure and socioeconomics is not available.

[revised manuscript text omitted]

**Structure of SOFIA models: used control factors and associated variables**

Grouping scheme (groups)
1 GrowthForm
2 GrowthFormCrop
3 LeafType
4 PFT

Human influence (human)
0 no
1 PD.med (global)
2 NLDI (global)
3 NLDI.g (per group)

Direct wetness effect (wet.dir)
0 no
1 CCI.SM.orig
2 (unused)
3 GPCC.P.orig
4 CRU.WET.orig

Long-term wetness/productivity effect (wetveg.longterm)
0 no
1 CCI.SM.orig.filter13
2 GPCC.P.orig.filter13
3 CRU.WET.orig.filter13
4 Liu.VOD.orig.filter13
5 GIMMS.FAPAR.orig.filter13

Direct vegetation effect (veg.dir)
0 no
1 Liu.VOD.orig.lagneg1
2 GIMMS.FAPAR.orig.lagneg1

Temperature effect (temp)
0 no
1 CRU.DTR.orig
2 CRU.T.orig.filter13

[revised manuscript text omitted]

1220 **Figure A 1: Importance of several predictor variables to predict monthly burned area using random forest. Importance is expressed as the percentage increment in mean squared error if a certain variable is not included in random forest. Thus, the most important variables cause the largest increment in MSE. Variables that include "orig" or "anom" indicates original absolute values and anomalies (relative to the mean seasonal cycle), respectively. "filterX" indicates mean values over the X precedent months before the actual month for which burned area should be predicted. In total 132 variables were included in this analysis but variables below**
1225 **rank 53 are not shown in this figure).**

[Figure]

Figure A 2: Representativeness of sampled 0.25° grid cells for global patterns of burned area (based on GFED burned data). (a) Spatial distribution of the grid cells of the optimization and evaluation data subsets and regions for regional analyses of results. Regions are TUND (tundra), BONE (boreal needle-leaved evergreen and mixed forests), BOND (boreal needle-leaved deciduous forests), TEFC (temperate forests and croplands), MEDI (Mediterranean regions), STEP (steppes), SAVC (savannahs and tropical croplands), and TRFO (tropical forests). (b) Distribution of mean annual burned area per region from the sampled grid cells. Numbers indicate the number of grid cells per regions. (c-f) Comparison of mean and maximum annual burned between all global grid cells and the sampled grid cells. (c) and (d) distribution of maximum and mean annual burned. (e) Quantiles of mean and maximum annual burned area. (f) Latitudinal gradients of annual burned area. Latitudinal gradients are smoothing splines fitted to the quantile 0.95 of mean and maximum annual burned area, respectively.

[Figure]

Figure A 3: Uncertainty in parameters of the SOFIA model SF.124421 after genetic optimization. Shown are distributions (outlines), mean values (= 1), and confidence intervals (bars) for mean values for each parameter. Plotted are parameters from equally good-performing parameter sets (i.e. > 0.8 * normalized likelihood NLL, NLL = LL / max(LL) with LL = exp(-SSE)).

[revised manuscript text omitted]